# Reprograming of the ubiquitin ligase Ubr1 by intrinsically disordered Roq1 through cooperating multifunctional motifs

Niklas Peters [ID] [1,4], Sibylle Kanngießer [ID] [1,4], Oliver Pajonk [ID] [1], Rafael Salazar Claros [ID] [1,3], Petra Hubbe [ID] [1], Axel Mogk [ID] [2] & Sebastian Schuck [ID] [1✉]

## Abstract

One way cells control the speed and specificity of protein degradation is by regulating the activity of ubiquitin ligases. Upon proteotoxic stress in yeast, the intrinsically disordered protein Roq1 binds the ubiquitin ligase Ubr1 as a pseudosubstrate, thereby modulating the degradation of substrates of the N-degron pathway and promoting the elimination of misfolded proteins. The mechanism underlying this reprograming of Ubr1 is unknown. Here, we show that Roq1 controls Ubr1 by means of two cooperating multifunctional motifs. The N-terminal arginine and a short hydrophobic motif of Roq1 interact with Ubr1 as part of a heterobivalent binding mechanism. Via its N-terminal arginine, Roq1 regulates the ubiquitination of various N-degron substrates and folded proteins. Via its hydrophobic motif, Roq1 accelerates the ubiquitination of misfolded proteins. These findings reveal how a small, intrinsically disordered protein with a simple architecture engages parallel channels of communication to reprogram a functionally complex ubiquitin ligase.

keywords Protein Degradation; Ubiquitin Ligase Regulation; SHRED
**Subject Category** Post-translational Modifications & Proteolysis

## Introduction

The regulation of protein degradation is vital for cell homeostasis. By accelerating or slowing the degradation of certain proteins, cells can tune their activities to changing physiological demands. In addition, cells boost their capacity to degrade aberrant proteins when stress conditions provoke protein misfolding (McShane and Selbach, 2022). In eukaryotes, the degradation of thousands of proteins is carried out by the ubiquitin-proteasome system. Its selectivity is ensured by E3 ubiquitin ligases, which recognize degradation determinants (degrons) in target proteins and attach ubiquitin chains to these proteins as marks for destruction by the proteasome. The activities of ubiquitin ligases are controlled by a variety of mechanisms, such as phosphorylation, auto-ubiquitination, and the interaction with protein or non-protein ligands (Vittal et al, 2015; Buetow and Huang, 2016; Zheng and Shabek, 2017). Unraveling these regulatory mechanisms is essential for understanding how cells orchestrate protein degradation.

A conserved and extensively studied ubiquitin ligase is Ubr1, which is best known for its central role in the N-degron pathway. It determines the half-life of proteins with specific N-terminal amino acid residues and thus controls a large number of processes, including chromosome segregation, inflammation, DNA repair and apoptosis (Varshavsky, 2011; Kim et al, 2021). Ubr1 consists of a single subunit that has a mass of about 200 kilodaltons and comprises a RING domain. The RING domain helps to bind a ubiquitin-loaded E2 ubiquitin-conjugating enzyme and enables the transfer of ubiquitin from the E2 enzyme onto lysine residues in substrate proteins (Deshaies and Joazeiro, 2009; Pan et al, 2021). Ubr1 recognizes N-degron substrates via two well-characterized binding sites, the type-1 site for proteins with positively charged N-terminal residues (type-1 N-degron substrates) and the type-2 site for proteins with bulky hydrophobic N-terminal residues (type-2 N-degron substrates). Furthermore, yeast Ubr1 contains a third, poorly-defined substrate binding site for folded proteins with internal degrons (Turner et al, 2000). These three binding sites communicate with each other so that occupancy of the type-1 and -2 sites allosterically and synergistically promotes substrate recognition by the third binding site (Du et al, 2002). Accordingly, the activity of Ubr1 towards certain endogenous proteins can be stimulated by peptides that bind to the type-1 or -2 sites, and this regulatory system controls peptide uptake in yeast (Turner et al, 2000). A similar system may counteract lipid droplet accumulation in hepatocytes (Zhang et al, 2022). Finally, Ubr1 can also recognize misfolded proteins, presumably through a fourth, yet uncharacterized substrate binding site (Eisele and Wolf, 2008; Heck et al, 2010).

In previous work in yeast, we discovered SHRED (stress-induced homeostatically regulated protein degradation), a regulatory cascade that is activated by proteotoxic stress, alters the substrate specificity of Ubr1 and thereby stimulates the degradation

[1]Heidelberg University Biochemistry Center, 69120 Heidelberg, Germany. [2]Center for Molecular Biology of Heidelberg University, 69120 Heidelberg, Germany. [3]Present address: Medical Research Council Protein Phosphorylation and Ubiquitylation Unit, School of Life Sciences, University of Dundee, Dundee DD1 5EH, UK. [4]These authors contributed equally: Niklas Peters, Sibylle Kanngießer. ✉E-mail: sebastian.schuck@bzh.uni-heidelberg.de

 

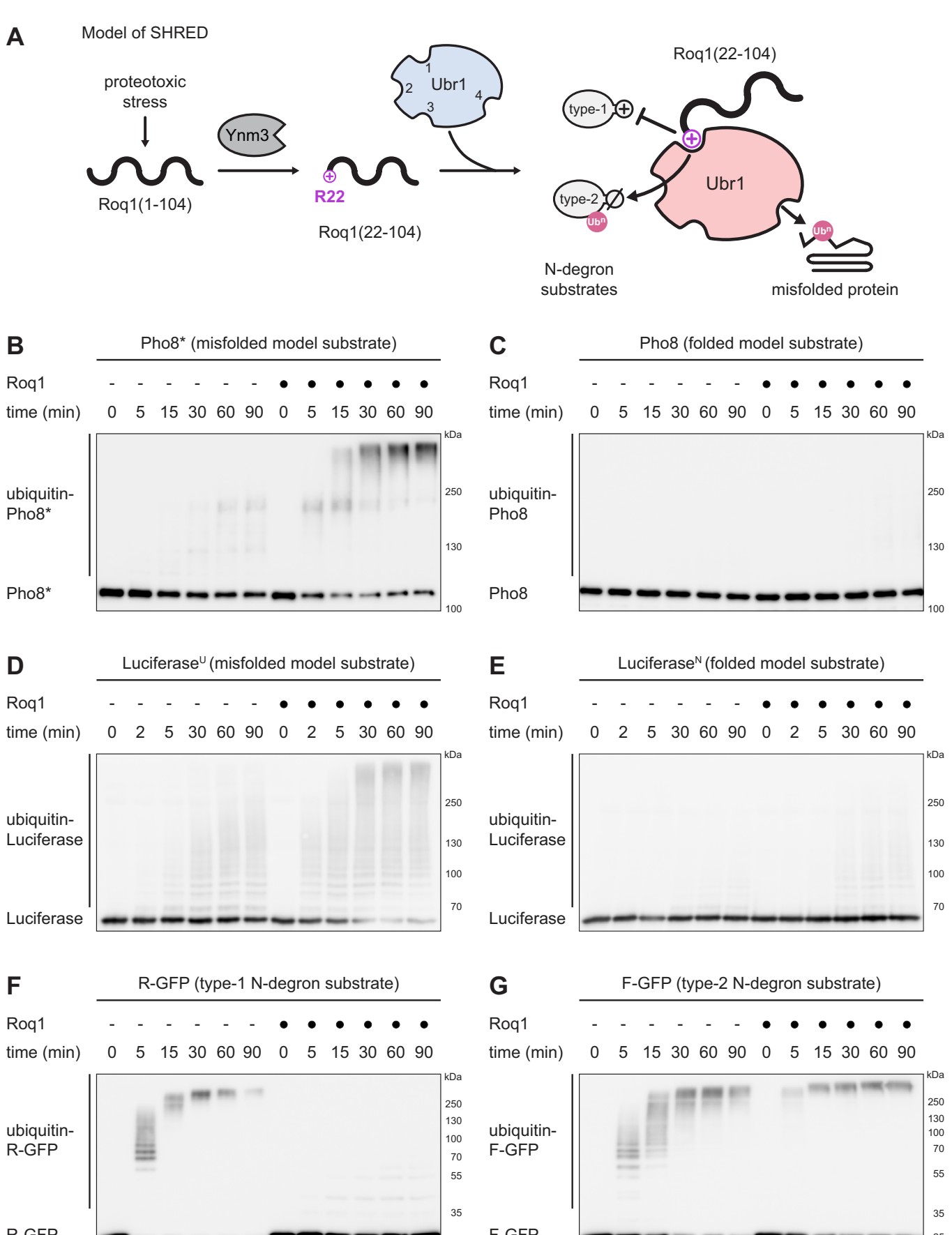

◀

**Figure 1.  Roq1 reprograms Ubr1 substrate specificity in vitro.**

(A) Model of stress-induced homeostatically regulated protein degradation (SHRED). Proteotoxic stress in yeast induces the synthesis of Roq1(1–104), which is cleaved by the protease Ynm3 to generate Roq1(22-104) with arginine-22 (R22) at its new N-terminus. Through R22, Roq1(22-104) binds to the Ubr1 type-1 site as a pseudosubstrate and enhances the degradation of misfolded proteins to restore cell homeostasis. In addition, binding of Roq1(22-104) to the Ubr1 type-1 site competitively inhibits recognition of type-1 N-degron substrates and allosterically promotes recognition of type-2 N-degron substrates. Numbers in unbound Ubr1 (blue) denote the known and putative substrate binding sites for type-1 N-degron substrates (1), type-2 N-degron substrates (2), folded proteins with internal degrons (3) and misfolded proteins (4). $Ub^n$ = ubiquitin chain. (B, C) Western blot of Pho8 from Pho8* or Pho8 ubiquitination assays in the absence or presence of Roq1(22-60) for the times indicated. (D, E) Western blot of Luciferase from Luciferase$^U$ or Luciferase$^N$ ubiquitination assays in the absence or presence of Roq1(22-60) for the times indicated. (F, G) Western blot of GFP from R-GFP or F-GFP ubiquitination assays in the absence or presence of Roq1(22-60) for the times indicated. Source data are available online for this figure.

of misfolded proteins (Fig. 1A; Szoradi et al, 2018). Specifically, various stress conditions that cause protein misfolding induce the synthesis of the small intrinsically disordered protein Roq1. Full-length Roq1 consists of 104 amino acid residues and is rapidly cleaved by the protease Ynm3. The resulting C-terminal cleavage fragment, Roq1(22-104), exposes arginine-22 (R22) at its new N-terminus. By means of the positively charged R22, Roq1(22-104) binds to the type-1 site of Ubr1 as a pseudosubstrate, enhances the ability of Ubr1 to eliminate misfolded proteins and thus increases cellular stress resistance. In addition, Roq1(22-104) competitively inhibits the degradation of type-1 substrates and promotes the degradation of type-2 substrates. Hence, SHRED involves comprehensive reprograming of Ubr1. However, beyond the interaction of Roq1 R22 and the Ubr1 type-1 site, the molecular mechanism of this unusual regulation remains unknown.

In this study, we take an in vitro reconstitution approach to understand Ubr1 reprograming by Roq1. We show that Roq1 differentially modulates the ubiquitination of distinct types of Ubr1 substrates. We then define two cooperating sequence motifs in Roq1 that are needed for binding and regulating Ubr1. Finally, we provide evidence that these motifs engage parallel channels of communication between regulatory and substrate binding sites in Ubr1 to control its substrate specificity.

## Results

### In vitro reconstitution of Ubr1 reprograming by Roq1

To understand the mechanism of SHRED, we reconstituted the reprograming of Ubr1 by Roq1 in vitro. Roq1 cleavage by Ynm3 can be bypassed with Roq1(22-104), which lacks the first 21 residues of full-length Roq1 (Szoradi et al, 2018). We therefore used Roq1(22-104) for reconstitution experiments. To assay Ubr1 activity, we combined purified Roq1(22-104), ubiquitin, the ubiquitin-activating enzyme Ube1, ATP, the ubiquitin-conjugating enzyme Rad6, Ubr1 and potential Ubr1 substrates. However, we noticed that Ubr1 was able to ubiquitinate Roq1(22-104), which, for our purposes, was an unwanted side reaction (Fig. EV1A; Pan et al, 2021). Since Roq1(22-104) does not contain lysines, this ubiquitination likely occurs on serines, threonines or cysteines (Kelsall, 2022). The ubiquitin-Roq1 linkage was resistant to the dithiothreitol used to stop the ubiquitination reactions but was sensitive to mild alkaline hydrolysis (Fig. EV1A). Hence, the ubiquitin-Roq1 linkage must consist of oxyester rather than amide bonds. To focus our analysis on the activity of Ubr1 towards designated substrate proteins, we sought to eliminate the

ubiquitination of Roq1. Shortening Roq1 to Roq1(22-60) diminished ubiquitination (Fig. EV1A) but Roq1(22-60) still supported stress-induced degradation of the SHRED reporter Rtn1Pho8*-GFP in vivo (Fig. EV1B). We therefore employed Roq1(22-60) for in vitro ubiquitination assays and Roq1(22-104) for interaction studies and in vivo experiments.

Using this simplified system, we tested several Ubr1 substrates. First, we analyzed the misfolded model protein Pho8*, which is a SHRED substrate in vivo (Szoradi et al, 2018). Ubr1 barely ubiquitinated Pho8* on its own but did so extensively in the presence of Roq1(22-60) (Fig. 1B). By contrast, Ubr1 did not appreciably ubiquitinate Pho8, the folded counterpart of Pho8*, regardless of the presence of Roq1 (Fig. 1C). Control experiments showed that Pho8* remained mostly soluble throughout the assay (Fig. EV1C) and that its ubiquitination depended on Ubr1 and occurred on lysines (Fig. EV1D,E). Roq1 strongly promoted Pho8* ubiquitination already when present at the same concentration as Ubr1, implying a tight interaction (Fig. EV1F). Equimolar concentrations of Roq1 and Ubr1 roughly corresponded to their relative levels during proteotoxic stress in yeast (Fig. EV1G). Nevertheless, we used a tenfold molar excess of Roq1 as a standard condition for in vitro assays. In this way, we aimed to ensure that we analyzed Roq1-bound Ubr1 rather than a mixture of Roq1-bound and Roq1-free Ubr1, even when we tested Roq1 variants with reduced affinity for Ubr1. Roq1(22-104) also enhanced Pho8* ubiquitination but less efficiently than Roq1(22-60) (Fig. EV1H). This may be explained by the simultaneous ubiquitination of Roq1(22-104), which diverts some Ubr1 from Pho8*. Next, we tested Luciferase, a destabilized version of which is a SHRED substrate in vivo (Szoradi et al, 2018). Roq1(22-60) stimulated Ubr1-mediated ubiquitination of unfolded Luciferase$^U$ and had a much weaker effect on native Luciferase$^N$ (Fig. 1D,E). Thus, Roq1 selectively promotes Ubr1-mediated ubiquitination of misfolded or unfolded proteins.

As a second type of substrate, we examined natural, well-folded proteins that are recognized by Ubr1 because of internal degrons. These were the transcriptional repressor Cup9, the DNA alkyltransferase Mgt1 and the kinase Chk1 (Turner et al, 2000; Hwang et al, 2009; Oh et al, 2017). Roq1(22-60) stimulated Ubr1-mediated ubiquitination of all three proteins (Fig. EV2). As a third type of substrate, we investigated proteins that are recognized by Ubr1 as part of the N-degron pathway. We showed previously that, in vivo, cleaved Roq1 binds to the Ubr1 type-1 site by means of its N-terminal R22 and thereby hampers the recognition of other proteins with positively charged N-terminal residues (Fig. 1A; Szoradi et al, 2018). In agreement with these findings, Roq1(22-60) inhibited the in vitro

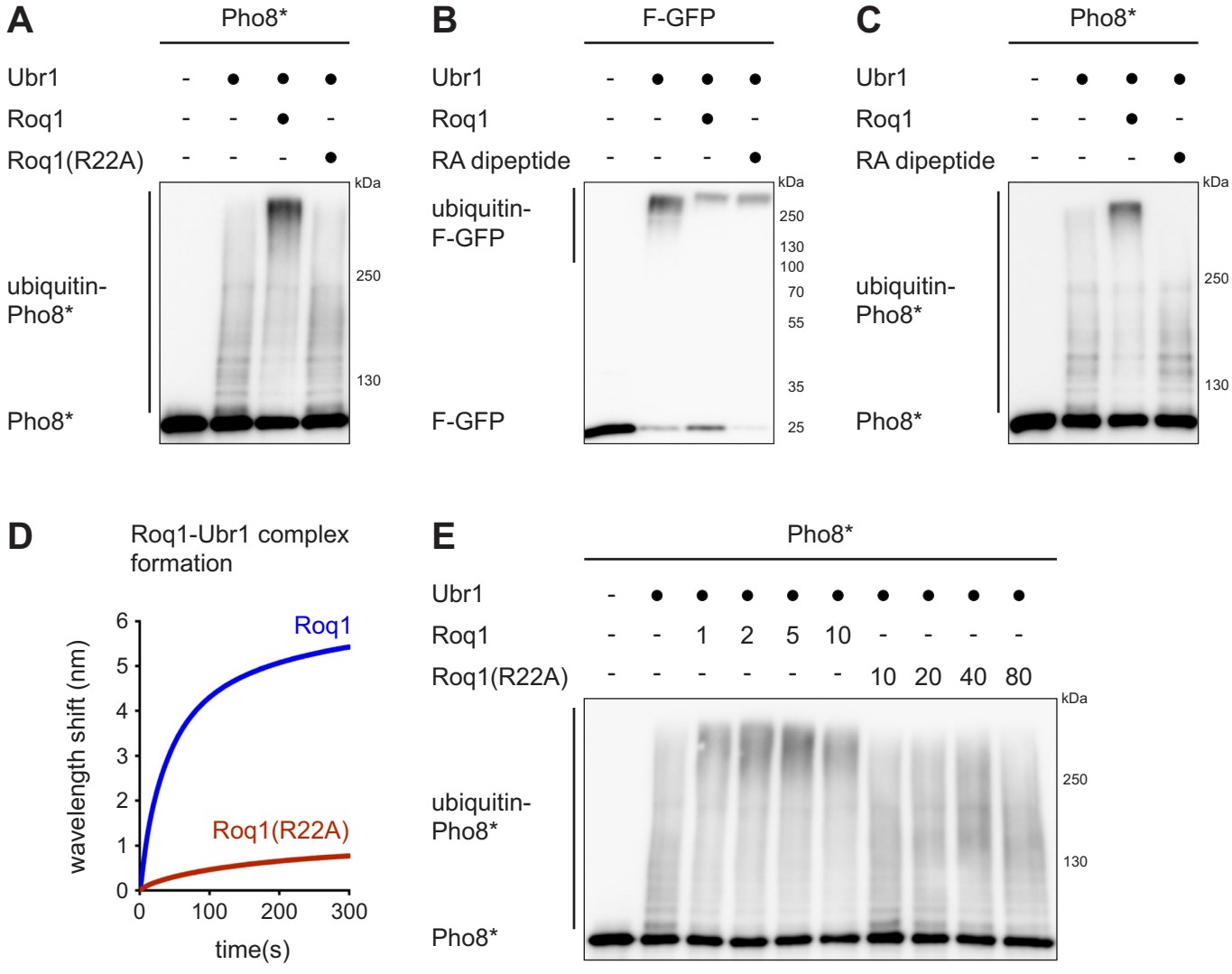

**Figure 2. Roq1 binds and regulates Ubr1 via R22 and at least one other determinant.**

(A) Western blot of Pho8 from Pho8* ubiquitination assays with Roq1(22-60) or Roq1(22-60)(R22A). Roq1:Ubr1 molar ratios were 10:1. $t = 90$ min. (B) Western blot of GFP from F-GFP ubiquitination assays without and with Roq1(22-60) or the RA dipeptide. Molar ratios were Roq1:Ubr1 = 10:1 and RA:Ubr1 = 4000:1. $t = 90$ min. (C) As in panel B but western blot of Pho8 from Pho8* ubiquitination assays. (D) Biolayer interferometry of Roq1-Ubr1 complex formation with immobilized Roq1(22-104) and soluble Ubr1. Data are the mean of three independent experiments. (E) Western blot of Pho8 from Pho8* ubiquitination assays with Roq1(22-60) or Roq1(22-60)(R22A). Numbers in the Roq1 rows indicate the Roq1:Ubr1 molar ratios used. $t = 90$ min. Source data are available online for this figure.

ubiquitination of the type-1 N-degron model substrate R-GFP (Fig. 1F). Occupancy of the Ubr1 type-1 site facilitates the turnover of type-2 N-degron substrates with bulky hydrophobic N-terminal residues (Baker and Varshavsky, 1991). Accordingly, cleaved Roq1 promotes the in vivo turnover of N-degron substrates that bind to the Ubr1 type-2 site (Szoradi et al, 2018). The same effect was apparent in vitro, where Roq1(22-60) enhanced Ubr1-mediated ubiquitination of the type-2 N-degron substrate F-GFP (Fig. 1G). These observations corroborate that Roq1 inhibits the ubiquitination of type-1 N-degron substrates but promotes the ubiquitination of type-2 N-degron substrates.

Collectively, the in vitro reconstitution experiments demonstrated that Roq1 alters the substrate specificity of Ubr1: it

promotes the ubiquitination of misfolded proteins, folded proteins with internal degrons and type-2 N-degron substrates, but inhibits the ubiquitination of type-1 N-degron substrates. Thus, the in vitro system contains all components minimally required for the reprograming of Ubr1 by Roq1.

## Roq1 contains a functionally essential hydrophobic motif

Next, we sought to determine how Roq1 reprograms Ubr1. We found previously that the N-terminal R22 of cleaved Roq1 is required for binding to the Ubr1 type-1 site and activation of SHRED in vivo (Figs. 1A and EV3; Szoradi et al, 2018). Accordingly, an R22A mutant variant of Roq1(22-60) failed to

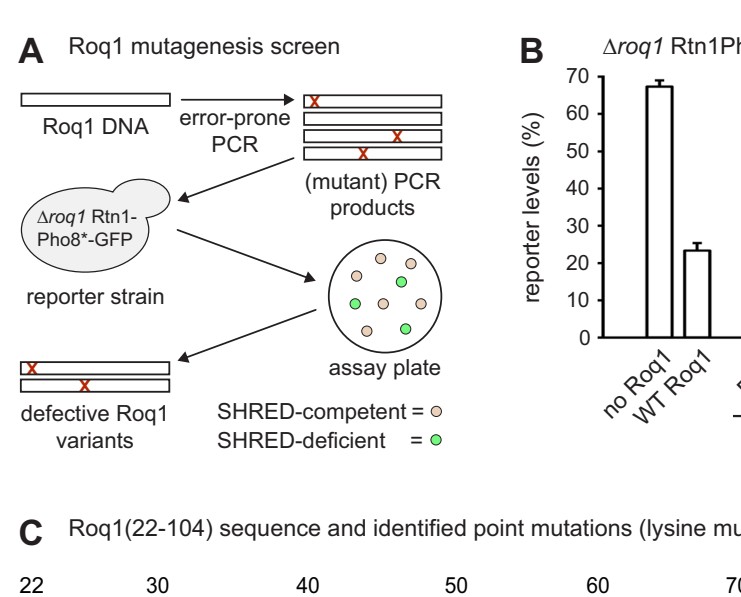

**A** Roq1 mutagenesis screen

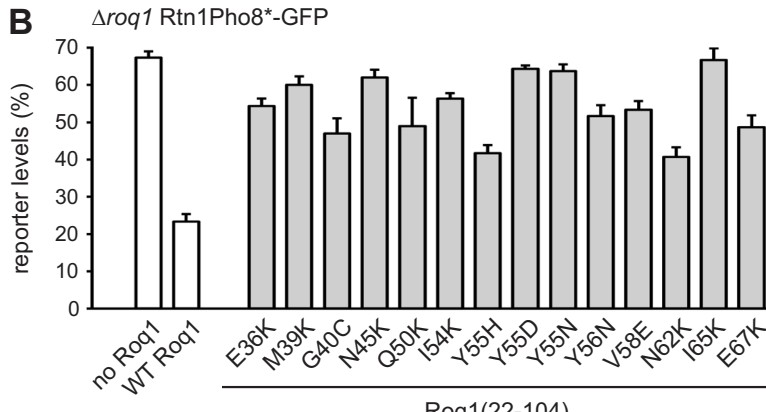

**B** Δ*roq1* Rtn1Pho8*-GFP

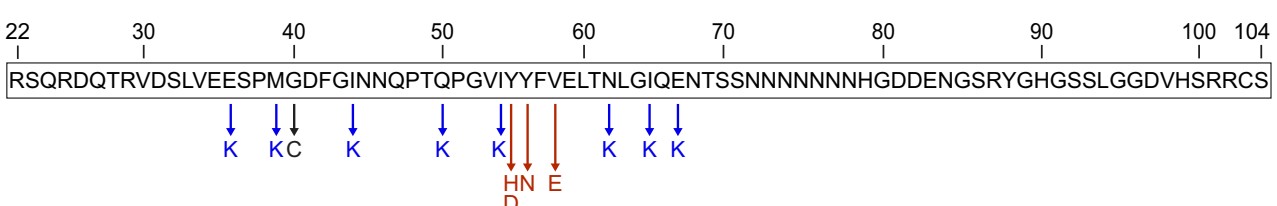

**C** Roq1(22-104) sequence and identified point mutations (lysine mutations in blue, polar mutations in red)

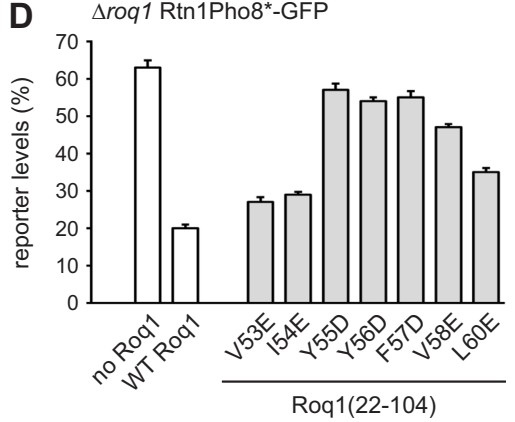

**D** Δ*roq1* Rtn1Pho8*-GFP

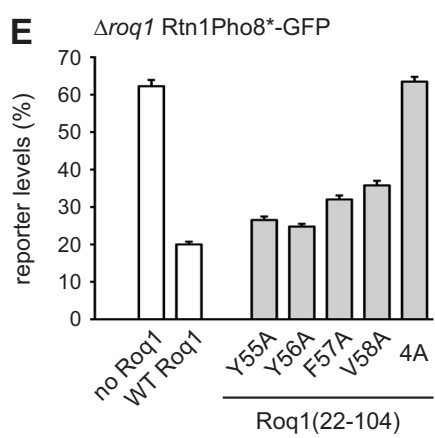

**E** Δ*roq1* Rtn1Pho8*-GFP

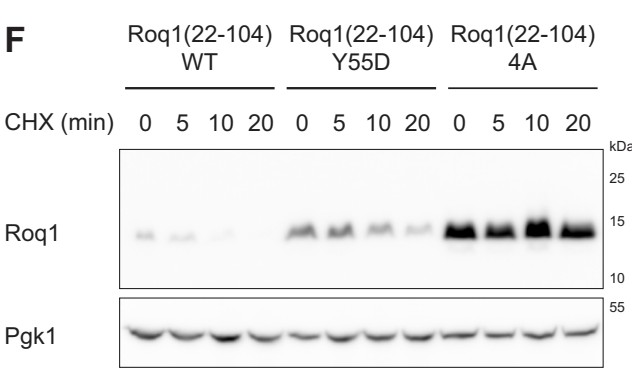

**F**

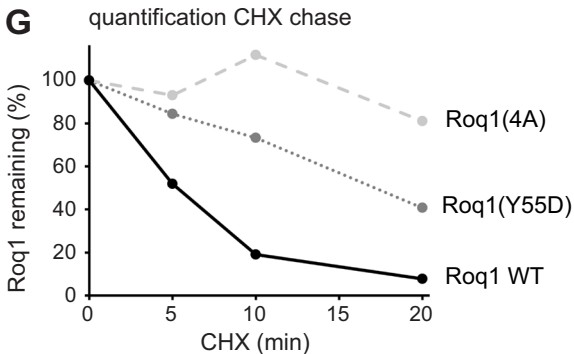

**G** quantification CHX chase

**Figure 3. Roq1 contains a functionally essential hydrophobic motif.**

(A) Workflow of the Roq1 mutagenesis screen. Roq1 variants mutagenized by error-prone PCR were introduced into a reporter strain lacking endogenous Roq1. Cells were allowed to form colonies on assay plates, SHRED-deficient colonies were picked and their defective Roq1 variants were sequenced. (B) Cellular levels of the SHRED reporter Rtn1Pho8*-GFP after tunicamycin treatment for 5 h relative to levels in untreated cells, as measured by flow cytometry. The *roq1* mutant cells contained an empty plasmid (no Roq1), a plasmid encoding wild-type Roq1 (WT Roq1) or plasmids encoding variants of ubiquitin-Roq1(22-104) fusions. Ubiquitin-Roq1 fusions are processed by cells to yield Roq1(22-104). WT, wild-type. Bars are the mean ± s.e.m.; $n = 3$ biological replicates. (C) Schematic of the Roq1(22-104) sequence showing the point mutations identified in the Roq1 mutagenesis screen. (D, E) As in (B). 4A = Y55A,Y56A,F57A,V58A. (F) Western blot of HA tag from cells expressing Ub-Roq1(22-104)-HA(74) or mutant versions containing a Y55D or a 4 A mutation. Cells were treated with cycloheximide to block protein synthesis for the time indicated. CHX cycloheximide. (G) Quantification of western blots as shown in (F). For each Roq1 variant, data are normalized to $t = 0$ min. Data are the mean of three technical replicates. Source data are available online for this figure.

stimulate Pho8* ubiquitination by Ubr1 in vitro (Fig. 2A). A simple model for Roq1 action is that binding of R22 to the Ubr1 type-1 site allosterically activates the recognition of both type-2 N-degron substrates and misfolded proteins. To test this idea, we attempted to replace Roq1 with an arginine-alanine (RA) dipeptide, which binds to the Ubr1 type-1 site (Baker and Varshavsky, 1991). The RA dipeptide enhanced ubiquitination of the type-2 N-degron substrate F-GFP, as expected (Fig. 2B). However, it did not stimulate the ubiquitination of Pho8*, even at a 4000-fold molar excess over Ubr1 (Fig. 2C). Hence, occupancy of the Ubr1 type-1 site alone does not promote the ubiquitination of misfolded proteins.

The data above imply that Roq1 harbors at least one determinant beyond R22 that is relevant for Ubr1 binding, Ubr1 regulation, or both. To study Roq1-Ubr1 interaction, we employed biolayer interferometry. We immobilized biotinylated Roq1(22-104) on a biosensor, added soluble Ubr1, and measured binding. Kinetic assays showed a robust association of Ubr1 with Roq1. The association was diminished but not abolished by the R22A mutation, further confirming the presence of a second Ubr1 binding site in Roq1 (Fig. 2D). Last, we asked whether Roq1 lacking R22 retained any capacity to activate Ubr1. As shown above, Roq1(R22A) did not affect Pho8* ubiquitination when used at a tenfold molar excess over Ubr1 (Fig. 2A). However, further raising its concentration slightly enhanced Pho8* ubiquitination (Fig. 2E, for example, note the upward shift of ubiquitin-Pho8* in the lanes with a 40- or 80-fold molar excess of Roq1(R22A) compared with the lane with a tenfold molar excess). Therefore, Roq1(R22A) still has a limited ability to activate Ubr1. The failure of Roq1(R22A) to activate SHRED even when overexpressed in yeast (Fig. EV3) may reflect that it was not possible to reach sufficiently high expression levels. Overall, these results show that cleaved Roq1 contains at minimum one determinant besides R22 that is important for both Ubr1 binding and regulation.

Bioinformatic predictions suggested that Roq1 is almost entirely disordered (Fig. EV4A,B; Jumper et al, 2021; Kurgan, 2022). In addition, sequence homologs of Roq1 can be recognized only in closely related fungi. Among fungal Roq1 homologs, the sequence around the Ynm3 cleavage site and a hydrophobic region in the middle of the protein sequence show the strongest conservation (Fig. EV4C). To identify functionally important features in Roq1 in an unbiased manner and potentially obtain informative Roq1 mutant variants, we carried out a genetic screen. As illustrated in Fig. 3A, we mutagenized full-length Roq1 by error-prone PCR, introduced the resulting PCR products into *roq1* knockout yeast harboring the SHRED reporter Rtn1Pho8*-GFP and seeded them onto solid medium. SHRED-competent cells degrade Rtn1Pho8*-

GFP when they form mature colonies and lose their fluorescence, whereas SHRED-deficient cells remain fluorescent (Szoradi et al, 2018). To identify inactive Roq1 variants, we visually screened for colonies that retained fluorescence and sequenced their Roq1. We found fourteen unique point mutations, all of which impaired stress-induced Rtn1Pho8*-GFP degradation as measured by flow cytometry (Fig. 3B). In these experiments, Roq1 variants were expressed under the strong *GPD* promoter, and all mutant variants were detectable by western blotting (Fig. EV4D). In contrast, wild-type Roq1 expressed under the *ROQ1* promoter was not abundant enough for detection (Szoradi et al, 2018; also see Fig. 4A). Thus, the mutant variants were present at levels exceeding those of endogenous Roq1 and should have been abundant enough to activate SHRED. We therefore conclude that the mutations do not simply destabilize Roq1 but genuinely disrupt its function.

The mutations were exclusively in the Roq1(22-104) fragment and mostly fell into two classes that we termed lysine mutations and polar mutations (Fig. 3C). Lysine mutations introduced lysine residues into the otherwise lysine-free Roq1(22-104) and were scattered across the Roq1 sequence. These mutations may inactivate Roq1 because they allow Ubr1 to ubiquitinate Roq1 and thereby prevent Roq1-Ubr1 interaction. Consistent with this interpretation, changing residue I54 to an arginine instead of a lysine yielded active Roq1 (Fig. EV4E). This observation suggests that, at least for this residue, the lysine mutation disrupts Roq1 function by allowing Roq1 ubiquitination, rather than by introducing a positive charge. Furthermore, individually changing all residues affected by lysine mutations to alanines did not perturb the SHRED activity of Roq1 (Fig. EV4F). Hence, these residues are not critical for Roq1 function and we did not analyze the lysine mutations further. The polar mutations introduced polar or charged residues in place of hydrophobic residues clustered at positions 55–58. This region is the least likely part of Roq1 to be disordered and also the most hydrophobic (Figs. EV4B and 3C). We therefore individually mutated each hydrophobic residue from V53 to L60 to negatively charged residues. This analysis defined the Y55-Y56-F57-V58 sequence as critical for Roq1 function (Fig. 3D). Individual mutation of these residues to alanines did not impair Ubr1 regulation by Roq1, but simultaneous mutation of all four residues to alanines abolished it (Fig. 3E). Hence, the YYFV sequence, which we call the hydrophobic motif, is a second feature of Roq1 that is essential for SHRED.

Of note, mutations in the hydrophobic motif increased the levels of Roq1(22-104), indicating that the hydrophobic motif destabilizes Roq1 in vivo and may serve as a degron (Fig. EV3B). This notion was confirmed by a cycloheximide chase experiment, which demonstrated that a Y55D mutation within the hydrophobic motif increased the steady-state levels of Roq1 and slowed its degradation, and that changing all four residues of the hydrophobic motif

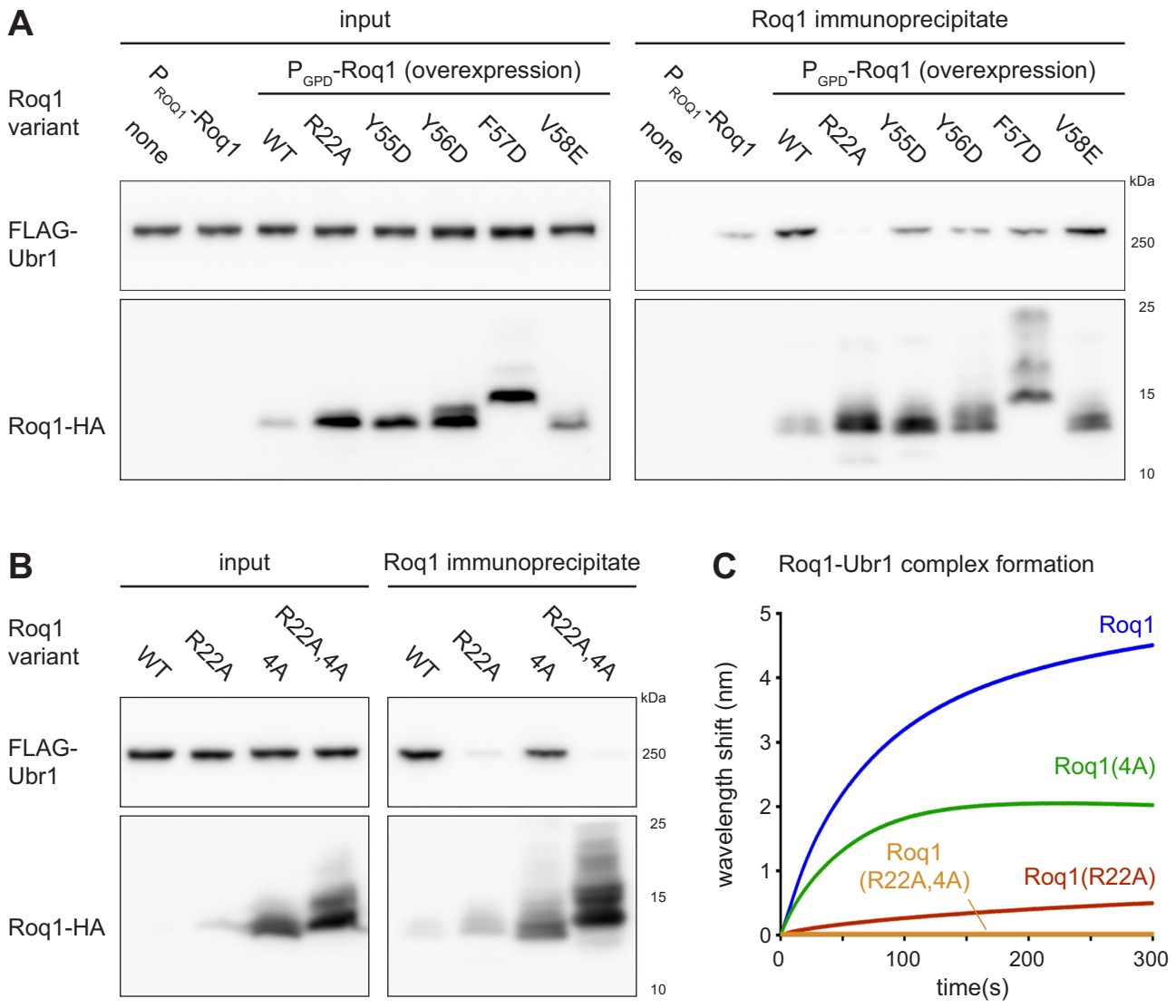

**Figure 4. Roq1 interacts with Ubr1 through a heterobivalent binding mechanism.**

(A) Western blot of FLAG and HA tags from cell lysates (input) or anti-HA immunoprecipitates of *roq1* knockout cells expressing FLAG-Ubr1 and HA-tagged Roq1 variants as indicated (Roq1 immunoprecipitate). All Roq1 variants were expressed as ubiquitin-Roq1(22-104)-HA(74) fusions, which are processed by cells to yield Roq1(22-104)-HA(74) with an internal HA tag. Note that Roq1(22-104)-HA(74) expressed under the endogenous *ROQ1* promoter is undetectable due to low expression levels. Roq1(22-104) naturally runs as a double band. The nature of the slower migrating band is unknown, as is the reason for the strong impact of the F57D mutation on Roq1 electrophoretic mobility. $P_{ROQ1}$ = *ROQ1* promoter, $P_{GPD}$ = *GPD* promoter, WT = wild-type. (B) As in (A). All Roq1 variants were expressed under the strong *GPD* promoter. (C) Biolayer interferometry of Roq1-Ubr1 complex formation with immobilized Roq1(22-104) and soluble Ubr1. Data are the mean of two independent experiments. 4A = Y55A,Y56A,F57A,V58A. Source data are available online for this figure.

to alanines stabilized Roq1 even more strongly (Fig. 3F,G). Roq1 is short-lived but Ubr1 has only a minor role in the turnover of wild-type Roq1 (Szoradi et al, 2018). Hence, Roq1 degradation must involve other ubiquitin ligases or occur in a ubiquitin-independent manner. In any event, it appears that the hydrophobic motif is both a degron and a critical determinant for Ubr1 regulation.

## Roq1 interacts with Ubr1 through a heterobivalent binding mechanism

Next, we asked whether the hydrophobic motif was required for the interaction of Roq1 and Ubr1. We expressed HA-tagged wild-type

and mutant variants of Roq1(22-104) in yeast harboring FLAG-Ubr1, immunoprecipitated Roq1 and probed for co-precipitation of Ubr1. In agreement with earlier findings (Szoradi et al, 2018), wild-type Roq1 efficiently precipitated Ubr1, and this interaction was strongly diminished by the R22A mutation (Fig. 4A). Point mutations that introduced negative charges into the hydrophobic motif reduced the affinity of Roq1 to Ubr1 to varying degrees. Mutation of all four residues of the hydrophobic motif to alanines reduced binding to Ubr1, and combined mutation of R22 and the hydrophobic motif abolished it (Fig. 4B).

An interpretation of these in vivo experiments is that Roq1 interacts with Ubr1 through binding of R22 to the Ubr1 type-1 site

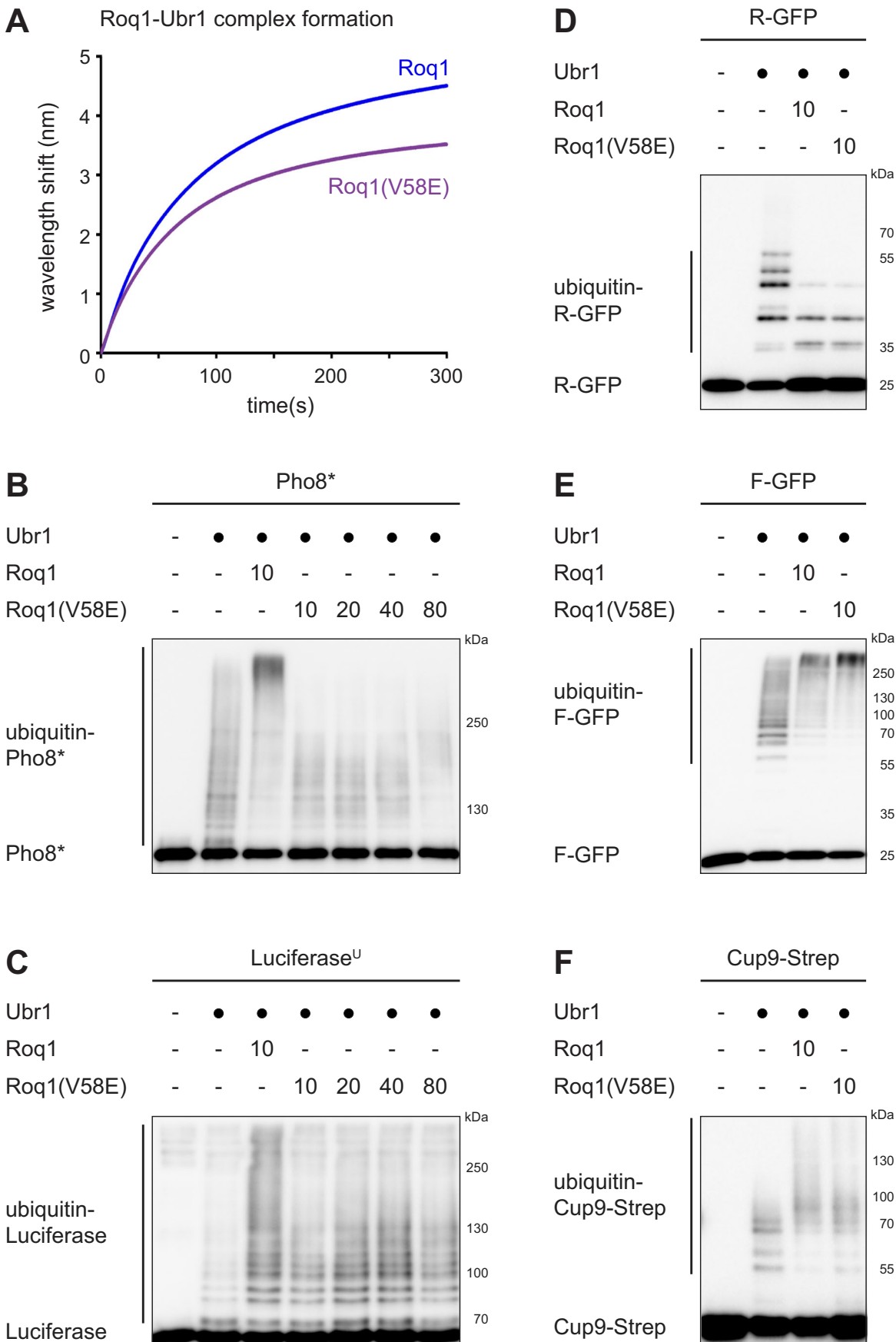

**Figure 5. The Roq1 hydrophobic motif selectively promotes the ubiquitination of misfolded proteins.**

(A) Biolayer interferometry of Roq1-Ubr1 complex formation with immobilized Roq1(22-104) and soluble Ubr1. Data are the mean of two independent experiments. The data for wild-type Roq1 are the same as in Fig. 4C. (B) Western blot of Pho8 from Pho8* ubiquitination assays with Roq1(22-60) or Roq1(22-60)(V58E). Numbers in the Roq1 rows indicate the Roq1:Ubr1 molar ratios used. $t = 90$ min. (C) As in (B) but western blot of Luciferase from Luciferase$^U$ assays. $t = 30$ min. (D) As in (B) but western blot of GFP from R-GFP assays. $t = 15$ min. (E) As in panel B but western blot of GFP from F-GFP assays. $t = 15$ min. (F) As in (B) but western blot of Strep tag from Cup9-Strep assays. $t = 15$ min. Source data are available online for this figure.

and binding of the hydrophobic motif to another region of Ubr1. To test this model in vitro, we performed biolayer interferometry with Roq1 variants and Ubr1. Individual mutation of R22 or the hydrophobic motif impaired Roq1-Ubr1 binding, and the combined mutation of both determinants rendered the interaction undetectable (Fig. 4C). These results provide further evidence for synergy between two distinct interfaces between Roq1 and Ubr1 and thus a heterobivalent binding mechanism.

## The Roq1 hydrophobic motif selectively promotes the ubiquitination of misfolded proteins

We next wanted to determine whether the hydrophobic motif merely served as a binding determinant or, like R22, additionally controlled Ubr1 activity. For this purpose, we took advantage of the V58E mutation identified in the genetic screen. The V58E mutation crippled the ability of Roq1 to drive degradation of the SHRED reporter Rtn1Pho8*-GFP in yeast (Fig. 3D). Nonetheless, Roq1(V58E) still co-immunoprecipitated Ubr1 from cell lysates nearly as efficiently as wild-type Roq1 (Fig. 4A). Biolayer interferometry confirmed that Roq1(V58E) was still able to associate with Ubr1, albeit less rapidly than wild-type Roq1 (Fig. 5A). These observations raised the possibility that the hydrophobic motif has separable binding and regulatory roles. Specifically, the hydrophobic motif could (1) stabilize the Roq1-Ubr1 interaction in a manner that involves but does not strictly depend on V58 and (2) activate Ubr1 in a manner that requires V58. To explore this idea, we compared the impact of wild-type Roq1 and Roq1(V58E) on the ubiquitination of different Ubr1 substrates. Roq1(V58E) could not stimulate Ubr1-mediated ubiquitination of Pho8* and only mildly enhanced ubiquitination of unfolded Luciferase$^U$, even when the concentration of Roq1(V58E) was raised to compensate for its reduced binding to Ubr1 (Fig. 5B,C). In striking contrast, wild-type Roq1 and Roq1(V58E) were equally effective at inhibiting ubiquitination of the type-1 N-degron substrate R-GFP, at stimulating ubiquitination of the type-2 N-degron substrate F-GFP and at stimulating ubiquitination of the folded Ubr1 substrate Cup9 (Fig. 5D–F). We conclude that the key event in the control of R-GFP, F-GFP and Cup9 ubiquitination is binding of the Roq1 N-terminus to the Ubr1 type-1 site. For misfolded Pho8* and unfolded Luciferase$^U$, however, the hydrophobic motif provides an additional regulatory activity that depends on V58.

The Roq1 hydrophobic motif could regulate Ubr1 in several ways. It could change the conformation of Ubr1 to enhance the recognition of misfolded proteins, stimulate the recruitment of the ubiquitin-conjugating enzyme Rad6 or alter the structure of the Rad6-ubiquitin-Ubr1 substrate complex so that substrate ubiquitination becomes more efficient. To test these possibilities, we first used an in vitro pulldown assay to determine whether Roq1 affected the recognition of Pho8 or Pho8* by Ubr1. Pho8* bound to Ubr1

more strongly than Pho8, but Roq1 did not enhance this interaction (Fig. EV5A). Second, we analyzed the interaction of Rad6 and Ubr1 by biolayer interferometry. Rad6 can bind to Ubr1 even when it is not conjugated to ubiquitin (Xie and Varshavsky, 1999), and the association of unloaded Rad6 and Ubr1 was indeed detectable. However, Roq1 did not enhance it (Fig. EV5B). Third, we asked how the hydrophobic motif stimulates substrate ubiquitination. Standard ubiquitination assays cannot reveal whether Roq1 promotes chain initiation, i.e., the attachment of the first ubiquitin to lysines in substrate proteins, makes chain elongation more efficient, or has a combination of effects. To focus on chain initiation, we employed lysine-free ubiquitin (no-K-ubiquitin), which can be attached to substrate proteins via its C-terminus but cannot form ubiquitin chains. When we used no-K-ubiquitin together with Pho8*, Ubr1 on its own was able to generate two discrete bands, which correspond to Pho8* conjugated with one or two molecules of no-K-ubiquitin (Fig. 6A). Hence, without Roq1, two different lysines of Pho8* can be efficiently ubiquitinated. Wild-type Roq1 increased the number of ubiquitinated lysines to at least five. Roq1(V58E) and the RA dipeptide had no effect, consistent with the absence of the regulatory activity provided by the hydrophobic motif. For F-GFP, Ubr1 on its own could attach ubiquitin to five distinct lysines (Fig. 6B). The RA dipeptide raised the number of ubiquitinated lysines to six and increased the efficiency of ubiquitination. As expected, Roq1 and Roq1(V58E) had no effects beyond that of the RA dipeptide. The same was true for Cup9 (Fig. 6C).

In summary, the Roq1 hydrophobic motif appears to have two functions. First, it stabilizes the association of Roq1 and Ubr1, which reinforces binding of R22 to the Ubr1 type-1 site. In this way, the hydrophobic motif helps R22 to regulate the ubiquitination of type-1 and type-2 N-degron substrates and folded substrates with internal degrons, such as Cup9. Second, the hydrophobic motif promotes the ubiquitination of misfolded proteins, at least in part by enhancing chain initiation. This second function specifically requires V58.

## Role of the Roq1 hydrophobic motif in protein degradation in vivo

To test the in vivo relevance of our findings, we introduced the R22A, V58E and 4A mutations into the endogenous *ROQ1* gene locus in yeast. As expected, the mutations disrupted stress-induced degradation of the SHRED reporter Rtn1Pho8*-GFP (Fig. 7A). To analyze the turnover of N-degron substrates, we used R- and F-mCherry-sfGFP, which are fusion proteins of the slowly-maturing red fluorescent mCherry and the fast-maturing green fluorescent sfGFP. Their mCherry/sfGFP fluorescence ratio reports on the average age of the fusion proteins and thus their stability (Khmelinskii et al, 2012). As before (Szoradi et al, 2018), we placed the Roq1 variants under the control of an inducible

# A

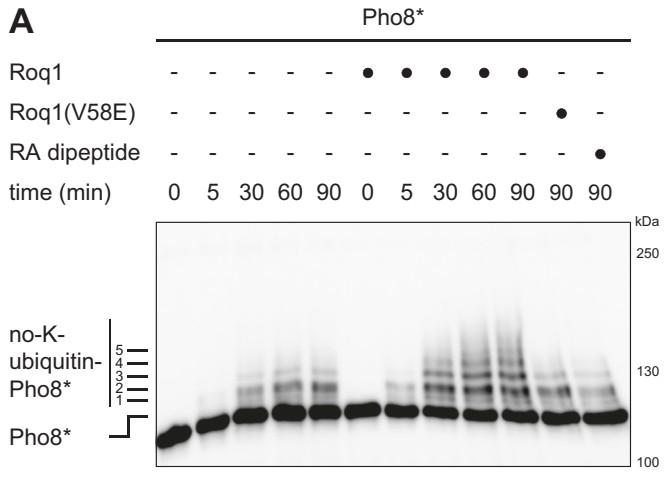

**Figure 6. Roq1 promotes ubiquitin chain initiation on Ubr1 substrate proteins.**

(A) Western blot of Pho8 from Pho8* ubiquitination assays with lysine-free ubiquitin (no-K-ubiquitin) and Roq1(22-60), Roq1(22-60)(V58E), or RA dipeptide. Numbers to the left of the gel indicate distinct conjugates of no-K-ubiquitin and Pho8*. (B) As in (A) but western blot of GFP from F-GFP assays. (C) As in (A) but western blot of Strep tag from Cup9-Strep assays. Source data are available online for this figure.

# B

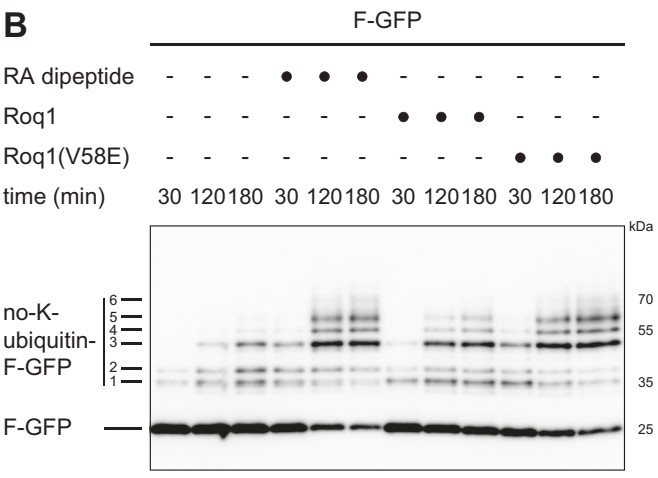

# C

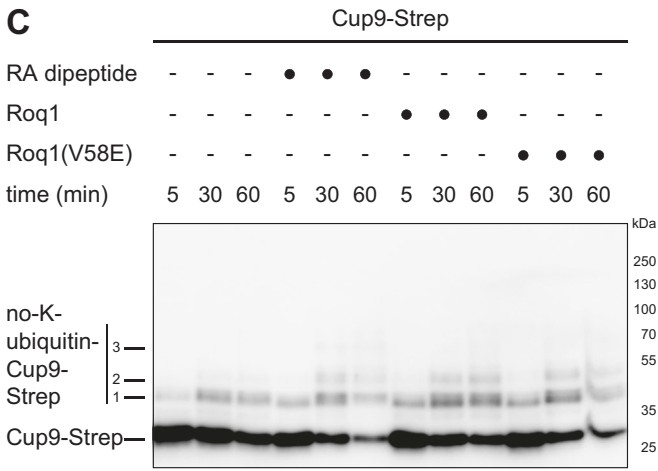

of the type-1 site actives substrate recognition by the type-2 site (Fig. 7C). Roq1(R22A) and Roq1(4A) stabilized R-mCherry-sfGFP only marginally, whereas Roq1(V58E) showed a stronger stabilizing effect (Fig. 7B). This stabilization of the type-1 N-degron substrate was not as pronounced as with wild-type Roq1, presumably because Roq1(V58E) binds to Ubr1 less strongly and therefore cannot occlude the Ubr1 type-1 site as effectively. Similarly, Roq1(R22A) and Roq1(4A) had little destabilizing effect on F-mCherry-sfGFP whereas Roq1(V58E) destabilized it like wild-type Roq1 (Fig. 7C). Hence, Roq1(R22A) and Roq1(4A) did not strongly impact the degradation of the misfolded SHRED reporter or the N-degron substrates. Roq1(V58E), however, was inactive towards the SHRED reporter but retained significant activity towards N-degron substrates. These results show that the V58E mutation partially uncouples the functions of the Roq1 hydrophobic motif in Ubr1 binding and the regulation of SHRED, in agreement with the in vitro findings.

## Functional Roq1 minimally consists of R22, a short linker, and the hydrophobic motif

To better understand the functional architecture of Roq1, we aimed to reduce it to its minimal parts. The data so far showed that Roq1 functionality in vitro and in vivo requires R22 and the hydrophobic motif. We first asked whether the two determinants needed to be present in the same polypeptide or could act as parts of distinct molecules. For this purpose, we combined Roq1(R22A), which possesses the hydrophobic motif but cannot bind the Ubr1 type-1 site, with Roq1(4A) or a RA dipeptide, which lack the hydrophobic motif but can occupy the Ubr1 type-1 site. However, none of these combinations was able to stimulate the in vitro ubiquitination of Pho8* by Ubr1 (Fig. EV5C,D). These results argue that R22 and the hydrophobic motif need to be part of the same polypeptide to be effective.

Next, we asked whether Roq1 regions besides R22 and the hydrophobic motif were functionally important. As shown above, the sequence beyond residue 60 is dispensable for Ubr1 regulation in vitro and SHRED activation in vivo (Fig. EV1B,H). We therefore turned to the sequence connecting R22 and the hydrophobic motif and tested a number of constructs in which this sequence was replaced with generic linkers, shortened, or both (Fig. 8A). Replacement of the 27 weakly conserved residues between S23 and P51 in Roq1(22-104) with a GSP-rich linker of identical length yielded Roq1 that was active in vivo (Fig. 8B; see Fig. EV4C for conservation and Fig. EV5E for expression levels). Hence, no functionally essential features are contained within the replaced sequence.

We then shortened the sequence that links R22 and the hydrophobic motif, which naturally consists of 32 amino acid residues (Fig. 8A). Linker shortening progressively destabilized

promoter system and compared the stability of N-degron substrates in the absence and presence of Roq1. Wild-type Roq1 stabilized R-mCherry-sfGFP because it competes with the type-1 N-degron substrate for binding to the Ubr1 type-1 site (Fig. 7B). By contrast, wild-type Roq1 destabilized F-mCherry-sfGFP because occupancy

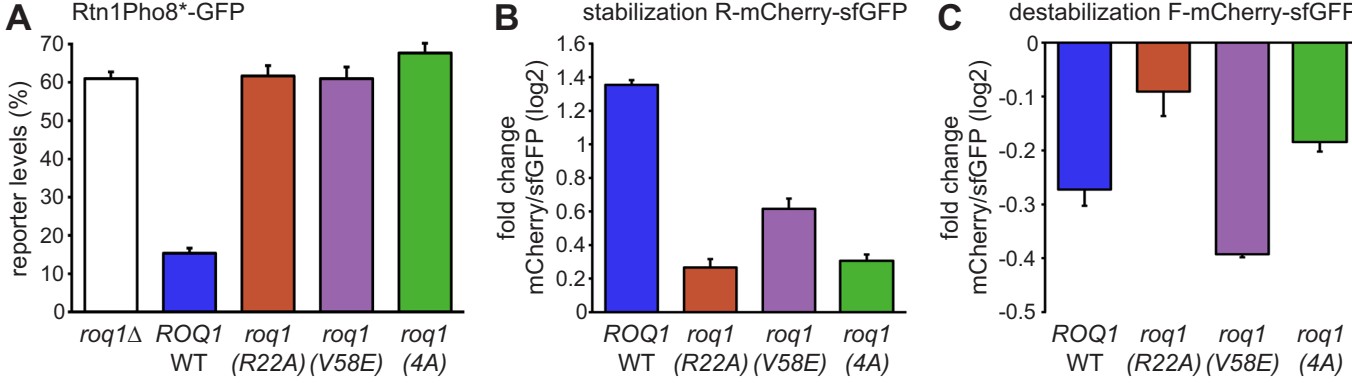

**Figure 7. Delineation of distinct activities of the Roq1 hydrophobic motif in vivo.**

(A) Cellular levels of the SHRED reporter Rtn1Pho8*-GFP after tunicamycin treatment for 5 h relative to the levels in untreated cells, as measured by flow cytometry. The indicated point mutations were introduced into the endogenous *ROQ1* gene. 4A = Y55A,Y56A,F57A,V58A, WT wild-type. Bars are the mean ± s.e.m.; $n = 3$ biological replicates. (B) Flow cytometry of mCherry/sfGFP fluorescence of untreated and estradiol-treated cells expressing the ubiquitin-R-mCherry-sfGFP type-1 N-degron substrate and Roq1 variants under an estradiol-inducible promoter system. Plotted on a log2 scale is the fold change of mCherry/sfGFP fluorescence upon Roq1 expression by estradiol treatment for 6 h. An increased mCherry/sfGFP ratio indicates stabilization of ubiquitin-R-mCherry-sfGFP upon Roq1 expression. Bars are the mean ± s.e.m.; $n = 3$ biological replicates. (C) As in panel B but with the ubiquitin-F-mCherry-sfGFP type-2 N-degron substrate. A decreased mCherry/sfGFP ratio indicates destabilization of Ubiquitin-F-mCherry-sfGFP upon Roq1 expression. Bars are the mean ± s.e.m.; $n = 3$ biological replicates. Source data are available online for this figure.

Roq1 (Fig. EV5F). Nevertheless, the Roq1 variants remained functional in vivo until Roq1 had been reduced to 63 residues in Roq1(63aa). This variant was only partially active, either because the linker was now too short or because the abundance of the protein had dropped below a critical threshold (Figs. 8B and EV5F). The shortest fully functional variant, Roq1(66aa), contained only 15 residues between R22 and the hydrophobic motif. Replacing most of these residues with GS- or GSP-based linkers partially inactivated Roq1 (Fig. 8B). However, these variants showed very low expression levels (Fig. EV5G), so that it was not possible to infer to what extent they could still regulate Ubr1. To clarify this point, we tested Roq1(22aa), Roq1(22aa_GS) and Roq1(22aa_GSP) in vitro. These variants lacked the sequence beyond the hydrophobic motif and were only 22 residues long (Fig. 8C). All three were active in vitro (Fig. 8D). These experiments define minimalistic Roq1-derived peptides that are able to regulate Ubr1. Remarkably, peptides of only 22 residues that are composed of an N-terminal arginine residue, a flexible linker and a short hydrophobic motif is all that is needed to profoundly change the activity and substrate specificity of a large ubiquitin ligase that consists of nearly 2000 residues.

## Discussion

Our findings reveal that Roq1 reprograms Ubr1 by simultaneously engaging two channels of communication. Roq1 uses R22 to occupy the Ubr1 type-1 site as a pseudosubstrate. This interaction activates the Ubr1 type-2 site and the binding site for folded substrates with internal degrons. In parallel, Roq1 uses the hydrophobic motif to bind to Ubr1 at a separate regulatory site. This interaction promotes the ubiquitination of misfolded proteins, presumably through a fourth substrate binding site. As a result, Roq1 alters Ubr1 substrate specificity in a coordinated manner (Fig. 8E).

Roq1 induces complex changes in Ubr1, but its architecture is simple. Its two key features, the N-terminal R22 and the YYFV sequence, are so-called short linear motifs (SLiMs). Generally, SLiMs provide monopartite interfaces for protein-protein interactions, even though they are too short to form stable three-dimensional structures. Through their interactions, they act as functional modules and serve as ligands, proteolytic cleavage sites, targeting signals, degrons and sites of posttranslational modification (Davey et al, 2012; Van Roey et al, 2014; Kumar et al, 2024). The two SLiMs of Roq1 are embedded in a completely disordered protein, which makes them maximally accessible for Ubr1. They cooperate in interacting with Ubr1 as they contact distinct sites in Ubr1. The resulting heterobivalent binding mechanism creates avidity and a strong Roq1-Ubr1 interaction, even though each SLiM individually is too small to generate high affinity. Such cooperative binding is found in many intrinsically disordered proteins (Holehouse and Kragelund, 2024). In the case of Roq1, cooperative binding is essential not only for a strong overall association with Ubr1 but also to ensure that R22 outcompetes N-degron substrates at the type-1 site and inhibits their degradation.

Remarkably, the Roq1 SLiMs are multifunctional so that diverse biological activities are built into Roq1 by means of only a few amino acid residues. Full-length Roq1 is cleaved by the protease Ynm3 in a relatively well-conserved IL'RSQR sequence (with ' indicating the cleavage site; Fig. EV4C). Hence, this sequence is a SLiM that initially serves as a proteolytic cleavage site and, after processing, as a ligand for the Ubr1 type-1 site. The hydrophobic motif has the YYFV sequence at its core, although flanking residues may contribute to Ubr1 binding. Reducing the hydrophobicity of the motif stabilized Roq1, indicating that the motif acts as both a Ubr1 ligand and a degron. Roq1 has a half-life of only about five minutes and its fast turnover may be important for cells to rapidly adjust Roq1 abundance to proteotoxic stress levels (Szoradi et al, 2018).

**A**  Roq1 linker variants

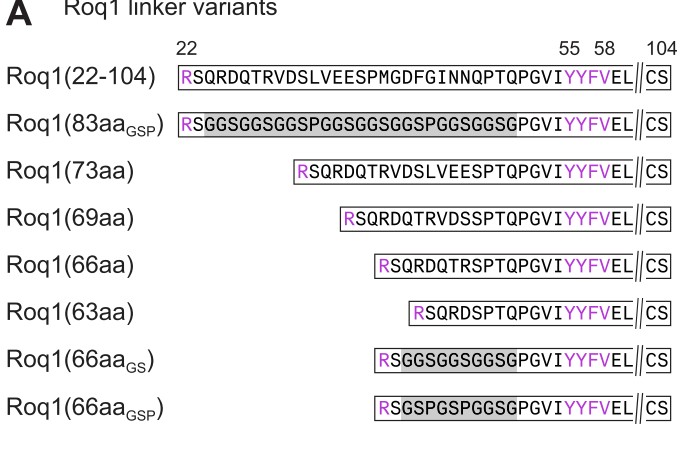

**B**  Δ*roq1* Rtn1Pho8*-GFP

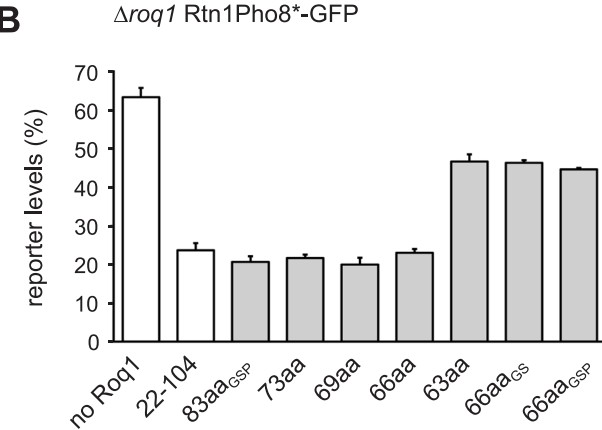

**C**  Linker replacement in shortened Roq1

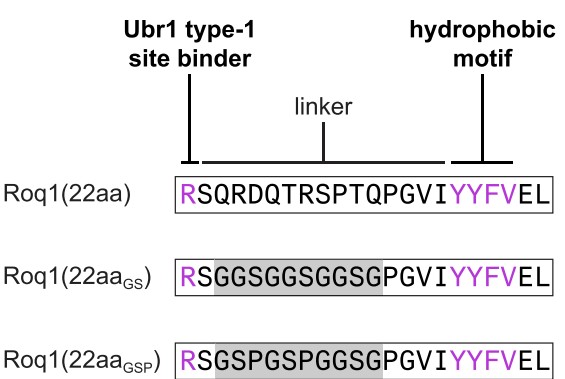

**D**

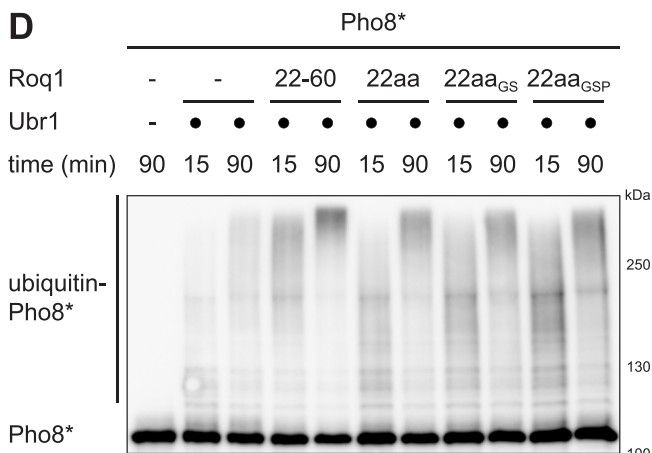

**E**  Model of Ubr1 reprograming

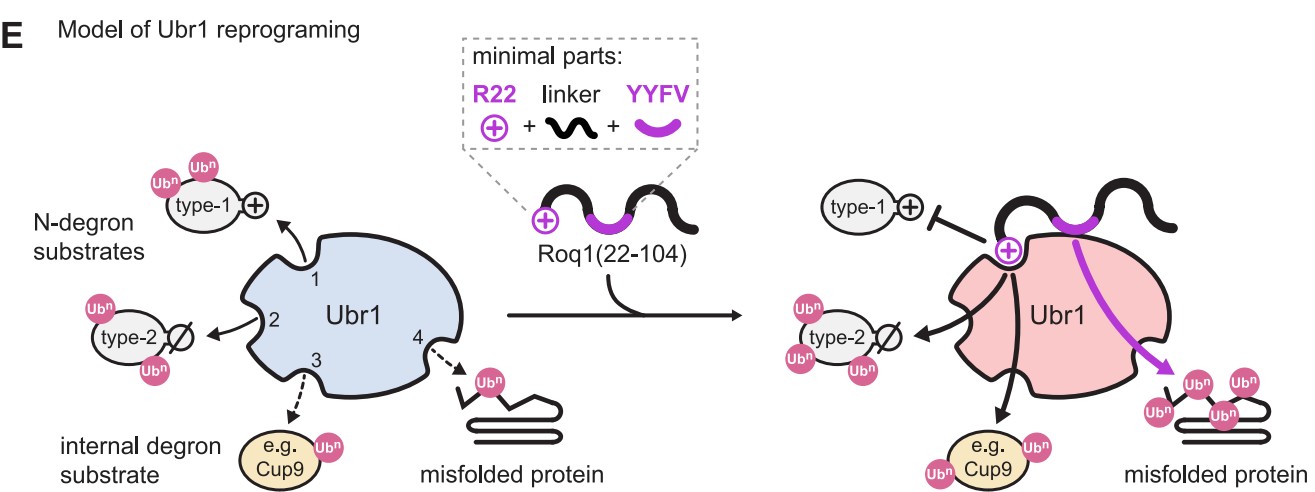

The region between R22 and the YYFV motif is 32 residues long and appears to serve primarily as a flexible linker. It can be shortened to 15 residues and largely be replaced with generic linkers without disturbing Roq1 function in vitro. Furthermore, it remains possible that even shorter linkers could still maintain function. Flexible linkers can span substantial distances. Assuming a Cα-Cα distance in the peptide backbone of 3.8 Å (Chakraborty et al, 2013), a maximally extended 32-residue linker would, theoretically, be about 12 nm long. This roughly equals the diameter of a large globular protein of 500 kDa (Erickson, 2009). Hence, the linker between the Roq1 SLiMs is suited to bridge distant binding sites on the surface of Ubr1.

**Figure 8. Functional Roq1 minimally consists of R22, a short linker, and the hydrophobic motif.**

(A) Schematic of Roq1 linker variants for the shortening and replacement of the sequence between R22 and the hydrophobic motif in Roq1(22-104). (B) Cellular levels of the SHRED reporter Rtn1Pho8*-GFP after tunicamycin treatment for 5 h relative to levels in untreated cells, as measured by flow cytometry. The *roq1* mutant cells contained an empty plasmid (no Roq1), a plasmid encoding wild-type Roq1 (WT Roq1) or plasmids encoding variants of ubiquitin-Roq1 fusions in which the linker between R22 and the hydrophobic motif was replaced, shortened, or both. Ubiquitin-Roq1 fusions are processed by cells to yield Roq1 starting with R22. Bars are the mean ± s.e.m.; $n = 3$ biological replicates. (C) Schematic of Roq1 constructs for the replacement of the linker between R22 and the hydrophobic motif in shortened Roq1(22aa). (D) Western blot of Pho8 from Pho8* ubiquitination assays with Roq1 variants with shortened or artificial linkers between R22 and the hydrophobic motif. (E) Model of Ubr1 reprograming by Roq1. Cleaved Roq1(22-104) interacts with Ubr1 through a heterobivalent binding mechanism that creates avidity. Binding involves physical contact of R22 with the Ubr1 type-1 site and of the YYFV hydrophobic motif with an undefined region in Ubr1. Occupancy of the Ubr1 type-1 site activates the type-2 substrate binding site and a binding site for folded substrates with internal degrons, such as Cup9. In parallel, the hydrophobic motif activates a putative fourth substrate binding site to enhance the degradation of misfolded proteins. Numbers in unbound Ubr1 (blue) denote the known and putative substrate binding sites for type-1 N-degron substrates (1), type-2 N-degron substrates (2), folded proteins with internal degrons (3) and misfolded proteins (4). The box above Roq1(22-104) highlights the minimally required functional elements of Roq1. Ub$^n$ = ubiquitin chain. Source data are available online for this figure.

How could the Roq1 hydrophobic motif enhance the ubiquitination of misfolded proteins? Roq1 does not increase the affinity of Ubr1 for Rad6 or misfolded Pho8*. Instead, it stimulates ubiquitin chain initiation on Pho8* and may additionally stimulate chain elongation. This effect could be explained in several ways. The hydrophobic motif could release autoinhibition of the putative fourth substrate binding site of Ubr1. This mechanism would be similar to how peptide binding to the Ubr1 type-1 and -2 sites releases autoinhibition of the binding site for proteins such as Cup9 (Du et al, 2002). Alternatively or additionally, Roq1 could help to optimally position ubiquitin-loaded Rad6 and/or misfolded substrates for efficient ubiquitin transfer, and Roq1 could increase the flexibility of Ubr1 so that more substrate lysines are accessible for ubiquitination. Resolving these issues will require structural studies of Roq1-Ubr1 substrate complexes. Such studies will also reveal how the hydrophobic motif binds Ubr1 and how Ubr1 binds misfolded proteins. We attempted to determine the structure of the Roq1-Ubr1 complex by cryo-electron microscopy but obtained only low-resolution data. Furthermore, structure prediction for the Roq1-Ubr1 complex by AlphaFold 3 (Abramson et al, 2024) did not yield a binding site for the hydrophobic motif that we could confirm experimentally (unpublished results). However, the interface between the Roq1 hydrophobic motif and Ubr1 is unlikely to involve the Ubr1 type-2 site because disabling this site through point mutations does not disrupt SHRED (Szoradi et al, 2018).

Another challenge is to translate insights from the in vitro studies into an understanding of Ubr1 function in vivo. Ubr1 ubiquitinates the lysine-free Roq1(22-104) in vitro. This modification appears dispensable because Roq1(22-60) is less extensively ubiquitinated than Roq1(22-104) but has even higher activity towards Ubr1. Nonetheless, our data show that the Rad6-Ubr1 complex can ubiquitinate serines or threonines. Whether this capacity is relevant in vivo needs to be tested. Furthermore, the outcome of Ubr1 reprograming in vivo is likely more complex than can be recapitulated easily in reconstitution experiments. We show that Roq1 activates the ubiquitination of both misfolded proteins and Cup9 in vitro. In contrast, Roq1 appears to suppress Cup9 degradation in vivo (Szoradi et al, 2018). In cells, where various Ubr1 substrates are present simultaneously, Roq1 may stimulate the ubiquitination of certain substrates more strongly than that of others. In effect, this could steer Ubr1 away from proteins that it would readily ubiquitinate if it were not engaged by preferred clients.

Our findings uncover a new mode of ubiquitin ligase regulation, which is defined by cooperating multifunctional motifs in a disordered protein. Elements of this mode have been found in other contexts. Similar to Roq1, the N-termini of human SMAC and YPEL5 block N-degron binding sites of certain ubiquitin ligases as pseudosubstrates, which has been termed degron mimicry (Mace and Day, 2023; Gottemukkala et al, 2024). Importantly, SMAC and YPEL5, through folded domains, make multiple contacts with their partner ubiquitin ligases, and the resulting avidity likely helps them to outcompete the binding of genuine N-degron substrates. The active N-terminus of SMAC is generated by proteolytic processing and SMAC thus also contains a dual-function sequence that is first a proteolytic cleavage site and then an N-degron mimic (Saita et al, 2017). Finally, the proposed action of the SARS-CoV-2 protein ORF10 on the Cullin-RING ubiquitin ligase CUL2 bears intriguing resemblance to the reprograming of Ubr1 by Roq1. ORF10 consists of 38 residues and binds to ZYG11B, a substrate adaptor of CUL2, by means of an N-terminal sequence that mimics a glycine-based N-degron. As a result, the degradation of a glycine/N-degron reporter is inhibited, but the degradation of the intraflagellar transport protein IFT46 is enhanced (Wang et al, 2022; Zhu et al, 2024). Hence, there are fascinating similarities between SMAC, YPEL5, ORF10, and Roq1. Intrinsically disordered proteins are often poorly conserved at the sequence level (Holehouse and Kragelund, 2024). In addition, the Roq1 SLiMs consist of only a handful of amino acid residues each. Therefore, it is unsurprising that no recognizable sequence homologs of Roq1 exist in humans. However, the compact elements needed for a Roq1-like regulatory mechanism may be easy to evolve. In fact, intrinsically disordered proteins often acquire SLiMs de novo (Davey et al, 2015). It therefore remains possible that proteins unrelated to Roq1 at the sequence levels but similar in overall construction regulate human Ubr1 homologs.

Ubiquitin ligases are involved in many diseases (Cruz Walma et al, 2022). Artificially controlling or re-directing their activities through inhibitors, molecular glues and proteolysis-targeting chimeras are promising strategies to block or eliminate disease-causing proteins (Cowan and Ciulli, 2022; Tsai et al, 2024; Wang et al, 2024). The regulatory principle embodied by Roq1 requires only a small number of building blocks to achieve far-reaching reprograming of ubiquitin ligases. Therefore, it may be possible to engineer novel Roq1-inspired regulators, potentially also for therapeutic application.

# Methods

## Reagents and tools table

| Reagent/resource | Reference or source | Identifier or catalog number |
| --- | --- | --- |
| **Experimental models** | | |
| *ubr1::nat-P$_{GPD}$-FLAG-UBR1 pep4Δ* | This study | SSY2908 |
| *pep4Δ::TRP1* | Schuck et al, 2014 | PWY0005 |
| *BFP-hph* | Schmidt et al, 2019 | SSY795 |
| *BFP-hph leu2::P$_{ADH}$-Rtn1Pho8*-GFP-LEU2* | Szoradi et al, 2018 | SSY831 |
| *BFP-hph roq1Δ::kan* | This study | SSY803 |
| *BFP-hph roq1Δ::kan leu2::P$_{ADH}$-Rtn1Pho8*-GFP-LEU2* | Szoradi et al, 2018 | SSY835 |
| *ADE2 leu2-3,112 trp1-1 ura3-1 his3-11,15 MAT a* | Szoradi et al, 2018 | SSY122 |
| *Roq1-ALFA::TRP1 Ubr1-ALFA::HIS3* | This study | SSY4598 |
| *BFP-hph roq1Δ::kan leu2::P$_{ADH}$-Rtn1Pho8*-GFP-LEU2 ura3::nat* | This study | SSY3543 |
| *BFP-hph roq1Δ::kan leu2::P$_{ADH}$-Rtn1Pho8*-GFP-LEU2 ura3::P$_{ROQ1}$-Roq1-nat* | This study | SSY3544 |
| *BFP-hph roq1Δ::kan leu2::P$_{ADH}$-Rtn1Pho8*-GFP-LEU2 ura3::P$_{ROQ1}$-Ub-Roq1(22-104)-nat* | This study | SSY3545 |
| *BFP-hph roq1Δ::kan leu2::P$_{ADH}$-Rtn1Pho8*-GFP-LEU2 ura3::P$_{ROQ1}$-Ub-Roq1(22-60)-nat* | This study | SSY3551 |
| *ubr1Δ::nat roq1Δ::HIS3* | Szoradi et al, 2018 | SSY792 |
| *ubr1Δ::nat roq1Δ::HIS3 leu2::P$_{ADH}$-FLAG-Ubr1-LEU2* | This study | SSY2959 |
| *BFP-hph roq1(R22A)* | This study | SSY4529 |
| *BFP-hph roq1(V58E)* | This study | SSY4549 |
| *BFP-hph roq1(4A)* | This study | SSY4577 |
| *BFP-hph roq1(R22A) leu2::P$_{ADH}$-Rtn1Pho8*-GFP-LEU2* | This study | SSY4552 |
| *BFP-hph roq1(V58E) leu2::P$_{ADH}$-Rtn1Pho8*-GFP-LEU2* | This study | SSY4553 |
| *BFP-hph roq1(4A) leu2::P$_{ADH}$-Rtn1Pho8*-GFP-LEU2* | This study | SSY4589 |
| *BFP-hph roq1::kan-GEM-P$_{GAL}$-roq1(R22A)* | This study | SSY4580 |
| *BFP-hph roq1::kan-GEM-P$_{GAL}$-roq1(V58E)* | This study | SSY4581 |
| *BFP-hph roq1::kan-GEM-P$_{GAL}$-roq1(4A)* | This study | SSY4595 |
| *BFP-hph roq1::nat-GEM-P$_{GAL}$-roq1 ura3::P$_{GPD}$-Ub-R-mCherry-sfGFP-kan* | Szoradi et al, 2018 | SSY1880 |
| *BFP-hph kan::GEM-P$_{GAL}$-roq1(R22A) ura3::P$_{GPD}$-Ub-R-mCherry-sfGFP-nat* | This study | SSY4582 |
| *BFP-hph kan::GEM-P$_{GAL}$-roq1(V58E) ura3::P$_{GPD}$-Ub-R-mCherry-sfGFP-nat* | This study | SSY4583 |
| *BFP-hph kan::GEM-P$_{GAL}$-roq1(4A) ura3::P$_{GPD}$-Ub-R-mCherry-sfGFP-nat* | This study | SSY4594 |
| *BFP-hph roq1::nat-GEM-P$_{GAL}$-roq1 ura3::P$_{GPD}$-Ub-F-mCherry-sfGFP-kan* | Szoradi et al, 2018 | SSY1870 |
| *BFP-hph kan::GEM-P$_{GAL}$-roq1(R22A) ura3::P$_{GPD}$-Ub-F-mCherry-sfGFP-nat* | This study | SSY4584 |
| *BFP-hph kan::GEM-P$_{GAL}$-roq1(V58E) ura3::P$_{GPD}$-Ub-F-mCherry-sfGFP-nat* | This study | SSY4585 |
| *BFP-hph kan::GEM-P$_{GAL}$-roq1(4A) ura3::P$_{GPD}$-Ub-F-mCherry-sfGFP-nat* | This study | SSY4593 |
| **Recombinant DNA** | | |
| pRS316-P$_{Roq1}$-Roq1 | pSS254 | Szoradi et al, 2018 |
| pCA528 | pSS685 | Andréasson et al, 2008 |
| pCA528-Roq1(22-104) | pSS850 | This study |
| pCA528-Roq1(22-104)(R22A) | pSS1081 | This study |
| pCA528-Roq1(22-104)-HA | pSS1298 | This study |
| pCA528-Roq1(22-60)-HA | pSS1340 | This study |
| pCA528-Roq1(22-60)(R22A)-HA | pSS1412 | This study |
| pRS305-P$_{ADH}$-Rtn1Pho8*-FLAG-GFP | pSS174 | Szoradi et al, 2018 |
| pCA528-Pho8* | pSS1233 | This study |
| pMAl-c2E-Hsp42-FLAG | pSS840 | Ungelenk et al, 2016 |
| pCA528-Pho8*-MBP | pSS1240 | This study |
| pCA528-Pho8-MBP | pSS1250 | This study |
| pCA528-GFP | pSS841 | Schmidt et al, 2009 |
| pCA528-R-GFP | pSS1126 | This study |
| pCA528-F-GFP | pSS1231 | This study |
| pCA528-Cup9-Strep | pSS1179 | This study |
| pCA528-Mgt1-MBP | pSS1260 | This study |
| YEplac181-P$_{ADH}$-FLAG-Ubr1 | pSS1478 | Xia et al, 2008 |
| YEplac181-P$_{ADH}$-Chk1-ALFA-FLAG | pSS1480 | This study |
| pCA528-Rad6 | pSS1211 | Axel Mogk |
| pET24a-Ulp1-His$_6$ | pSS686 | Matthias Mayer |
| pET3a-Ubiquitin | pSS1209 | Frauke Melchior |
| pRS306N | pSS380 | Taxis and Knop, 2006 |
| pRS306N-P$_{ROQ1}$-Roq1 | pSS1292 | This study |
| pRS306N-P$_{ROQ1}$-Ub-Roq1(22-104) | pSS1293 | This study |
| pRS306N-P$_{ROQ1}$-Ub-Roq1(22-60) | pSS1305 | This study |
| pFA6a-ALFA-klTRP1 | pSS1363 | This study |
| pFA6a-ALFA-HIS3MX6 | pSS1382 | This study |
| pRS416-P$_{GPD}$ | pSS027 | Mumberg et al, 1995 |
| pRS316-P$_{ROQ1}$-Roq1-HA(74) | pSS776 | Szoradi et al, 2018 |
| pRS316-P$_{ROQ1}$-Roq1(R22A)-HA(74) | pSS835 | Szoradi et al, 2018 |
| pRS316-P$_{ROQ1}$-Ub-Roq1(22-104)-HA(74) | pSS1124 | This study |

| Reagent/resource | Reference or source | Identifier or catalog number |
|---|---|---|
| pRS316-P$_{ROQ1}$-Ub-Roq1(22-104)(R22A)-HA(74) | pSS827 | Szoradi et al, 2018 |
| pRS416-P$_{GPD}$-Ub-Roq1(22-104)-HA(74) | pSS764 | Szoradi et al, 2018 |
| pRS416-P$_{GPD}$-Ub-Roq1(22-104)(R22A)-HA(74) | pSS927 | Szoradi et al, 2018 |
| pRS416-P$_{GPD}$-Roq1-HA(74) | pSS649 | Szoradi et al, 2018 |
| pRS416-P$_{GPD}$-Ub-Roq1(22-104)-HA(74) | pSS764 | Szoradi et al, 2018 |
| pRS416-P$_{GPD}$-Ub-Roq1(22-104)(E36K)-HA(74) | pSS1067 | This study |
| pRS416-P$_{GPD}$-Ub-Roq1(22-104)(M39K)-HA(74) | pSS1068 | This study |
| pRS416-P$_{GPD}$-Ub-Roq1(22-104)(G40C)-HA(74) | pSS1069 | This study |
| pRS416-P$_{GPD}$-Ub-Roq1(22-104)(N45K)-HA(74) | pSS1070 | This study |
| pRS416-P$_{GPD}$-Ub-Roq1(22-104)(Q50K)-HA(74) | pSS1071 | This study |
| pRS416-P$_{GPD}$-Ub-Roq1(22-104)(I54K)-HA(74) | pSS1072 | This study |
| pRS416-P$_{GPD}$-Ub-Roq1(22-104)(Y55H)-HA(74) | pSS1073 | This study |
| pRS416-P$_{GPD}$-Ub-Roq1(22-104)(Y55N)-HA(74) | pSS1074 | This study |
| pRS416-P$_{GPD}$-Ub-Roq1(22-104)(Y55D)-HA(74) | pSS1075 | This study |
| pRS416-P$_{GPD}$-Ub-Roq1(22-104)(Y56N)-HA(74) | pSS1076 | This study |
| pRS416-P$_{GPD}$-Ub-Roq1(22-104)(V58E)-HA(74) | pSS1077 | This study |
| pRS416-P$_{GPD}$-Ub-Roq1(22-104)(N62K)-HA(74) | pSS1078 | This study |
| pRS416-P$_{GPD}$-Ub-Roq1(22-104)(I65K)-HA(74) | pSS1079 | This study |
| pRS416-P$_{GPD}$-Ub-Roq1(22-104)(E67K)-HA(74) | pSS1080 | This study |
| pRS416-P$_{GPD}$-Ub-Roq1(22-104)(I54R)-HA(74) | pSS1463 | This study |
| pRS416-P$_{GPD}$-Ub-Roq1(22-104)(E36A)-HA(74) | pSS1133 | This study |
| pRS416-P$_{GPD}$-Ub-Roq1(22-104)(M39A)-HA(74) | pSS1134 | This study |
| pRS416-P$_{GPD}$-Ub-Roq1(22-104)(N45A)-HA(74) | pSS1135 | This study |
| pRS416-P$_{GPD}$-Ub-Roq1(22-104)(Q50A)-HA(74) | pSS1136 | This study |
| pRS416-P$_{GPD}$-Ub-Roq1(22-104)(I54A)-HA(74) | pSS1137 | This study |
| pRS416-P$_{GPD}$-Ub-Roq1(22-104)(N62A)-HA(74) | pSS1139 | This study |
| pRS416-P$_{GPD}$-Ub-Roq1(22-104)(I65A)-HA(74) | pSS1140 | This study |

| Reagent/resource | Reference or source | Identifier or catalog number |
|---|---|---|
| pRS416-P$_{GPD}$-Ub-Roq1(22-104)(E67A)-HA(74) | pSS1141 | This study |
| pRS416-P$_{GPD}$-Ub-Roq1(22-104)(V53E)-HA(74) | pSS1187 | This study |
| pRS416-P$_{GPD}$-Ub-Roq1(22-104)(I54E)-HA(74) | pSS1142 | This study |
| pRS416-P$_{GPD}$-Ub-Roq1(22-104)(Y56D)-HA(74) | pSS1188 | This study |
| pRS416-P$_{GPD}$-Ub-Roq1(22-104)(F57D)-HA(74) | pSS1177 | This study |
| pRS416-P$_{GPD}$-Ub-Roq1(22-104)(L60E)-HA(74) | pSS1143 | This study |
| pRS416-P$_{GPD}$-Ub-Roq1(22-104)(Y55A)-HA(74) | pSS1138 | This study |
| pRS416-P$_{GPD}$-Ub-Roq1(22-104)(Y56A)-HA(74) | pSS1189 | This study |
| pRS416-P$_{GPD}$-Ub-Roq1(22-104)(F57A)-HA(74) | pSS1176 | This study |
| pRS416-P$_{GPD}$-Ub-Roq1(22-104)(V58A)-HA(74) | pSS1190 | This study |
| pRS416-P$_{GPD}$-Ub-Roq1(22-104)(Y55A,Y56A,F57A,V58A)-HA(74) | pSS1300 | This study |
| pRS415-P$_{ADH}$-FLAG-Ubr1 | pSS930 | Szoradi et al, 2018 |
| pRS416-P$_{GPD}$-Ub-Roq1(22-104)-HA(74)(R22A,Y55A,Y56A,F57A,V58A) | pSS1301 | This study |
| pCA528-Roq1(22-104)(Y55A,Y56A,F57A,V58A) | pSS1283 | This study |
| pCA528-Roq1(22-104)(R22A,Y55A,Y56A,F57A,V58A) | pSS1285 | This study |
| pCA528-Roq1(22-104)(V58E) | pSS1208 | This study |
| pCA528-Roq1(22-60)(V58E)-HA | pSS1416 | This study |
| pRS306K-P$_{GPD}$-Ub-R-mCherry-sfGFP | pSS725 | Khmelinskii et al, 2012 |
| pRS306K-P$_{GPD}$-Ub-F-mCherry-sfGFP | pSS715 | Khmelinskii et al, 2012 |
| pRS306N-P$_{GPD}$-Ub-R-mCherry-sfGFP | pSS1589 | This study |
| pRS306N-P$_{GPD}$-Ub-F-mCherry-sfGFP | pSS1590 | This study |
| pRS416-P$_{GPD}$-Ub-Roq1(83aa$_{GSP}$)-HA(74) | pSS1591 | This study |
| pRS416-P$_{GPD}$-Ub-Roq1(73aa)-HA(74) | pSS1592 | This study |
| pRS416-P$_{GPD}$-Ub-Roq1(69aa)-HA(74) | pSS1593 | This study |
| pRS416-P$_{GPD}$-Ub-Roq1(66aa)-HA(74) | pSS1594 | This study |
| pRS416-P$_{GPD}$-Ub-Roq1(63aa)-HA(74) | pSS1595 | This study |
| pRS416-P$_{GPD}$-Ub-Roq1(66aa$_{GS}$)-HA(74) | pSS1596 | This study |
| pRS416-P$_{GPD}$-Ub-Roq1(66a$_{GSP}$)-HA(74) | pSS1597 | This study |
| pCA528-Roq1(22aa) | pSS1456 | This study |
| pCA528-Roq1(22aa$_{GS}$) | pSS1454 | This study |
| pCA528-Roq1(22aa$_{GSP}$) | pSS1455 | This study |
| **Antibodies** | | |
| Rat anti-HA (clone 3F10) | Roche | 11867423001; AB_390918 |

| Reagent/resource | Reference or source | Identifier or catalog number |
|---|---|---|
| Mouse anti-Pho8 (clone 1D3A10) | Abcam | Ab113688; AB_10860792 |
| Goat anti-Luciferase | Merck | AB3256; AB_91422 |
| Mouse anti-GFP (clones 7.1/13.1) | Roche | 11814460001; AB_390913 |
| Anti-ALFA IRDye 800CW | NanoTag Biotechnologies | N1502-Li800-L; AB_3075985 |
| Mouse anti-Pgk1 (clone 22C5D8) | Abcam | ab113687; AB_10861977 |
| Strep-Tactin HRP | Iba Lifesciences | 2-1502-001 |
| Mouse anti-MBP (clone MBP-17) | Merck | M6295; AB_260613 |
| Mouse anti-FLAG (clone M2) | Merck | F1804; AB_262044 |
| Donkey anti-rat HRP | Jackson ImmunoResearch | 712-035-153; AB_2340639 |
| Goat anti-mouse HRP | Thermo Fisher Scientific | 31432; AB_228302 |
| Rabbit anti-goat HRP | Jackson ImmunoResearch | 305-035-045; AB_2339403 |
| Goat anti-mouse Alexa-680 | Invitrogen | A21057; AB_141436 |
| **Oligonucleotides and other sequence-based reagents** | | |
| Primer sequences | This study | Appendix Table S1 |
| **Chemicals, enzymes, and other reagents** | | |
| Adenosine triphosphate (ATP) | Sigma-Aldrich | A2383-5G |
| Amylose resin | New England Biolabs (NEB) | E8021L |
| Anti-DYKDDDDK (FLAG) affinity resin | Thermo Fisher Scientific | A36803 |
| Arg-Ala dipeptide (H-RA-OH) | Peptides & Elephants | EP06653/ 1809W16 |
| cOmplete protease inhibitors | Roche | 04693116001 |
| cOmplete protease inhibitors (EDTA free) | Roche | 04693132001 |
| Cycloheximide | Applichem | A0879 |
| D-desthiobiotin | Sigma-Aldrich | D1411-500MG |
| DYKDDDDK (FLAG) peptide | Thermo Fisher Scientific | A36806 |
| Estradiol | Sigma-Aldrich | E8875 |
| Glutathione Sepharose 4B | Amersham | 17075601 |
| Guanidinium hydrochloride | Sigma-Aldrich | 50950-1KG |
| L-Glutathione, reduced | Sigma-Aldrich | G4251-10G |
| HPDP-Biotin | Thermo Fisher Scientific | 21341 |
| Isopropyl β-D-1-thiogalactopyranoside (IPTG) | ZellX Biochem | ZXB-06-100-100 |
| Leu-Ala dipeptide (H-LA-OH) | Peptides & Elephants | EP06553/ 1809W17 |

| Reagent/resource | Reference or source | Identifier or catalog number |
|---|---|---|
| NHS-PEG4 biotin | Thermo Fisher Scientific | 21330 |
| Protino Ni-IDA beads | Macherey-Nagel | 745210.30 |
| Recombinant Human Ubiquitin-Activating Enzyme (UBE1) | R&D Systems (Bio-Techne) | E-305-025 |
| Recombinant human ubiquitin (K0) | R&D Systems (Bio-Techne) | UM-NOK-01M |
| Recombinant human ubiquitin (K0) | MoBiTec | E1610-UBP |
| Recombinant human ubiquitin (WT) | R&D Systems (Bio-Techne) | U-100H-10M |
| Recombinant luciferase | Promega | E1702 |
| D-(+)-Trehalose dihydrate | Sigma-Aldrich | T9531-10G |
| Tunicamycin | Sigma-Aldrich | 654380-50MG |
| **Software** | | |
| ImageJ | 2.0.0-rc-69/ 1.52n | National Institute of Health |
| Image Studio | 5.2.5 | LI-COR |
| FACS software | FlowJo 10.10.0 | |
| FACS software | BD FACSDiva | |
| Octet software | Data Analysis HT 12.0 | Sartorius |
| **Other** | | |
| Pierce™ Rapid Gold BCA Protein Assay Kit | Thermo Scientific | A53225 |
| SuperSignal™ West Pico PLUS Chemiluminescent Substrate | Thermo Scientific | 34577 |
| Äkta Pure 25 | Cytiva | |
| Amersham Imager 600 | GE Healthcare | |
| FACS Canto II | BD | |
| FastPrep-24 | MP | |
| Glutathione Sepharose 4B | Amersham | 17075601 |
| Hiload 16/600 Superdex S30 prep grade | Cytiva | 28989331 |
| Hiload 16/60 Superdex S75 prep grade | GE Healthcare | 28-9893-33 |
| Hiload 16/60 Superdex S200 size exclusion prep grade resin | GE Healthcare | 17-1043-01 |
| HisTrap FF crude column, 1 ml | Cytiva | 17-5319-01 |
| MBPTrap HP column, 1 ml | Cytiva | 29048641 |
| GSTrap4B column, 1 ml | Cytiva | 29-0486-09 |
| Microfluidizer M-110L | Microfluidics | |
| OctetRed96e | Sartorius | |
| Odyssey CLx LI-COR imager | LI-COR | |
| Resource S column | Cytiva | 17117801 |
| StrepTrap HP column, 1 ml | Cytiva | 29-0486-53 |

*4A* Y55A,Y56A,F57A,V58A, *BFP-hph* his3::P~GPD~-BFP-hph, *cat#* catalog number, *GEM* Gal4DBD-EstR-Msn2TAD, *HRP* horseradish peroxidase, *RRID* research resource identifier.

## Plasmids

Plasmids were constructed by Gibson assembly and site-directed mutagenesis and are listed in the Reagents and Tools table. Oligonucleotides used for plasmid generation are listed in Appendix Table S1. To create pCA528-Roq1(22–104), Roq1(22–104) from pRS316-$P_{ROQ1}$-Roq1 (Szoradi et al, 2018) was inserted into pET24a-His$_6$-SUMO (pCA528; Andreasson et al, 2008). Additional insertion of HA or ALFA tags yielded pCA528-Roq1(22–104)-HA and -ALFA. pCA528-Roq1(22–60)-HA was created by replacing Roq1(61–104) of pCA528-Roq1(22–104) with a sequence encoding an HA tag and a stop codon. To generate p416-$P_{GPD}$-Ub-Roq1(22–60), a stop codon was introduced into pRS416-$P_{GPD}$-Ub-Roq1(22–104)-HA(74). To generate pRS316-$P_{ROQ1}$-Ub-Roq1(22–104)-HA(74), an HA tag was inserted into pRS316-$P_{ROQ1}$-Ub-Roq1(22–104) (Szoradi et al, 2018). To create pCA528-Pho8*-MBP, Pho8* from pRS305-$P_{ADH}$-Rtn1Pho8*-FLAG-GFP (Szoradi et al, 2018) was inserted into pCA528, generating pCA528-Pho8*. Subsequently, MBP from pMal-c2E-Hsp42-FLAG (Ungelenk et al, 2016) was modified with an upstream sequence encoding a GGSGGGSGG linker and inserted into pCA528-Pho8*. pCA528-Pho8-MBP was created by reverting the F352S mutation in pCA528-Pho8*-MBP. pCA528-R-GFP and pCA528-F-GFP were generated by inserting codons for arginine or phenylalanine and a sequence encoding a SKGEELFYGV linker into pCA528-GFP (Schmidt et al, 2009). To generate pCA528-Cup9-Strep, Cup9 from yeast genomic DNA was modified with a Strep tag II and inserted into pCA528. To generate pCA528-Mgt1-MBP, Mgt1 from yeast genomic DNA was inserted into pCA528-Pho8*-MBP, replacing Pho8*. To generate YEplac181-$P_{ADH}$-Chk1-ALFA-FLAG, Chk1 from yeast genomic DNA was modified with an ALFA-FLAG tag and inserted into YEplac181-$P_{ADH}$-FLAG-Ubr1 (pFLAGUBR1SBX; Xia et al, 2008), replacing FLAG-Ubr1. pRS306N-$P_{GPD}$-Ub-R/F-mCherry-sfGFP were derived from the corresponding pRS306K-based plasmids described in Khmelinskii et al, 2012.

## Protein purification

Ubr1 was purified essentially as described (Du et al, 2002). Protease-deficient yeast expressing FLAG-Ubr1 (SSY2908) were grown to OD$_{600}$ = 2 at 30 °C in 12 L YPD medium, washed with cold PBS, resuspended in 6 ×10 ml lysis buffer (50 mM HEPES pH 7.5, 0.2 M NaCl, 10% glycerol, 0.5% NP-40, 1 mM PMSF, Roche complete protease inhibitors) and disrupted by 10 passages through a Microfluidics M-110L microfluidizer at 1200 bar. Lysates were cleared at 11,200 × $g$ for 30 min and the supernatant was incubated with 0.9 ml anti-FLAG agarose beads (Thermo Fisher Scientific) at 4 °C for 2 h. Beads were washed with 20 ml buffer A (50 mM HEPES pH 7.5, 1 M NaCl, 10% glycerol, 0.5% NP-40) and 20 ml buffer B (50 mM HEPES pH 7.5, 0.2 M NaCl, 10% glycerol), and FLAG-Ubr1 was eluted with 5 × 500 µl buffer B containing 1 mg/ml FLAG peptide (Thermo Fisher Scientific). Protein-containing eluates were pooled and concentrated with an Amicon Ultra centrifugal filter with a 100 kDa molecular weight cut-off.

Chk1 was purified from protease-deficient yeast (PWY0005) harboring YEPlac181-$P_{ADH}$-Chk1-ALFA-FLAG. Cells were grown to OD$_{600}$ = 2 at 30 °C in SCD medium without leucine, and Chk1-ALFA-FLAG was isolated as described for FLAG-Ubr1.

Roq1 variants were purified from *E. coli* BL21(DE3) harboring pCA528-based expression plasmids for His$_6$-SUMO-Roq1. Cells were grown to OD$_{600}$ = 0.7 at 37 °C in 6 L 2xYT medium with 25 µg/ml kanamycin and 34 µg/ml chloramphenicol, and Roq1 expression was induced with 1 mM IPTG (ZellBio) at 37 °C for 3 h. Cells were harvested at 1000 × $g$ at 4 °C for 15 min, washed with cold water and resuspended in 60 ml lysis buffer (50 mM Bis-Tris pH 6.0, 500 mM NaCl, 20 mM imidazole, 5 mM MgCl$_2$, 2 mM 2-mercaptoethanol [2-ME], 1 mM PMSF, Roche protease inhibitors without EDTA). A dollop of DNase (Merck) was added and the cell suspension was stirred at 4 °C for 20 min. Cells were disrupted by 6 passages through an M-110L microfluidizer. For subsequent immobilized metal affinity chromatography (IMAC), the lysate was cleared at 11,200 × $g$ at 4 °C for 30 min and incubated with 1.5 g Ni-IDA beads (Macherey-Nagel) for 60 min. The beads were applied to a gravity flow column, washed with lysis buffer, wash buffer (50 mM Bis-Tris pH 6.0, 50 mM imidazole, 500 mM NaCl, 5 mM MgCl$_2$, 2 mM 2-ME), wash buffer with 750 mM NaCl and wash buffer with 5 mM ATP. Bound protein was eluted with elution buffer (50 mM Bis-Tris pH 6.0, 150 mM NaCl, 250 mM imidazole, 2 mM 2-ME) and protein-containing fractions were pooled. To remove the His$_6$-SUMO, the eluate was mixed with 25 µl His-tagged SUMO protease Ulp1 (purified as described below) and dialyzed with a 3 kDa molecular weight cut-off against dialysis buffer (50 mM Bis-Tris pH 6.0, 150 mM NaCl, 2 mM 2-ME) at 4 °C overnight. To remove cleaved His$_6$-SUMO, uncleaved His$_6$-SUMO-Roq1 and Ulp1-His$_6$, Ni-IDA beads were added for 30 min and collected with a gravity flow column. The flow-through was concentrated and applied to a Hiload 16/600 Superdex S30 prep grade size exclusion column (Cytiva) equilibrated with 50 mM HEPES pH 7.5, 150 mM NaCl. Protein-containing fractions were pooled and adjusted to 10% (v/v) glycerol.

To purify Rad6, *E. coli* BL21(DE3) harboring pCA528-Rad6 for the expression of His$_6$-SUMO-Rad6 were grown in 2xYT medium with antibiotics, treated with IPTG at 37 °C for 4 h and lysed as above in 50 mM NaH$_2$PO$_4$ pH 8.0, 300 mM NaCl, 5 mM MgCl$_2$, 2 mM 2-ME, 10 mM imidazole and Roche protease inhibitors without EDTA. After removal of the His$_6$-SUMO tag with the SUMO protease Ulp1 (see above), Rad6 was further purified on a Hiload 16/60 Superdex S75 prep grade size exclusion column (GE HealthCare) equilibrated with 25 mM HEPES pH 7.5, 25 mM KCl, 5 mM MgCl$_2$, 1 mM DTT. Protein-containing fractions were pooled and adjusted to 10% (v/v) glycerol.

R-GFP and F-GFP were purified from *E. coli* BL21(DE3) harboring pCA528-R-GFP or pCA528-F-GFP for the expression of His$_6$-SUMO-R-GFP or His$_6$-SUMO-F-GFP. The same protocol was used as for the purification of Rad6 (see above), except that a Hiload 16/60 Superdex S200 prep grade column (GE HealthCare) was used.

To purify Pho8 and Pho8*, *E. coli* BL21(DE3) harboring pCA528-Pho8-MBP or pCA528-Pho8*-MBP for the expression of His$_6$-SUMO-Pho8/Pho8*-MBP were grown in LB medium containing 25 µg/ml kanamycin and 34 µg/ml chloramphenicol, treated with IPTG at 30 °C for 3 h and lysed as above in 50 mM HEPES pH 7.5, 500 mM NaCl, 5 mM MgCl$_2$, 10% glycerol, 10 mM imidazole, 0.4% CHAPS, 1 mM PMSF and Roche protease inhibitors without EDTA. Lysates were cleared and applied to a HisTrap FF crude affinity chromatography column (GE HealthCare) equilibrated with IMAC buffer (50 mM HEPES pH 7.5, 500 mM NaCl, 5 mM MgCl$_2$, 10% glycerol). The column was washed with 10 volumes IMAC buffer containing 10 mM imidazole and 10 mM ATP, and

protein was eluted with IMAC buffer containing 250 mM imidazole. The eluate was mixed with Ulp1 and dialyzed overnight against dialysis buffer (50 mM HEPES pH 7.5, 150 mM NaCl, 10 mM MgCl$_2$, 2 mM 2-ME). The dialysate was further batch-purified with amylose resin (NEB) equilibrated with dialysis buffer, bound protein was eluted with 10 mM maltose in dialysis buffer, the eluate was concentrated and loaded onto a Hiload 16/60 Superdex S200 size exclusion prep grade column equilibrated with 50 mM HEPES pH 7.5, 150 mM NaCl, 10 mM MgCl$_2$. Protein-containing fractions were pooled and adjusted to 10% (v/v) glycerol.

To purify Cup9, *E. coli* BL21(DE3) harboring pCA528-Cup9-Strep for the expression of His$_6$-SUMO-Cup9-Strep were grown in LB medium with antibiotics, treated with IPTG at 37 °C for 3 h and lysed as above in 20 mM HEPES pH 7.0, 500 mM NaCl, 10 mM imidazole, 5 mM MgCl$_2$, 2 mM 2-ME, 1 mM PMSF and Roche protease inhibitors without EDTA. IMAC purification and Ulp1 digest were done as described for Roq1. The dialysate was loaded onto a StrepTrap HP column (Cytiva) equilibrated with Strep buffer (20 mM HEPES pH 7.0, 500 mM NaCl). The column was washed with 10 volumes Strep buffer, Cup9-Strep was eluted with 2.5 mM D-desthiobiotin in Strep buffer with 2 mM 2-ME and applied to a Hiload 16/60 Superdex S200 size exclusion prep grade column equilibrated with 20 mM HEPES pH 7.5, 500 mM NaCl, 1 mM DTT. Protein-containing fractions were pooled and adjusted to 10% (v/v) glycerol.

To purify Mgt1, *E. coli* BL21(DE3) harboring pCA528-Mgt1-MBP for the expression of His$_6$-SUMO-Mgt1-MBP were grown in LB medium with antibiotics, treated with IPTG at 30 °C for 3 h and lysed as above. After IMAC, protein-containing fractions were pooled and batch-purified with amylose resin (NEB) equilibrated with 50 mM HEPES pH 7.5, 500 mM NaCl. The column was washed, and protein was eluted with 10 mM maltose, digested with Ulp1, dialyzed, concentrated, and applied to a Hiload 16/60 Superdex S75 prep grade size exclusion column (GE HealthCare) equilibrated with 50 mM HEPES pH 7.5, 150 mM NaCl, 10 mM MgCl$_2$. Protein-containing fractions were pooled and adjusted to 10% (v/v) glycerol.

To purify Ulp1 protease, *E. coli* BL21(DE3) harboring pET24a-Ulp1-His$_6$ were grown at 30 °C in 2xYT medium containing 100 µg/ml ampicillin and 34 µg/ml chloramphenicol, and Ulp1 expression was induced with 0.5 mM ITPG at 20 °C overnight. Cells were lysed in 20 mM Tris pH 7.9, 100 mM KCl, 0.6% Brij-58, 5 mM MgCl$_2$, 2 mM 2-ME, 1 mM PMSF, and Roche protease inhibitors without EDTA. Ulp1-His$_6$ was immobilized on Ni-IDA beads, washed with 300 ml wash buffer A (20 mM Tris pH 7.9, 2 M urea, 100 mM KCl, 0.1% Brij-58, 2 mM 2-ME), 300 ml buffer B (20 mM Tris pH 7.9, 1 M KCl, 0.1% Brij-58, 2 mM 2-ME) and 300 ml buffer C (20 mM Tris pH 7.9, 100 mM KCl, 2 mM 2-ME) and eluted with lysis buffer containing 250 mM imidazole. Protein-containing fractions were pooled and dialyzed overnight against dialysis buffer (40 mM HEPES pH 7.9, 150 mM KCl, 10% glycerol, 2 mM 2-ME). Aggregates were removed by centrifugation at 4000 rpm at 4 °C for 15 min, protein was concentrated to 8 mg/ml and adjusted to 50% (v/v).

To purify untagged ubiquitin, *E. coli* BL21(DE3) harboring pET3a-ubiquitin was grown in LB medium with 100 µg/ml ampicillin and 34 µg/ml chloramphenicol, treated with 1 mM IPTG at 37 °C for 4 h and lysed in 50 mM Tris-HCl pH 7.6, 0.02% NP-40,

10 mM MgCl$_2$, 1 mM DTT, 1 mM PMSF and Roche complete protease inhibitors. Proteins were precipitated with cold 70% perchloric acid and separated from soluble ubiquitin by centrifugation. The supernatant was dialyzed against 50 mM ammonium acetate pH 4.5. Ubiquitin was further purified by cation exchange on a Resource S column (Cytiva) equilibrated with dialysis buffer and eluted with a linear gradient of 0–0.5 M NaCl in 50 mM ammonium acetate pH 4.5 over 20 column volumes. Protein-containing fractions were pooled, concentrated, and applied to a Hiload 16/60 Superdex S75 prep grade size exclusion column (GE HealthCare) equilibrated with 50 mM Tris-HCl pH 7.6, 150 mM NaCl, 1 mM DTT.

The purity of the isolated recombinant proteins was assessed by SDS-PAGE and Coomassie staining (see Appendix Fig. S1).

## Chemical unfolding of Luciferase

To obtain unfolded but soluble Luciferase$^U$, recombinant firefly Luciferase (Promega) was diluted fourfold in denaturation buffer (30 mM Tris pH 7.4, 50 mM KCl, 5 mM MgCl$_2$, 7.5 M guanidinium chloride, 10 mM DTT) and incubated on ice for 30 min. Luciferase was then diluted 100-fold in stabilization buffer (30 mM Tris pH 7.4, 50 mM KCl, 5 mM MgCl$_2$, 0.2 M trehalose, 1 mM DTT) according to Singer and Lindquist, 1998. To eliminate aggregates, samples were centrifuged at $13{,}200 \times g$ at 4 °C for 30 min and the supernatant was used for in vitro ubiquitination assays. To obtain folded Luciferase$^N$, recombinant firefly Luciferase was treated as above except that guanidinium chloride was omitted.

## In vitro ubiquitination assays

Ubiquitination assays were done in 10 µl containing 0.1 µM Ube1 (R&D Systems), 4 µM Rad6, 0.25 µM Ubr1, 80 µM wild-type or lysine-free ubiquitin (MoBiTec) and 0.2 µM substrate protein in reaction buffer (50 mM HEPES pH 7.5, 0.15 M NaCl, 10 mM MgCl$_2$, 5 mM ATP). Roq1 variants were used at 2.5 µM unless stated otherwise. The RA dipeptide (peptides & elephants) was used at a concentration of 1 mM. Reactions were incubated at 30 °C for the times indicated and stopped with SDS-PAGE sample buffer containing DTT and used for western blotting. Each ubiquitination assay was performed at least twice to ensure reproducibility of the results. To hydrolyze ester bonds between ubiquitin and substrate proteins, ubiquitination reactions were set up as above and incubated at 30 °C for 30 min. SDS-PAGE sample buffer was added and samples were incubated at 65 °C for 5 min. NaOH was added to 0.25 mol/L, samples were incubated at room temperature for 10 min, neutralized with 0.25 mol/L HCl and used for western blotting. Control samples were treated as above except that NaOH and HCl were replaced with water.

## Western blotting

Proteins were separated by Tris-glycine or Tris-tricine SDS-PAGE and transferred onto nitrocellulose membranes by wet blotting. Membranes were probed with primary and HRP-coupled secondary antibodies, HRP conjugates or fluorescently labeled nanobodies (Reagents and Tools table). For chemiluminescence detection, membranes were incubated with SuperSignal West Pico Plus substrate (Thermo Fisher Scientific) and analyzed with an

Amersham Imager 600 (GE HealthCare). For fluorescence detection, membranes were analyzed with an Odyssey CLx imaging system (LI-COR).

## Pho8* solubility assay

To determine Pho8* solubility, duplicate 10 µl ubiquitination assays including Roq1 were set up as described above. After 0 or 90 min, samples were saved as input or centrifuged at $18,000 \times g$ at 4 °C for 30 min. Supernatants were collected and pellets were resuspended in 10 µl reaction buffer (50 mM HEPES, pH 7.5, 0.15 M NaCl, 10 mM $MgCl_2$). Equal volumes of input, supernatant and pellet samples were analyzed by western blotting.

## Biolayer interferometry

The Roq1-Ubr1 interaction was analyzed with the OctetRed96e system (Sartorius). Roq1(22-104) variants were biotinylated at C103, the single cysteine in Roq1, by incubation with a 1.2-fold molar excess of thiol-reactive EZ-Link™ HPDP-Biotin (Thermo Fisher Scientific) at room temperature for 90 min. Unbound biotin was removed with a Zeba spin desalting column (Thermo Fisher Scientific) and biotinylated Roq1 was immobilized on streptavidin biosensors (Sartorius) that had been hydrated in assay buffer (50 mM HEPES, 150 mM NaCl, 10 mM $MgCl_2$, 0.05% Tween-20, 0.02% BSA) for 10 min. Binding assays were performed in black 96-well plates (Greiner) in 200 µl at 1000 rpm. Assays consisted of (1) a washing step in assay buffer for 1 min, (2) a loading step with 1.25 µg/ml wild-type Roq1 or mutant variants for 10 min, (3) a washing step for 1 min, (4) a baseline step for data normalization for 1 min, and (5) an association step with 50 nM Ubr1 (Figs. 4C and 5A) or 100 nM Ubr1 (Fig. 2D) for 5 min. Rad6-Ubr1 interaction was analyzed with Rad6 biotinylated with a 1.2-fold molar excess of the amino-reactive EZ-Link™ NHS-PEG4-Biotin (Thermo Fisher Scientific) at room temperature for 30 min. The setup of the assay was the same as above except that 0.5 µg/ml Rad6 were used for loading. To test the effect of Roq1 on Rad6-Ubr1 association, 100 nM Ubr1 and 200 nM or 1 µM Roq1 were used. The Roq1-Ubr1 and Rad6-Ubr1 interactions were essentially irreversible so that only the association could be analyzed.

## Bioinformatic analyses

The structure prediction for full-length Roq1 was retrieved from the AlphaFold Protein Structure Database (Varadi et al, 2024). The disorder prediction was obtained by averaging estimates for the disorder tendency across the full-length Roq1 sequence from 12 prediction algorithms, namely AIUPred, DisEMBL, DISpro, Espritz, IsUnstruct, IUPred3, Metapredict, MFDp, NetsurfP 3.0, PONDR VSL2, PrDOS and RONN (Kurgan, 2022). To identify Roq1 sequence homologs, a protein-protein BLAST search excluding S. cerevisiae was carried out via the NCBI home page. An alignment of the retrieved sequences from *Saccharomyces* and *Kazachstania* species was done with CLUSTALW.

## Yeast strains

Strains used in this study were derived from S. cerevisiae W303 mating type a and are listed in the Reagents and Tools table. Gene

tagging with the ALFA tag to generate strain SSY4598 was done with PCR products of plasmids pSS1363 and pSS1382. To generate strain SSY2959, the $P_{ADH}$-FLAG-Ubr1 cassette was amplified from plasmid pRS415-$P_{ADH}$-FLAG-Ubr1 (pSS930) and integrated into the *leu2* locus of SSY792. To generate SSY3543, 3544, 3545 and 3551, the expression cassettes of pRS306N-derived plasmids were linearized by PCR and integrated into the *ura3* locus of SSY835. Point mutations were introduced into the endogenous *ROQ1* gene by the delitto perfetto approach (Storici and Resnick, 2006).

## Yeast culture and cell lysis

Cells were grown at 30 °C in SCD medium, which consisted of 0.7% yeast nitrogen base without amino acids (Sigma, Taufkirchen, Germany), amino acids, and 2% glucose. Where appropriate, uracil was omitted to maintain plasmid selection. For subsequent western blots, cultures were grown to mid-log phase ($OD_{600} = 0.5$-1), cells were collected by centrifugation, washed with water, resuspended in lysis buffer (50 mM HEPES pH 7.5, 0.5 mM EDTA, 1 mM PMSF, Roche complete protease inhibitors), and disrupted by bead beating with a FastPrep-24 (MP Biomedicals). Proteins were solubilized by the addition of 1.5% (w/v) SDS and incubation at 65 °C for 5 min. Lysates were cleared at $16,000 \times g$ at 4 °C for 2 min, and protein concentrations were determined with a BCA kit (Pierce).

To determine the abundance of Ubr1 and Roq1 upon proteotoxic stress, wild-type control cells (SSY122) and cells with ALFA-tagged Ubr1 and Roq1 (SSY4598) were grown to mid-log phase, left untreated or treated with 1 µg/ml tunicamycin for 2 h. Cells were harvested, lysed as described above and 50 µg total protein per sample were analyzed by western blotting. For cycloheximide chase experiments, cells were grown to mid-log phase in SCD-Ura medium, 15 ml aliquots were collected as $t = 0$ samples, 50 µg/ml cycloheximide (Sigma) were added to the remainder of the cultures to block protein synthesis and further aliquots were collected after 5, 10 and 20 min. Cells were lysed as described above and equal fractions of each lysate, corresponding to approximately 40 µg total protein, were analyzed by western blotting.

## Flow cytometry

To measure SHRED, yeast expressing the reporter protein Rtn1Pho8*-GFP and cytosolic BFP were grown to mid-log phase in 1 ml SCD or SCD-Ura medium in 96 deep-well plates at 30 °C. Cultures were diluted to $OD_{600} = 0.05$ and were left untreated or treated with 2 µg/ml tunicamycin (Merck) for 5 h. Fluorescence was measured with a FACS Canto flow cytometer equipped with a high-throughput sampler (BD Biosciences). To determine reporter levels, GFP fluorescence was first corrected for autofluorescence by subtracting the fluorescence of identically treated control cells not expressing GFP. Corrected GFP fluorescence was divided by BFP fluorescence to normalize for cellular translation capacity. To determine the effect of stress treatment, GFP/BFP ratios of treated cells were divided by those of corresponding untreated cells. The resulting reporter levels were expressed in per cent of those at $t = 0$. Roq1 variants lacking the first 21 amino acid residues were expressed as N-terminal ubiquitin fusions, which, after processing by endogenous ubiquitin proteases, yielded Roq1 cleavage fragments starting with R22.

To measure the stability of N-degron substrates, cells were grown in 96 deep-well plates as described above and left untreated or treated with 800 nM β-estradiol (Sigma) for 6 h. Estradiol activates the Gal4-ER-Msn2 transcription factor regulating expression of Roq1 variants placed under the control of the *GAL* promoter (Pincus et al, 2014). mCherry and sfGFP fluorescence were determined, corrected for autofluorescence, and the fold change of the mCherry/sfGFP ratio upon estradiol treatment was calculated. Data were plotted on a $\log_2$ scale so that the same fold increase or fold decrease resulted in the same positive or negative numerical change.

## Roq1 mutagenesis screen

Roq1-HA(74) was amplified from pRS416-$P_{GPD}$-Roq1-HA(74) (Szoradi et al, 2018) by error-prone PCR. Reactions consisted of 1 ng/μl plasmid DNA, 0.5 μM primer GPDprom-165, 0.5 μM primer CYCterm-66, 200 μM dATP, 200 μM dGTP, 1 mM dTTP, 1 mM dCTP, 1.5 mM $MgCl_2$, 0.25 mM $MnCl_2$ and 0.04 U/μl Taq polymerase (Sigma, D1806) in PCR buffer without $MgCl_2$ (Sigma, D4545). Sequencing of PCR products showed that these conditions resulted in about 0.4 mutations per 600-bp amplicon. PCR products were then introduced into yeast by gap repair cloning. Roq1 was excised from pRS416-$P_{GPD}$-Roq1-HA(74) by restriction digest with SpeI, XhoI, and BseRI, and the plasmid backbone was used, together with the products of the error-prone PCR, for co-transformation of the SSY835 reporter strain. Transformants were plated onto SCD-URA plates and about 20,000 colonies were visually screened for impaired degradation of the Rtn1Pho8*-GFP reporter as judged by retention of GFP fluorescence. Candidates were validated with the flow cytometry-based SHRED assay described below. To exclude mutations that destabilized the Roq1 protein, the remaining candidates were tested for Roq1 expression by western blotting. Plasmids from candidates with detectable Roq1 expression were sequenced, which revealed 14 unique point mutations in the Roq1 coding sequence. To confirm that the defects in reporter degradation were caused by mutations in plasmid-encoded Roq1 rather than chromosomal mutations, the point mutations were introduced into pRS416-$P_{GPD}$-Roq1(22-104)-HA(74), the resulting plasmids were introduced into SSY835 and SHRED activity was measured by flow cytometry.

## Immunoprecipitation

Twenty ODs of yeast cells in mid-log phase were lysed by bead beating in IP buffer (25 mM HEPES pH 7.5, 100 mM NaCl, 1% Triton X-100, 0.5 mM EDTA, 1 mM PMSF, Roche complete protease inhibitors). Lysates were cleared at $12,000 \times g$ at 4 °C for 5 min. 3% of the lysate were kept to analyze Roq1 in the input, 15% were kept to analyze Ubr1 in the input and 50% were used to precipitate Roq1 with 30 μl anti-HA-agarose beads (Sigma) at 4 °C for 30 min. Beads were washed three times with cold lysis buffer and bound protein was eluted with SDS-PAGE sample buffer at 65 °C for 5 min. For the analysis of Ubr1 in the input sample, Ubr1 was first immunoprecipitated with 20 μl anti-FLAG agarose beads (clone M2, Sigma). This step reduced variability between samples, which occurred when Ubr1 levels were analyzed directly by western blotting of whole-cell lysates.

## In vitro pulldown

To assess Pho8/Pho8*-Ubr1 interaction, 0.25 μM Pho8-MBP/Pho8*-MBP, 0.25 μM FLAG-Ubr1 and, where indicated, 2.5 μM Roq1(22-60)-HA in 30 μl pulldown buffer (50 mM HEPES pH 7.5, 150 mM NaCl, 10 mM $MgCl_2$, 0.66 μg/μl BSA) were incubated at 30 °C for 90 min. 12 μl amylose resin (NEB) equilibrated in pulldown buffer was added, and samples were incubated with rotation at 4 °C for 2 h. Beads were washed for 3 ×5 min with 100 μl wash buffer (50 mM HEPES pH 7.5, 150 mM NaCl, 10 mM $MgCl_2$), protein was eluted with 2 ×10 μl elution buffer (50 mM HEPES pH 7.5, 150 mM NaCl, 10 mM $MgCl_2$, 10 mM maltose) at 4 °C for 15 min and eluates were analyzed by western blotting.

## Data availability

No deposited datasets are associated with this publication.

The source data of this paper are collected in the following database record: biostudies:S-SCDT-10_1038-S44318-025-00375-7.

## Peer review information

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

## Acknowledgements

The authors are indebted to Chi-Ting Ho, Ilia Kats, Matthias Mayer, Frauke Melchior, and Carola Sparn for reagents, protocols, and advice. The authors thank Claudio Joazeiro, Matthias Mayer, Dimitris Papagiannidis, Anne Schlaitz and all Schookees for comments on the manuscript. Flow cytometry measurements were done at the Flow Cytometry & FACS Core Facility at the Center for Molecular Biology of Heidelberg University (ZMBH). This work was supported by grant SCHU 2364/2-1 from the German Research Foundation (DFG) to SS and a fellowship from the Boehringer Ingelheim Fonds to NP. The authors gratefully acknowledge the data storage service SDS@hd supported by the Ministry of Science, Research and the Arts Baden-Württemberg (MWK) and the DFG through grant INST 35/1503-1 FUGG. For the publication fee the authors acknowledge financial support by Heidelberg University.

## Author contributions

**Niklas Peters**: Conceptualization; Investigation; Writing—original draft; Writing—review and editing. **Sibylle Kanngießer**: Conceptualization; Investigation; Writing—original draft; Writing—review and editing. **Oliver Pajonk**: Investigation; Writing—review and editing. **Rafael Salazar Claros**: Investigation; Writing—review and editing. **Petra Hubbe**: Investigation; Writing—review and editing. **Axel Mogk**: Resources; Writing—review and editing. **Sebastian Schuck**: Conceptualization; Supervision; Funding acquisition; Writing—original draft; Writing—review and editing.

Source data underlying figure panels in this paper may have individual authorship assigned. Where available, figure panel/source data authorship is listed in the following database record: biostudies:S-SCDT-10_1038-S44318-025-00375-7.

## Funding

## Disclosure and competing interests statement

The authors declare no competing interests.

# Expanded View Figures

**Figure EV1.  In vitro reconstitution of Ubr1 regulation by Roq1.**

(A) Western blot of HA tag from ubiquitination assays with Roq1(22-104)-HA and Roq1(22-60)-HA in the absence of a designated Ubr1 substrate. Ubiquitination reactions were stopped after 30 min with buffer containing dithiothreitol and, where indicated, treated with NaOH to hydrolyze ester bonds between Roq1 and ubiquitin. Ubr1 catalyzes the formation of NaOH-sensitive conjugates of Roq1(22-104) and ubiquitin, which is almost completely lost by shortening Roq1 to Roq1(22-60). Note that the bands of unmodified Roq1(22-60) are weaker than those of unmodified Roq1(22-104) even though equal molar amounts of the two proteins were used for the assays. The reason for this difference is that the small Roq1(22-60) is only weakly bound by the nitrocellulose membrane used for western blotting so that its amounts are lower than they should be. Less ubiquitinated Roq1 is generated from Roq1(22-60) than from Roq1(22-104). WT, wild-type. (B) Cellular levels of the SHRED reporter Rtn1Pho8*-GFP after tunicamycin treatment for 5 h relative to levels in untreated cells, as measured by flow cytometry. The *roq1* mutant cells contained an empty plasmid (no Roq1), a plasmid encoding wild-type Roq1 (WT Roq1), ubiquitin-Roq1(22-104) or ubiquitin-Roq1(22-60). The ubiquitin fusions are processed by cells to yield Roq1(22-104) or Roq1(22-60) starting with R22. Bars are the mean ± s.e.m.; $n = 3$ biological replicates. (C) Western blot of Pho8 from solubility assays of Pho8*. Ubiquitination assays including Roq1(22-60) were carried out for 0 or 90 min and soluble and insoluble Pho8* were separated by centrifugation. T = total; S = supernatant; P = pellet. (D) Western blot of Pho8 from Pho8* ubiquitination assays with and without Ubr1 and Roq1(22-60). No Pho8* ubiquitination occurs in the absence of Ubr1. (E). Western blot of Pho8 from Pho8* ubiquitination assays with and without Roq1(22-60). Ubiquitination reactions were stopped after 90 min and, where indicated, treated with NaOH to hydrolyze ester bonds between Pho8* and ubiquitin. Ubiquitin-Pho8* conjugates were resistant to alkaline hydrolysis, showing that they consisted of amide rather than oxyester bonds. (F) Western blot of Pho8 from Pho8* ubiquitination assays with different concentrations of Roq1(22-60). Roq1(22-60) was omitted or used at molar ratios of 1:1, 2:1, 5:1 and 10:1 relative to Ubr1. (G) Western blot of ALFA tag from untreated and tunicamycin-treated control cells (strain SSY122) that do not express an ALFA-tagged protein or cells with chromosomally ALFA-tagged Ubr1 and Roq1 (strain SSY4598). The levels of Ubr1 and Roq1 are similar in cells exposed to tunicamycin-induced proteotoxic stress. Note that proteolytic cleavage of ALFA-tagged Roq1 is inefficient, likely because of the position of the tag at the extreme C-terminus (Szoradi et al, 2018). Asterisks mark non-specific bands. n.t., no treatment; Tm, tunicamycin. (H) Western blot of Pho8 from Pho8* ubiquitination assays without and with Roq1(22-104) or Roq1(22-60) for the times indicated. Source data are available online for this figure.

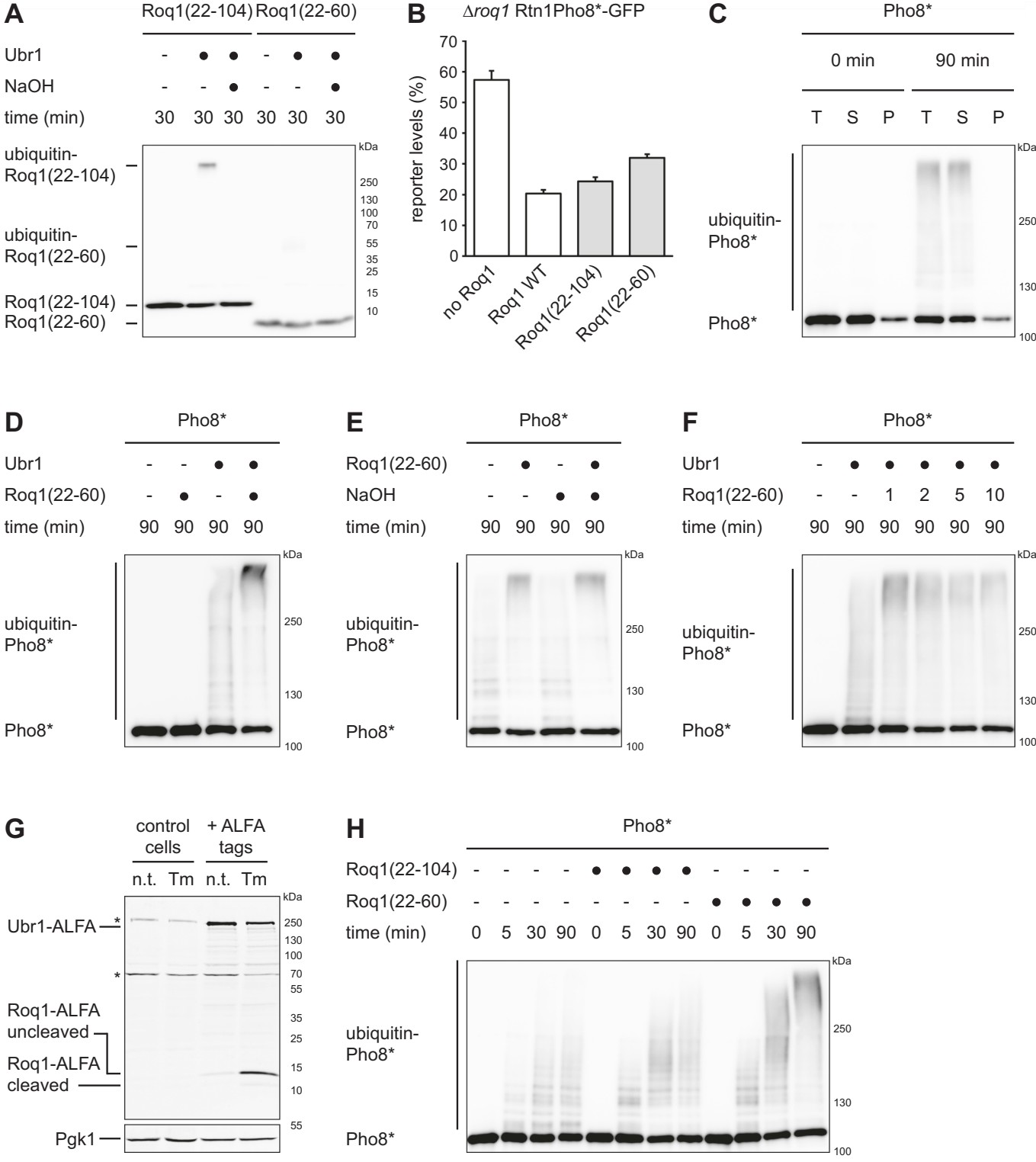

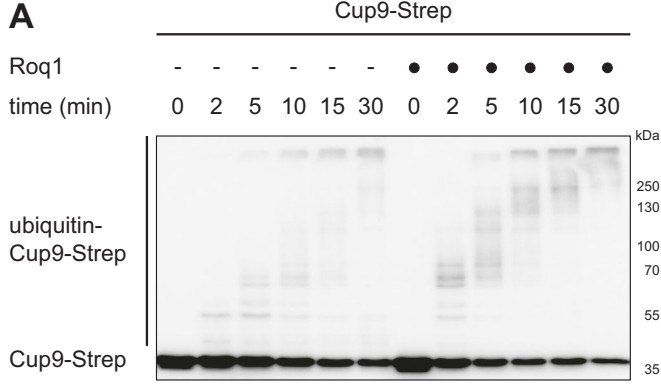

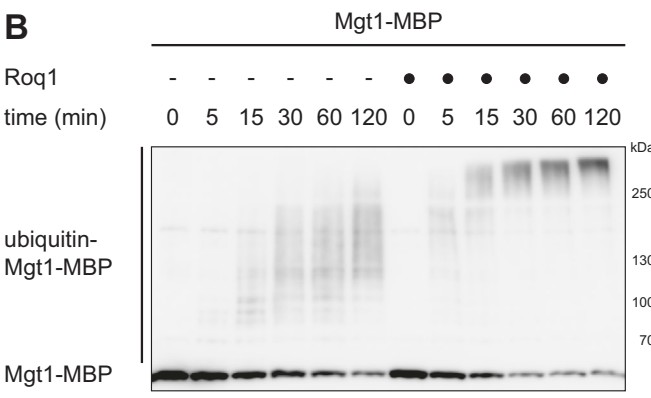

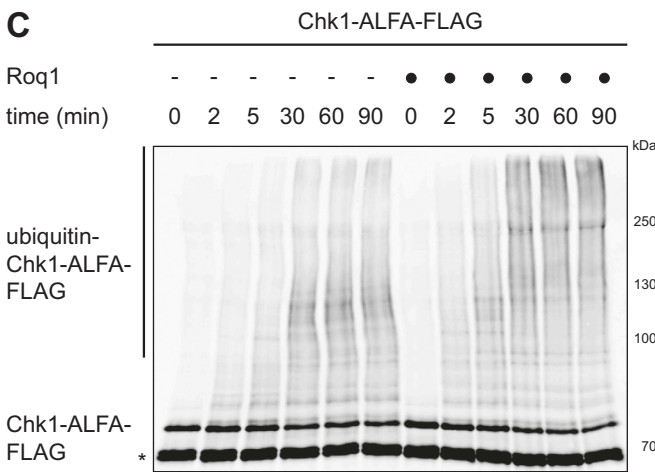

**Figure EV2. Roq1 stimulates ubiquitination of folded Ubr1 substrate proteins with internal degrons.**

(A) Western blot of Strep tag from Cup9-Strep ubiquitination assays without and with Roq1(22-60) for the times indicated. (B) Western blot of maltose-binding protein (MBP) tag from Mgt1-MBP ubiquitination assays without and with Roq1(22-60) for the times indicated. (C) Western blot of FLAG tag from Chk1-ALFA-FLAG ubiquitination assays without and with Roq1(22-60) for the times indicated. The asterisk denotes truncated Chk1 that arose during the expression and purification of Chk1-ALFA-FLAG.Source data are available online for this figure.

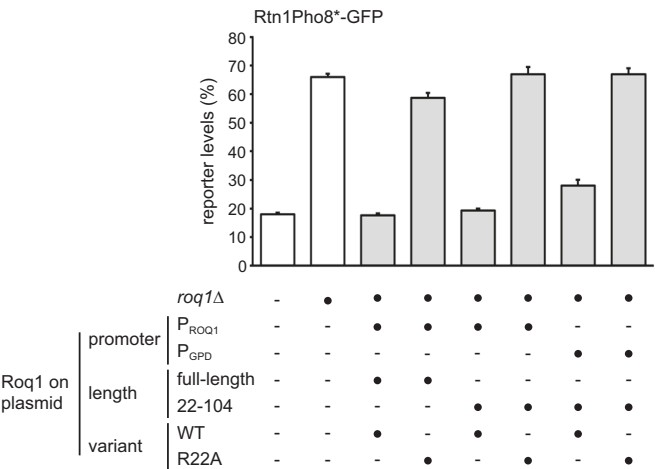

**Figure EV3.  Overexpressed Roq1(R22A) does not activate Ubr1.**

Cellular levels of the SHRED reporter Rtn1Pho8*-GFP after tunicamycin treatment for 5 h relative to levels in untreated cells, as measured by flow cytometry. Cells were wild-type or lacked chromosomal *ROQ1*. The *roq1* mutants contained plasmids encoding Roq1 variants that were expressed under the endogenous *ROQ1* promoter (P$_{ROQ1}$) or the strong *GPD* promoter (P$_{GPD}$), were full-length or lacked the first 21 residues (22-104), and were otherwise wild-type or contained the R22A mutation. Roq1 with the R22A mutation did not support SHRED, even when overexpressed. Full-length Roq1 and ubiquitin-fused Roq1(22-104) are processed in cells by Ynm3 or ubiquitin proteases to yield Roq1(22-104). Ubiquitin-fused Roq1(22-104) was used when Roq1 was expressed under the *GPD* promoter to avoid that cleavage of Roq1 by Ynm3 becomes limiting for the amounts of Roq1(22-104) available for Ubr1 activation. Bars are the mean ± s.e.m.; $n = 3$ biological replicates. Source data are available online for this figure.

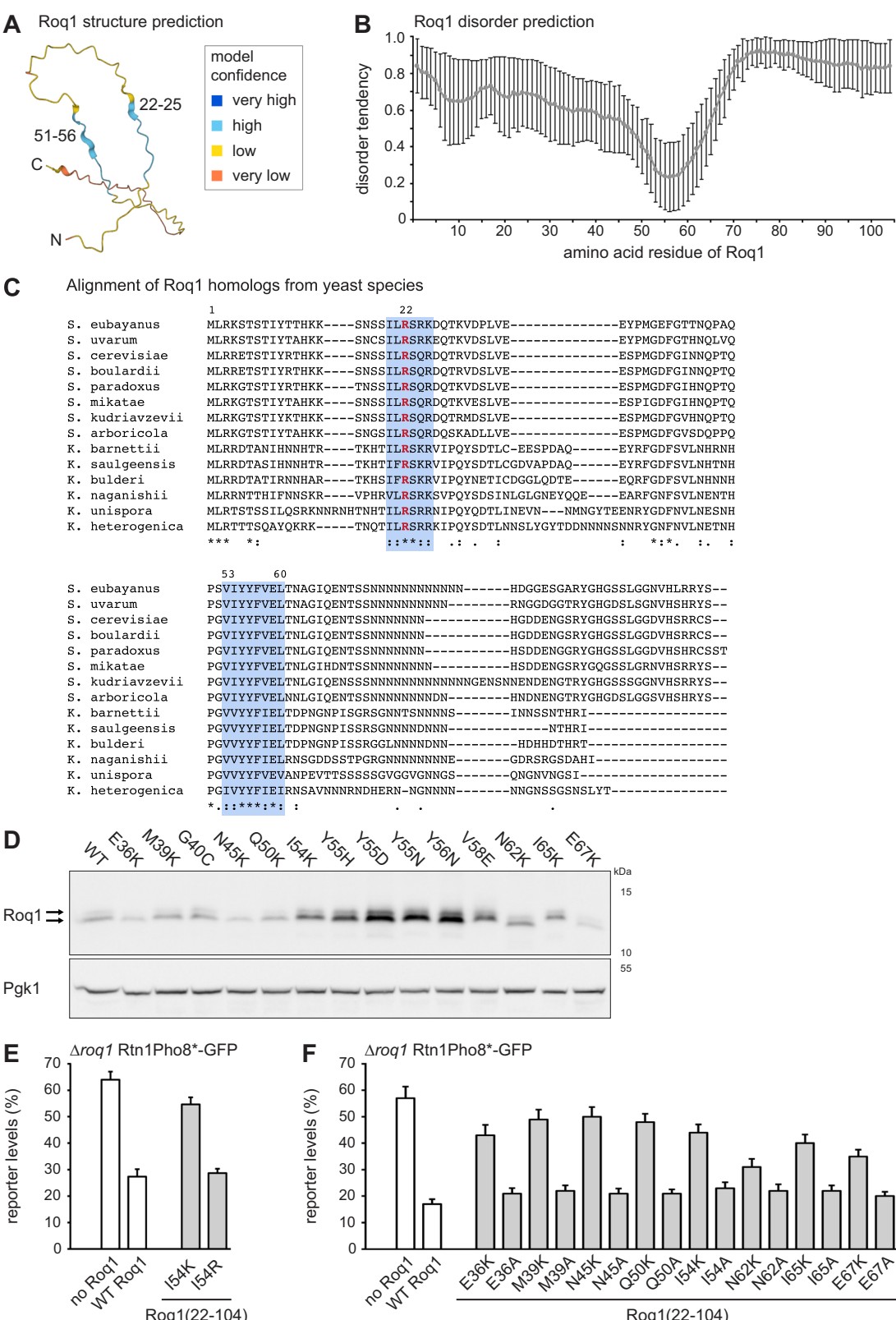

**A** Roq1 structure prediction

**B** Roq1 disorder prediction

**C** Alignment of Roq1 homologs from yeast species

**D**

**E** Δroq1 Rtn1Pho8*-GFP

**F** Δroq1 Rtn1Pho8*-GFP

◀ **Figure EV4.   Roq1 contains a functionally essential hydrophobic motif.**

(A) Roq1 structure prediction by AlphaFold. (B) Roq1 disorder prediction. The plot shows the average disorder tendency across the Roq1(1–104) sequence on a scale of 0 to 1. Data are the mean of the predictions of twelve different algorithms for disorder prediction. Error bars show the standard deviation. (C) Multiple sequence alignment of Roq1 homologs from fourteen yeast species. The numbering of the residues corresponds to the *S. cerevisiae* sequence. * fully conserved residue, : strongly conserved residue. weakly conserved residue. (D) Western blot of HA tag and Pgk1 from *roq1* mutant strains expressing ubiquitin-Roq1(22-104)-HA(74) variants under the control of the strong *GPD* promoter. Roq1(22-104) naturally runs as a double band. The nature of the slower migrating band is unknown. Pgk1 served as a loading control. WT, wild-type. (E, F) Cellular levels of the SHRED reporter Rtn1Pho8*-GFP after tunicamycin treatment for 5 h relative to the levels in untreated cells, as measured by flow cytometry. The *roq1* mutant cells contained an empty plasmid (no Roq1), a plasmid encoding wild-type Roq1 (WT Roq1) or plasmids encoding variants of ubiquitin-Roq1(22-104). Ubiquitin-Roq1 fusions are processed by cells to yield Roq1(22-104). Bars are the mean ± s.e.m.; $n = 3$ biological replicates. Source data are available online for this figure.

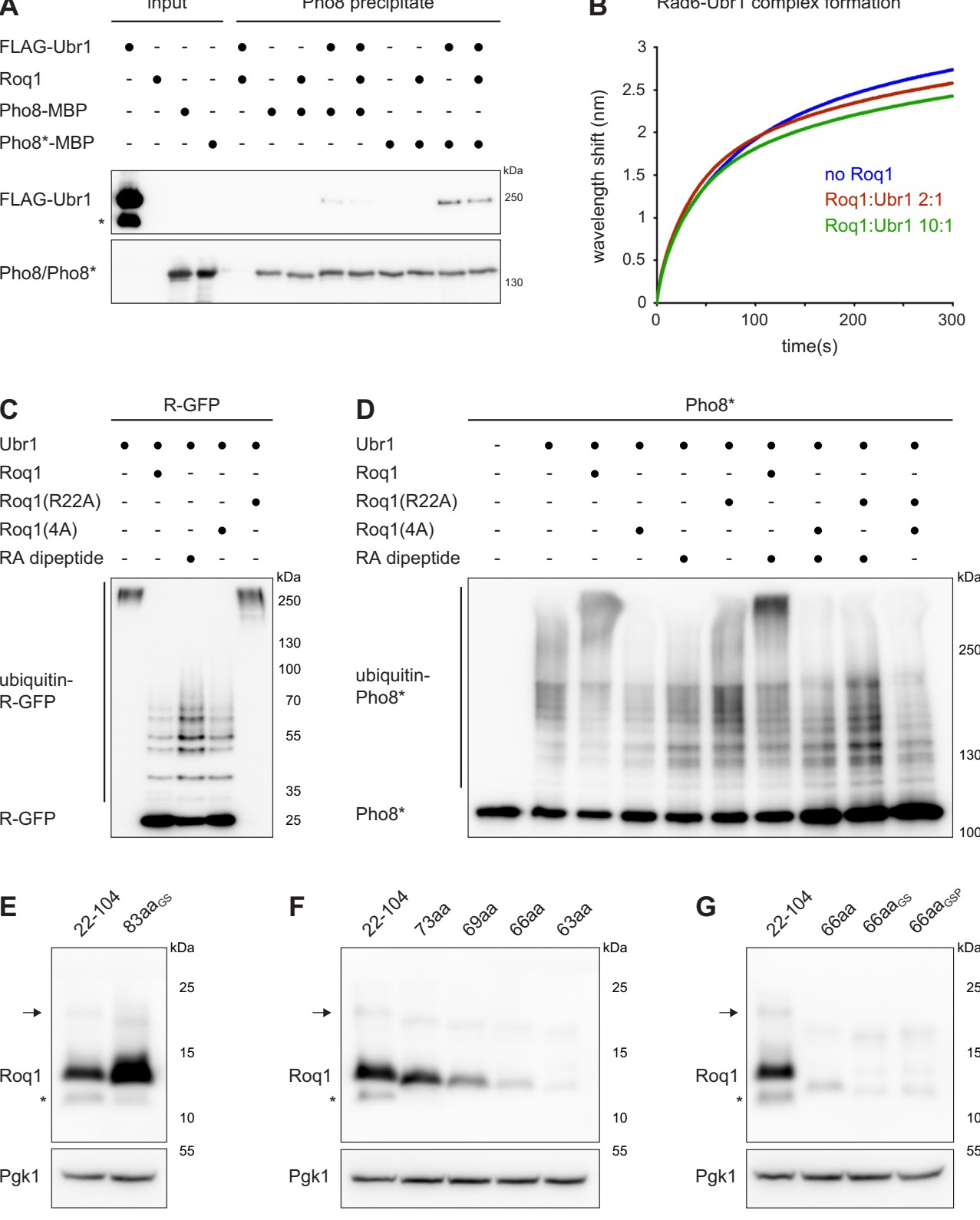

**Figure EV5. Effect of Roq1 on Pho8\* recognition and Rad6 recruitment, effect of separating R22 and the hydrophobic motif different molecules, and expression levels of Roq1 linker variants.**

(A) Western blots of FLAG tag and Pho8 from input and Pho8 precipitate of an in vitro pulldown assays with FLAG-Ubr1, Pho8/Pho8\* fused to maltose-binding protein (MBP), and Roq1(22–60)-HA as indicated. Pho8 or Pho8\*-MBP were precipitated with amylose resin. The asterisk denotes truncated Ubr1 that arose during the expression and purification of FLAG-Ubr1. (B) Biolayer interferometry of Rad6-Ubr1 complex formation with immobilized Rad6 and soluble Ubr1. Complex formation was tested in the absence of Roq1(22–104) and with Roq1(22–104):Ubr1 molar ratios of 2:1 or 10:1. Data are the mean of two independent experiments. (C) Western blot of GFP from R-GFP ubiquitination assays with Roq1(22–60), Roq1(22–60)(R22A), Roq1(22–60)(4A) or an RA dipeptide. Wild-type Roq1, the RA dipeptide and Roq1(4A) are able to bind the Ubr1 type-1 site and suppress ubiquitination of R-GFP, whereas Roq1(R22A) is not. 4A = Y55A,Y56A,F57A,V58A. (D) Western blot of Pho8 from Pho8\* ubiquitination assays with combinations of Roq1(22–60), Roq1(22–60)(R22A), Roq1(22–60)(4A) and an RA dipeptide. Combining a type-1 site binder, Roq1(4A) or the RA dipeptide, with the hydrophobic motif-containing Roq1(R22A) does not reconstitute Roq1 activity. 4A = Y55A,Y56A,F57A,V58A. (E) Western blot of HA tag and Pgk1 from cells expressing Roq1(22–104)-HA(74) or Roq1(83aa$_{GSP}$)-HA(74), in which the sequence between S23 and P51 of Roq1 was replaced with a GSP-based linker. The arrow indicates unprocessed ubiquitin-Roq1, in which the N-terminal ubiquitin has not been removed by ubiquitin proteases. The asterisk marks truncated Roq1. (F) Western blot of HA tag and Pgk1 from cells expressing Roq1(22–104)-HA(74) or Roq1 variants in which the sequence between S23 and P51 was successively shortened. The arrow indicates unprocessed ubiquitin-Roq1, in which the N-terminal ubiquitin has not been removed by ubiquitin proteases. The asterisks marks truncated Roq1. (G) Western blot of HA tag and Pgk1 from cells expressing Roq1(22–104)-HA(74), Roq1(63aa)-HA(74) with a shortened sequence between S23 and P51, Roq1(63aa$_{GS}$)-HA(74) with a GS-based linker replacing the shortened sequence or Roq1(63aa$_{GSP}$)-HA(74) with a GSP-based linker replacing the shortened sequence. The arrow indicates unprocessed ubiquitin-Roq1, in which the N-terminal ubiquitin has not been removed by ubiquitin proteases. The asterisk marks truncated Roq1. Source data are available online for this figure.

