## [Peer Review File · The EMBO Journal]

Reprogramming of the ubiquitin ligase Ubr1 by intrinsically disordered Roq1 through cooperating multifunctional motifs

Niklas Peters, Sibylle Kanngießer, Oliver Pajonk, Rafael Salazar Claros, Petra Hubbe, Axel Mogk, and Sebastian Schuck

Corresponding author(s): Sebastian Schuck (sebastian.schuck@bzh.uni-heidelberg.de)

Review Timeline:

Submission Date:	12th Jul 24
Editorial Decision:	16th Sep 24
Revision Received:	13th Dec 24
Editorial Decision:	20th Jan 25
Revision Received:	22nd Jan 25
Accepted:	24th Jan 25

Editor: Hartmut Vodermaier

Transaction Report:

Dear Dr. Schuck,

Thank you for submitting your manuscript on Ubr1 E3 reprogramming by Roq1, and sincere apologies for the delay in getting back to you with referee reports at this time of the year. We have now received feedback from three experts in ubiquitin ligases and degron recognition, copied below for your information. As you will see, the referees are somewhat divided in their assessment of the overall significance of the presented analyses and results; yet, they all raise a number of substantive and partially overlapping concerns that would currently seem to preclude publication of this work in a broad general journal such as The EMBO Journal.

Since it is not clear if and how these issues could be adequately addressed during a single, regular round of major revision, I would at this point invite you to discuss the reports with your coworkers, and to then send me a tentative point-by-point response, detailing how you would envision clarifying the key concerns of the referees. I would especially be interested in hearing your thoughts regarding testing/ruling out of alternative scenarios, and supporting the functional/physiological significance of the use of Roq1(22-60), which appears to be a recurrent concern in the reports. Based on such a revision proposal, I would be happy to discuss these plans with you directly, in order to find out whether a major revision for The EMBO Journal would seem realistic, or whether a less substantively revised version might alternatively be suitable for one of our sister journals. It would be great if you could get back to me with such a revision plan over the course of next two weeks.

Looking forward to hearing from you,

Best regards,

Hartmut Vodermaier

Referee #1 (Report for Author)

Peters et al. establish an in vitro assay to investigate re-programming of the ubiquitin ligase Ubr1 by the intrinsically-disordered protein Roq1. Using this assay they nicely show that a truncated version of Roq1 (aa 22-60) binds to Ubr1 and facilitates ubiquitylation of unfolded proteins in a process termed SHRED. Ubiquitylation of proteins harboring a bulky hydrophobic residue at the amino-terminus or containing an internal degradation site (degron) is also facilitated while the modification of proteins exposing a positively-charged amino-terminus is negatively affected. As already known, this largely depends on the binding of Roq1 to Ubr1 via an arginine residue at the amino-terminus. By experiments in yeast cells the authors identify a hydrophobic patch in Roq1 that is specifically required for reprogramming Ubr1 for the ubiquitylation of unfolded proteins but

is not involved in modulating the activity of Ubr1 towards substrates containing an internal or an N-degron. This motif seems to constitute a second interaction site to Ubr1. Finally, the authors show that a construct comprising an arginine residue at the amino-terminus connected to a stretch of hydrophobic amino acids by a flexible linker region suffices to induce ubiquitylation of SHRED substrates by Ubr1.

Most of the experiments are well done and include appropriate controls. The authors provide solid evidence that a distinct binding site comprising a hydrophobic region in Roq1 selectively modulates the activity of the Ubr1 ubiquitin ligase towards unfolded proteins. This may be of importance for the design of compounds to regulate such events. Although the mechanism underlying this re-programming process remains enigmatic, the presented study should pave the way for future analyses. Some points should be experimentally addressed by the authors to increase the quality of the work and to better support their arguments. Most importantly, we lack information on the binding of Roq1 to substrates and on the stability and the Ubr1-binding activity of the truncated Roq1 22-60 version in yeast cells. Other than that, I recommend publication of a revised version of the presented work in the "EMBO Journal".

Major points:

Fig. S1G implies that Roq1 22-60 is slightly more active than Roq1 22-104 in the in vitro ubiquitination assay. This may be due to competition for components of the ubiquitination apparatus by Roq1 22-104 and the substrate. Still, the overexpression of Roq1 22-60 in yeast cells is required to rescue a roq1 knockout to re-program Ubr1 for SHRED (Fig. S1B and Fig. S5), whereas lower expression of Roq1 22-104 suffices for this. This may be due to reduced stability/amounts of Roq1 22-60 in yeast or the binding of Roq1 22-60 to Ubr1 in yeast may be compromised. The cellular levels of both proteins should be compared as well as their ability to bind Ubr1, e.g. by using HA-tagged versions of the proteins. These experiments are crucial for understanding the results of this work since the in vitro ubiquitylation assays were exclusively done with the truncated Roq1 protein and these data were directly correlated with yeast experiments involving the full-length construct. In the end, such analysis may also provide information on the carboxy-terminus of Roq1 in regulating its cellular levels.

Given the activity of Roq1_R22A in the ubiquitination assay in Fig. 2E, I wonder if adding excess amounts of a hydrophobic peptide alone can modulate the activity of Ubr1 towards the ubiquitination of unfolded proteins. Related to this: how does the expression/overexpression of the Roq1_R22A variant affect Rtn1Pho8*-GFP reporter levels in yeast cells when compared to the other Roq1 constructs?

Does Roq1 bind unfolded proteins and, if so, is the hydrophobic patch involved in establishing this interaction? Although Roq1 does not seem to facilitate binding of Pho8*-MBP to Flag-Ubr1 (Fig. S4A) it could still assist in the positioning of the substrate.

The crosslinking experiment shown in Fig. 4E should be complemented by a Roq1 variant containing the Bpa crosslinker at positions that are unlikely to directly interact with Ubr1. This

would confirm the specificity of the observed crosslinks as indicators of the binding of Y55 and Y56 in Roq1 to Ubr1.

Minor issues:

The authors use a carboxy-terminal-truncated Roq1 variant (Roq1 22-60) in their in vitro studies to circumvent ubiquitylation of a longer construct (Roq1 22-104) by Ubr1 and "to focus our analysis on the activity of Ubr1 towards designated substrate proteins". In Fig. S1A, they show that Roq1 22-60 is still ubiquitylated. I do not fully agree with the authors that this is to a substantially lesser extent than Roq1 22-104, considering the lower amounts of Roq1 22-60 examined here. The corresponding statement in the Discussion section (line 416) claiming that Roq1 22-60 is "not ubiquitinated" should be changed.

The cellular amounts of Roq1 versions harboring amino acid substitutions in the hydrophobic patch are increased when compared to other Roq1 variants expressed from the same promoter (Fig. 4A, Fig. S3B). The author's claim that this hydrophobic stretch represents a degron to regulate the amounts of the protein could be experimentally addressed, e.g. by cycloheximide decay assays.

The authors should investigate the function of their synthetic constructs comprising an amino-terminal arginine residue and a hydrophobic motif in yeast cells.

Referee #2 (Report for Author)

The goal of this manuscript is to investigate the mechanism by which Roq1 alters the substrate specificity of Ubr1. This manuscript is a follow-up study to the authors' previous paper published in *Molecular Cell* (Vol. 70, p.1025, 2018). In the current study, the authors show that in addition to the N-terminal arginine of Roq1 (R22) that has been revealed previously as important in regulating Ubr1 activity, a short hydrophobic motif (YYFV, 55-58) in the Roq1(22-60) peptide is also crucial. The authors have employed a Roq1(V58E) mutant variant to characterize how this short Roq1 hydrophobic motif regulates Ubr1 activity, revealing that not only does the YYFV motif serve as a binding determinant, but it also regulates Ubr1 activity. Since the experimental systems deployed in this study are highly artificial, it is difficult to judge if the findings indeed reflect what occurs in vivo.

Major points:

1. The authors must use "full-length" Roq1 expressed from the endogenous locus (i.e., no overexpression) in vivo to demonstrate the role of Roq1's hydrophobic motifs in Ubr1 regulation. In this study, the authors have mainly relied on in vitro experiments to decipher Roq1 function in vivo. Nevertheless, as mentioned by the authors in p.14, it is challenging to translate insights from in vitro studies into an understanding of Ubr1 function in vivo. As the authors state, "the outcome of Ubr1 reprogramming in vivo is likely more complex than can be recapitulated easily in reconstitution experiments. We show that Roq1 activates the ubiquitination of both misfolded proteins and Cup9 in vitro. In contrast, Roq1 appears to suppress Cup9 degradation in vivo." (p.14, ln. 414). Moreover, rather than using full-length Roq1 to deduce the function of Roq1, the authors have primarily used an "artificially shortened" version of Roq1, i.e., Roq1(22-60), for their in vitro

reconstitution assays and in vivo reporter assays. The authors argue that Roq1(22-60) is sufficient to represent the function of wild-type Roq1. Nevertheless, Roq1(22-60) did not support Rho8*-GFP degradation when expressed under the Roq1 promoter (Fig. S5), and only partially supported Rho8*-GFP degradation when expressed under the strong GPD promoter (Fig. S1B), indicating that in fact Roq1(22-60) does not adequately represent wild-type Roq1.

Lastly, the authors applied a non-physiologically relevant protein concentration to examine the activity of Roq1(22-60) in their in vitro reconstitution assays, i.e., a "10-fold molar excess of Roq1(22-60) over Ubr1" (p.5, ln. 110). As revealed by the authors, varying the protein concentrations elicited different results: "Roq1(R22A) did not affect Pho8* ubiquitination when used at a 10-fold molar excess over Ubr1 (Figure 2A). However, further raising its concentration slightly enhanced Pho8* ubiquitination (Figure 2E, for example note the upward shift of ubiquitin-Pho8* in the lane with a 40-fold molar excess of Roq1(R22A) compared with the lane with a 10-fold molar excess). Therefore, Roq1(R22A) still has a limited ability to activate Ubr1" (p.7). Therefore, it is misleading to conclude endogenous Roq1 function based on reconstitution experiments that use a 10-fold molar excess of Roq1(22-60) over Ubr1. The authors need to establish what is the "endogenous" protein concentration of Roq1(22-104) as well as the molar ratio of endogenous Roq1 to Ubr1 in wild-type yeast cells after tunicamycin treatment. Based on the authors' own previous work (Mol. Cell 70, p.1025), Roq1 is intrinsically labile and so the endogenous protein abundance of Roq1 is likely to be low.

2. The authors completely rely on artificial reporter models, such as Rho8*-GFP, to measure Ubr1 activity. They need to test degradation of endogenous misfolded protein substrates to ensure that their conclusions remain the same.

3. The authors have concluded that, apart from binding, the hydrophobic motif of Roq1 exerts additional regulatory activity based solely on their characterization of the Roq1(V58E) mutant variant (Fig. 5 and Fig. 6). They argue that Roq1(V58E) binds Ubr1 almost as efficiently as wild-type Ubr1 (Fig. 4A). Nevertheless, despite that pulldown levels of Flag-Ubr1 look similar in Fig. 4A, the level of Roq1(V58E)-HA pulled down is much greater than that of wild-type Roq1-HA, suggesting that Roq1(V58E) binds Ubr1 less efficiently than wild-type Roq1. Therefore, the results from the Roq1(V58E) experiments are insufficient to infer that endogenous Roq1 exerts additional regulatory activity on Ubr1.

4. It is inappropriate to compare the activity of Roq1 mutant variants without considering their protein abundance.

(1) Control Western blots showing the protein abundances of Roq1 variants for Fig. 3F, 3G, 7B, 7D, S1B, S3C, S3D, and S5 are all lacking.

(2) The protein abundances of the Roq1 E36K, N45K and E67K variants are less than those of wild-type Roq1 (Fig. S3D).

Minor point

1. Labeling of the molecular weight markers is lacking for Figs. 1B-1G, 2A-2C, 2E, 5B-5F, 6A-6C, 7D, S1A, S1C-S1G, S2, and S4A.

Referee #3 (Report for Author)

Peters et al. investigate how the Ubr1 ubiquitin ligase is modulated by the Roq1 pseudo-substrate

using in vitro assays and validation of key findings in vivo. The authors find that processed Roq1 uses two SLiMs to interact with and modulate Ubr1: the N-terminal arginine, which likely interacts with the type-1 binding pocket of Ubr1, and a newly identified short hydrophobic motif. The authors argue that both SLiMs are required for Roq1 function, with distinct impacts on Ubr1 activity. The region between the SLiMs only functions as a linker, ensuring cooperative binding of the SLiMs needed for Roq1 function. Overall, the experiments are well performed and documented, and the manuscript is clearly written and a nice read.

In my opinion, the claims regarding the function of the linker region and cooperative binding should be strengthened as outlined below.

1. The authors propose a model where a single Roq1 molecule binds to Ubr1 using two SLiMs, and the linker region serves only to connect the two SLiMs for cooperative binding. According to the authors, one argument for this is that replacing the linker region with short artificial linkers preserves Roq1 function in vitro. However, from the data shown in Fig.7d it looks like while Roq1(22 aa), Roq1(GGS) and Roq1(GSP) are equally active, they are less active compared to the 22-60 variant. Can full length artificial linkers in the context of the 22-60 variant and/or full length Roq1 preserve Roq1 function in vitro and/or in vivo?

2. Based on the authors' model, a shortened linker is expected to reduce Roq1 activity. Less trivially, so should a substantially longer one, which the authors could test in vitro and/or in vivo. Moreover, it would be important to examine the impact of linker mutations and replacements in the context of full length Roq1 in vivo.

3. An alternative possibility is that Roq1 oligomerizes (for example via the linker region) and SLiMs from two Roq1 molecules interact with Ubr1. The gradual loss of Roq1 functionality with shorter linkers (Fig.7b) is consistent with this hypothesis. The authors could distinguish between the two models by testing how combining two Roq1 variants (each mutated in one of the SLiMs, with or without the linker region) affects SHRED and Ubr1 in vitro. And/or by combining Roq1(R22A) with arginine dipeptides in vitro. This could also help strengthen the argument for cooperative binding.

4. On the point of arginine dipeptides, the authors could use them to strengthen the point that Roq1 binds to the type-1 pocket of Ubr1.

Minor points

5. Molecular weight markers are missing on most immunoblots.

6. The authors state that "Roq1(R22A) did not affect Pho8* ubiquitination when used at a 10-fold molar excess over Ubr1 (Figure 2A). However, further raising its concentration slightly enhanced Pho8* ubiquitination (Figure 2E, for example note the upward shift of ubiquitin-Pho8* in the lane with a 40-fold molar excess of Roq1(R22A) compared with the lane with a 10-fold molar excess)." In these and other experiments there appears to be inhibition at amounts above 40-fold excess. A comment on this would be appreciated.

7. "Flexible linkers can span substantial distances. Assuming a Ca-Ca distance in the peptide backbone of 3.8 Å (Chakraborty et al, 2013), a maximally extended 15-residue linker would, theoretically, be more than 5 nm long." Not clear why the authors chose to refer to the artificially shortened linker here, as the natural >30 residue long one would make their point even more striking.
8. "Not shown" data should be shown or not referred to. For example, "Roq1(GGS) and Roq1(GSP) were inactive in vivo, possibly due to low stability (not shown)."
9. From the plasmid table, it is unclear what the sequences of the Roq1 variants in Fig.7A is. What exactly is deleted to shorten the linker region?
10. In the methods section, it is unclear how R-GFP and F-GFP are purified and prepared, (tag, protease, etc).
11. Whenever possible the authors should show some characterization of the purified proteins (e.g. Coomassie gels).

Dear Dr. Vodermaier,

Thank you for inviting us to suggest a plan for a potential revision. We believe that it is possible to adequately address the referees' comments in a regular round of revision and we would aim to resubmit the manuscript until the end of the year.

The referees' comments are incisive and addressing them will strengthen our manuscript. An issue raised by all referees is the physiological relevance of our biochemical in vitro findings. For technical reasons, we mostly used Roq1(22-60), a shortened version of Roq1, for investigating Ubr1 reprogramming in vitro. To test the relevance of our conclusions in vivo, we propose to introduce the mutations we analyzed with Roq1(22-60) into full-length Roq1, express the resulting Roq1 variants in yeast using the endogenous *ROQ1* promoter and determine their SHRED activity. In this way, we would employ Roq1(22-60) strictly as an in vitro tool and systematically validate our conclusions in vivo with full-length Roq1. We think that this strategy is preferable over investigating the artificial Roq1(22-60) in vivo. Besides the issue of physiological relevance, referees 1 and 2 raised questions about the stability and relative abundance of Roq1 variants. We will address these issues in vivo using full-length Roq1 where appropriate. We will attempt to improve detection of Roq1 by Western blotting so that we can use expression under the endogenous *ROQ1* promoter also here. Finally, the referees suggested various additions and controls, some of which are critical and some of which we view as extensions that are dispensable for our main conclusions.

Specifically, we suggest the following plan for new experiments:

1. We will restrict the use of Roq1(22-60) to in vitro experiments and test the relevance of our in vitro results for SHRED activity in yeast using full-length Roq1 variants expressed under the endogenous *ROQ1* promoter (Referee 1, point 1; Referee 2, point 1; Referee 3, points 1 and 2). For Roq1(V58E), we will additionally use N-degron reporters to test our model of a substrate-specific regulatory activity of the hydrophobic motif (Referee 2, point 3).
2. We will assess whether the Roq1 C-terminal region is important for stability (Referee 1, point 2) and determine the relative abundance of the different variants with mutations in the linker and the hydrophobic motif (Referee 1, point 8; Referee 2, point 4).
3. We could test whether the Roq1 hydrophobic motif binds Pho8* (Referee 1, point 4). As explained in the point-by-point response, we would like to consider this as optional.
4. We will include a negative control for the photo-crosslinking (Referee 1, point 5).
5. We will test whether the hydrophobic motif indeed acts as a degron (Referee 1, point 7).
6. We will attempt to determine the relative abundance of Ubr1 and Roq1 in vivo (Referee 2, point 1). This would be a nice but non-essential addition.
7. We will test the alternative model that Roq1 acts as an oligomer (Referee 3, point 3).

We provide detailed information on these and all other referee comments in the tentative point-by-point response below. While the planned experiments will require considerable work, including the generation of quite a few yeast strains, they need almost no technical innovation and we therefore believe that most of these experiments will be straightforward.

We look forward to hearing your thoughts on our revision plan and the proposed timeline.

Best regards,
Sebastian Schuck

Tentative point-by-point response

Referee 1

Peters et al. establish an in vitro assay to investigate re-programming of the ubiquitin ligase Ubr1 by the intrinsically-disordered protein Roq1. Using this assay they nicely show that a truncated version of Roq1 (aa 22-60) binds to Ubr1 and facilitates ubiquitylation of unfolded proteins in a process termed SHRED. Ubiquitylation of proteins harboring a bulky hydrophobic residue at the amino-terminus or containing an internal degradation site (degron) is also facilitated while the modification of proteins exposing a positively-charged amino-terminus is negatively affected. As already known, this largely depends on the binding of Roq1 to Ubr1 via an arginine residue at the amino-terminus. By experiments in yeast cells the authors identify a hydrophobic patch in Roq1 that is specifically required for reprogramming Ubr1 for the ubiquitylation of unfolded proteins but is not involved in modulating the activity of Ubr1 towards substrates containing an internal or an N-degron. This motif seems to constitute a second interaction site to Ubr1. Finally, the authors show that a construct comprising an arginine residue at the amino-terminus connected to a stretch of hydrophobic amino acids by a flexible linker region suffices to induce ubiquitylation of SHRED substrates by Ubr1.

Most of the experiments are well done and include appropriate controls. The authors provide solid evidence that a distinct binding site comprising a hydrophobic region in Roq1 selectively modulates the activity of the Ubr1 ubiquitin ligase towards unfolded proteins. This may be of importance for the design of compounds to regulate such events. Although the mechanism underlying this re-programming process remains enigmatic, the presented study should pave the way for future analyses. Some points should be experimentally addressed by the authors to increase the quality of the work and to better support their arguments. Most importantly, we lack information on the binding of Roq1 to substrates and on the stability and the Ubr1-binding activity of the truncated Roq1 22-60 version in yeast cells. Other than that, I recommend publication of a revised version of the presented work in the "EMBO Journal".

Major points:

1. Fig. S1G implies that Roq1 22-60 is slightly more active than Roq1 22-104 in the in vitro ubiquitination assay. This may be due to competition for components of the ubiquitination apparatus by Roq1 22-104 and the substrate. Still, the overexpression of Roq1 22-60 in yeast cells is required to rescue a roq1 knockout to re-program Ubr1 for SHRED (Fig. S1B and Fig. S5), whereas lower expression of Roq1 22-104 suffices for this. This may be due to reduced stability/amounts of Roq1 22-60 in yeast or the binding of Roq1 22-60 to Ubr1 in yeast may be compromised. The cellular levels of both proteins should be compared as well as their ability to bind Ubr1, e.g. by using HA-tagged versions of the proteins. These experiments are crucial for understanding the results of this work since the in vitro ubiquitylation assays were exclusively done with the truncated Roq1 protein and these data were directly correlated with yeast experiments involving the full-length construct.

This comment raises the valid point that we should determine rigorously whether the in vitro results with truncated Roq1(22-60) apply to the situation in vivo. To do this, we will systematically test the conclusions we obtained with Roq1(22-60) through experiments in yeast using full-length Roq1 expressed under its endogenous promoter. Specifically, we will mutate the Roq1 hydrophobic motif or the linker and measure the activity of the resulting variants using our standard SHRED reporter (also see point 8 below; Referee 2, point 1 and 3; Referee 3, point 1 and 2).

2. In the end, such analysis may also provide information on the carboxy-terminus of Roq1 in regulating its cellular levels.

We agree and will determine if the C-terminal region is indeed important for Roq1 stability.

3. Given the activity of Roq1_R22A in the ubiquitination assay in Fig. 2E, I wonder if adding excess amounts of a hydrophobic peptide alone can modulate the activity of Ubr1 towards the ubiquitination of unfolded proteins. Related to this: how does the expression/overexpression of the Roq1_R22A variant affect Rtn1Pho8*-GFP reporter levels in yeast cells when compared to the other Roq1 constructs?

Adding large amounts of a hydrophobic peptide to the ubiquitination reactions is likely not feasible due to peptide insolubility and stickiness. In fact, the Roq1 hydrophobic motif may have evolved in the context of an intrinsically disordered protein precisely in order for it to be solvent-exposed yet soluble. Thus, Roq1 may be thought of as a hydrophobic peptide kept in solution by surrounding hydrophilic sequences, a view we will mention in the discussion.

We have already tested the in vivo activity of Roq1(R22A) expressed under the *GPD* promoter, which is the strongest promoter available in yeast and seen that Roq1(R22A) is still inactive. We will include these data in the revised manuscript.

4. Does Roq1 bind unfolded proteins and, if so, is the hydrophobic patch involved in establishing this interaction? Although Roq1 does not seem to facilitate binding of Pho8*-MBP to Flag-Ubr1 (Fig. S4A) it could still assist in the positioning of the substrate.

We asked whether Roq1 affects binding of Pho8* to Ubr1 (Figure S4A) but did not test whether Roq1 binds Pho8*. We could modify our pulldown experiments to address this. Detecting some affinity of the Roq1 hydrophobic motif for Pho8* would be unsurprising because the misfolded Pho8* likely exposes hydrophobic residues. Therefore, it would be unclear what a binary hydrophobic Roq1-Pho8* interaction would mean for the Roq1-Ubr1-Pho8* complex. Only the atomic structure of the ternary complex will clarify the issue. However, Roq1 could indeed assist in the positioning the substrate and will mention this scenario in the discussion.

5. The crosslinking experiment shown in Fig. 4E should be complemented by a Roq1 variant containing the Bpa crosslinker at positions that are unlikely to directly interact with Ubr1. This would confirm the specificity of the observed crosslinks as indicators of the binding of Y55 and Y56 in Roq1 to Ubr1.

We will do this control experiment.

Minor issues:

6. The authors use a carboxy-terminal-truncated Roq1 variant (Roq1 22-60) in their in vitro studies to circumvent ubiquitylation of a longer construct (Roq1 22-104) by Ubr1 and "to focus our analysis on the activity of Ubr1 towards designated substrate proteins". In Fig. S1A, they show that Roq1 22-60 is still ubiquitylated. I do not fully agree with the authors that this is to a substantially lesser extent than Roq1 22-104, considering the lower amounts of Roq1 22-60 examined here. The corresponding statement in the Discussion section (line 416) claiming that Roq1 22-60 is "not ubiquitinated" should be changed.

We agree and will phrase our description more precisely. Roq1(22-60) clearly receives fewer ubiquitin moieties than Roq1(22-104), which is what we meant to convey.

7. The cellular amounts of Roq1 versions harboring amino acid substitutions in the hydrophobic patch are increased when compared to other Roq1 variants expressed from the same promotor (Fig. 4A, Fig. S3B). The author's claim that this hydrophobic stretch represents a degron to regulate the amounts of the protein could be experimentally addressed, e.g. by cycloheximide decay assays.

We have done similar cycloheximide chase experiments with Roq1 before (Szoradi et al, 2018, Figure S6) and will clarify this issue.

8. The authors should investigate the function of their synthetic constructs comprising an amino-terminal arginine residue and a hydrophobic motif in yeast cells.

We have seen that synthetic Roq1 constructs consisting of R22-(artificial linker)-(hydrophobic motif) are inactive in vivo, presumably because they are unstable in cells (p 12, line 339 – 340). Whether this interpretation is true and whether the stability of Roq1 depends on the natural linker and/or the C-terminal region will be shown by the vivo experiments described above (see point 1) and below (see Referee 2, points 1 and 3; Referee 3, points 1 and 2).

Referee 2

The goal of this manuscript is to investigate the mechanism by which Roq1 alters the substrate specificity of Ubr1. This manuscript is a follow-up study to the authors' previous paper published in Molecular Cell (Vol. 70, p.1025, 2018). In the current study, the authors show that in addition to the N-terminal arginine of Roq1 (R22) that has been revealed previously as important in regulating Ubr1 activity, a short hydrophobic motif (YYFV, 55-58) in the Roq1(22-60) peptide is also crucial. The authors have employed a Roq1(V58E) mutant variant to characterize how this short Roq1 hydrophobic motif regulates Ubr1 activity, revealing that not only does the YYFV motif serve as a binding determinant, but it also regulates Ubr1 activity. Since the experimental systems deployed in this study are highly artificial, it is difficult to judge if the findings indeed reflect what occurs in vivo.

Major points:

1. The authors must use "full-length" Roq1 expressed from the endogenous locus (i.e., no overexpression) in vivo to demonstrate the role of Roq1's hydrophobic motifs in Ubr1 regulation.

In this study, the authors have mainly relied on in vitro experiments to decipher Roq1 function in vivo. Nevertheless, as mentioned by the authors in p.14, it is challenging to translate insights from in vitro studies into an understanding of Ubr1 function in vivo. As the authors state, "the outcome of Ubr1 reprogramming in vivo is likely more complex than can be recapitulated easily in reconstitution experiments. We show that Roq1 activates the ubiquitination of both misfolded proteins and Cup9 in vitro. In contrast, Roq1 appears to suppress Cup9 degradation in vivo." (p.14, ln. 414).

Moreover, rather than using full-length Roq1 to deduce the function of Roq1, the authors have primarily used an "artificially shortened" version of Roq1, i.e., Roq1(22-60), for their in vitro reconstitution assays and in vivo reporter assays. The authors argue that Roq1(22-60) is sufficient to represent the function of wild-type Roq1. Nevertheless, Roq1(22-60) did not support Rtn1-Pho8-GFP degradation when expressed under the ROQ1 promoter (Fig. S5), and only partially supported Rtn1-Pho8*-GFP degradation when expressed under the strong GPD promoter (Fig. S1B), indicating that in fact Roq1(22-60) does not adequately represent wild-type Roq1.*

Lastly, the authors applied a non-physiologically relevant protein concentration to examine the activity of Roq1(22-60) in their in vitro reconstitution assays, i.e., a "10-fold molar excess of Roq1(22-60) over Ubr1" (p.5, ln. 110). As revealed by the authors, varying the protein concentrations elicited different results: "Roq1(R22A) did not affect Pho8 ubiquitination when used at a 10-fold molar excess over Ubr1 (Figure 2A). However, further raising its concentration slightly enhanced Pho8* ubiquitination (Figure 2E, for example note the upward shift of ubiquitin-Pho8* in the lane with a 40-fold molar excess of Roq1(R22A) compared with the lane with a 10-fold molar excess). Therefore, Roq1(R22A) still has a limited ability to activate Ubr1" (p.7). Therefore, it is misleading to conclude endogenous Roq1 function based on reconstitution experiments that use a 10-fold molar excess of*

Roq1(22-60) over Ubr1. The authors need to establish what is the "endogenous" protein concentration of Roq1(22-104) as well as the molar ratio of endogenous Roq1 to Ubr1 in wild-type yeast cells after tunicamycin treatment. Based on the authors' own previous work (Mol. Cell 70, p.1025), Roq1 is intrinsically labile and so the endogenous protein abundance of Roq1 is likely to be low.

We agree that we should rigorously test the in vivo relevance of our in vitro findings obtained with Roq1(22-60). For this, we will use yeast expressing full-length Roq1 variants under the control of the endogenous *ROQ1* promoter, including variants with mutations in the hydrophobic motif (also see Referee 1, point 1).

Regarding the relative abundance of Roq1 and Ubr1, we show that, in vitro, a molar ratio of Roq1(22-60) to Ubr1 of 1:1 is nearly sufficient for full Ubr1 activation (Figure S1F). We then used molar ratios of 10:1 or higher to make the in vitro ubiquitination assays as robust as possible. Determining the relative levels of Roq1 and Ubr1 in yeast will be challenging given the low abundance of both proteins. In addition, any finding will come with the caveat that Roq1 and Ubr1 likely have different subcellular distributions. Ubr1 is primarily nuclear (Prasad et al, JCB 2018), whereas the tiny Roq1 may freely pass the nuclear pore complex and equilibrate between cytosol and nucleus. It is therefore unclear what conclusions we would be able to draw from these experiments. Nevertheless, we will give this a try.

2. The authors completely rely on artificial reporter models, such as Rtn1-Pho8*-GFP, to measure Ubr1 activity. They need to test degradation of endogenous misfolded protein substrates to ensure that their conclusions remain the same.

We do not have such substrates at our disposal. We previously identified endogenous SHRED substrates, i.e. proteins that become Roq1-dependent Ubr1 substrates during stress (Szoradi et al, 2018). The extent of their degradation is minor (likely because only a small fraction of an endogenous substrate misfolds even during stress), their degradation does not exclusively depend on SHRED (likely because of redundancy between protein quality control pathways), or both. Therefore, these substrates are prohibitively inefficient SHRED reporters. In contrast, Rtn1-Pho8*-GFP is extensively degraded in a stress-, Roq1- and Ubr1-dependent manner, making it an excellent tool to investigate the mechanism of the SHRED pathway. Defining the physiological substrate spectrum of SHRED is an issue outside the scope of this study.

3. The authors have concluded that, apart from binding, the hydrophobic motif of Roq1 exerts additional regulatory activity based solely on their characterization of the Roq1(V58E) mutant variant (Fig. 5 and Fig. 6). They argue that Roq1(V58E) binds Ubr1 almost as efficiently as wild-type Ubr1 (Fig. 4A). Nevertheless, despite that pulldown levels of Flag-Ubr1 look similar in Fig. 4A, the level of Roq1(V58E)-HA pulled down is much greater than that of wild-type Roq1-HA, suggesting that Roq1(V58E) binds Ubr1 less efficiently than wild-type Roq1. Therefore, the results from the Roq1(V58E) experiments are insufficient to infer that endogenous Roq1 exerts additional regulatory activity on Ubr1.

This is an important point. The conclusion that the hydrophobic motif has specific regulatory activity primarily rests on in vitro findings: the Roq1(V58E) retains substantial binding to Ubr1, has the same effect as wild-type Roq1 on N-degron substrates but is almost inactive towards

misfolded substrates (Figure 5). In vivo, Roq1(V58E) also retains substantial Ubr1 binding and is inactive towards our standard SHRED reporter (Figures 3D and 4A). To strengthen the in vivo experiments, we will test the activity of Roq1(V58E) towards reporters of the N-degron pathway. We have used these N-degron reporters before (Szoradi et al, 2018).

4. It is inappropriate to compare the activity of Roq1 mutant variants without considering their protein abundance.

(1) Control Western blots showing the protein abundances of Roq1 variants for Fig. 3F, 3G, 7B, 7D, S1B, S3C, S3D, and S5 are all lacking.

(2) The protein abundances of the Roq1 E36K, N45K and E67K variants are less than those of wild-type Roq1 (Fig. S3D).

We did the in vivo experiments to assess whether different Roq1 variants still possessed SHRED activity. A defect would be trivial if a mutant variant had a lower abundance than native Roq1. However, starting with the screen (Figures 3D, S3C and S3D), we expressed Roq1 variants under the strong *GPD* promoter and, as a result, all mutant variants had higher levels than native Roq1 expressed under the *ROQ1* promoter. We therefore conclude that all mutant variants have genuine SHRED defects (Figure S3B and page 7, line 188 – 195).

The referee is right, though, regarding comparisons between different Roq1 variants because even a lowly abundant mutant with substantial residual activity could appear more strongly impaired than an abundant mutant with little residual activity. For instance, we agree that the residual activities of the linker and truncation mutants in Figures 7B, S1B and S5 are difficult to assess. We will clarify this issue with Western blots of tagged variants, expressed under the endogenous *ROQ1* promoter if possible (also see Referee 1, points 2 and 8).

Minor point

1. Labeling of the molecular weight markers is lacking for Figs. 1B-1G, 2A-2C, 2E, 5B-5F, 6A-6C, 7D, S1A, S1C-S1G, S2, and S4A.

Will be added.

Referee 3

Peters et al. investigate how the Ubr1 ubiquitin ligase is modulated by the Roq1 pseudo-substrate using in vitro assays and validation of key findings in vivo. The authors find that processed Roq1 uses two SLiMs to interact with and modulate Ubr1: the N-terminal arginine, which likely interacts with the type-1 binding pocket of Ubr1, and a newly identified short hydrophobic motif. The authors argue that both SLiMs are required for Roq1 function, with distinct impacts on Ubr1 activity. The region between the SLiMs only functions as a linker, ensuring cooperative binding of the SLiMs needed for Roq1 function. Overall, the experiments are well performed and documented, and the manuscript is clearly written and a nice read.

In my opinion, the claims regarding the function of the linker region and cooperative binding should be strengthened as outlined below.

Major points:

1. The authors propose a model where a single Roq1 molecule binds to Ubr1 using two SLiMs, and the linker region serves only to connect the two SLiMs for cooperative binding. According to the authors, one argument for this is that replacing the linker region with short artificial linkers preserves Roq1 function in vitro. However, from the data shown in Fig.7d it looks like while Roq1(22 aa), Roq1(GGS) and Roq1(GSP) are equally active, they are less active compared to the 22-60 variant. Can full length artificial linkers in the context of the 22-60 variant and/or full length Roq1 preserve Roq1 function in vitro and/or in vivo?

2. Based on the authors' model, a shortened linker is expected to reduce Roq1 activity. Less trivially, so should a substantially longer one, which the authors could test in vitro and/or in vivo. Moreover, it would be important to examine the impact of linker mutations and replacements in the context of full length Roq1 in vivo.

We will address both points by experiments in yeast with Roq1 variants with artificial linkers of different lengths. We will do this in the context of otherwise full-length Roq1 expressed under its endogenous promoter (also see Referee 1, point 1; Referee 2, point 1 and 3).

3. An alternative possibility is that Roq1 oligomerizes (for example via the linker region) and SLiMs from two Roq1 molecules interact with Ubr1. The gradual loss of Roq1 functionality with shorter linkers (Fig.7b) is consistent with this hypothesis. The authors could distinguish between the two models by testing how combining two Roq1 variants (each mutated in one of the SLiMs, with or without the linker region) affects SHRED and Ubr1 in vitro. And/or by combining Roq1(R22A) with arginine dipeptides in vitro. This could also help strengthen the argument for cooperative binding.

We already combined Roq1(R22A) with the arginine dipeptide. This combination could not rescue Roq1 activity. We will additionally do the suggested experiment of combining different Roq1 variants, which is straightforward.

4. On the point of arginine dipeptides, the authors could use them to strengthen the point that Roq1 binds to the type-1 pocket of Ubr1.

The referee suggests a competition experiment to ask whether arginine dipeptides, which block the type-1 site, prevent Roq1 from binding Ubr1. We did this experiment but arginine dipeptides could not disrupt the Roq1-Ubr1 interaction. This observation is consistent with the high Roq1-Ubr1 affinity resulting from the cooperation of the two Roq1 SLiMs but constitutes negative data. However, we have done a different competition experiment that supports Roq1 binding to the Ubr1 type-1 site with positive data. Specifically, we found that arginine dipeptides interfere with the ubiquitination of R-GFP, a Ubr1 substrate that is recognized through the type-1 site. Roq1 had the same effect. By contrast, Roq1(R22A), which lacks the arginine needed for type-1 site binding, did not affect R-GFP ubiquitination. Considering our previous in vivo evidence for Roq1 binding to the type-1 site (Szoradi et al, 2018), we are unsure whether these data should be included. We will present them in our rebuttal letter and ask the referee for advice.

Minor points

5. Molecular weight markers are missing on most immunoblots.

Will be added.

6. The authors state that "Roq1(R22A) did not affect Pho8* ubiquitination when used at a 10-fold molar excess over Ubr1 (Figure 2A). However, further raising its concentration slightly enhanced Pho8* ubiquitination (Figure 2E, for example note the upward shift of ubiquitin-Pho8* in the lane with a 40-fold molar excess of Roq1(R22A) compared with the lane with a 10-fold molar excess)." In these and other experiments there appears to be inhibition at amounts above 40-fold excess. A comment on this would be appreciated.

We will add such a comment in the results section.

7. "Flexible linkers can span substantial distances. Assuming a C α -C α distance in the peptide backbone of 3.8 Å (Chakraborty et al, 2013), a maximally extended 15-residue linker would, theoretically, be more than 5 nm long." Not clear why the authors chose to refer to the artificially shortened linker here, as the natural >30 residue long one would make their point even more striking.

Good suggestion, which we will incorporate in the discussion.

8. "Not shown" data should be shown or not referred to. For example, "Roq1(GGS) and Roq1(GSP) were inactive in vivo, possibly due to low stability (not shown)."

These 'data not shown' will be superseded by data from the in vivo experiments with full-length Roq1 variants (see point 1 and 2, above).

9. From the plasmid table, it is unclear what the sequences of the Roq1 variants in Fig.7A is. What exactly is deleted to shorten the linker region?

We will include this Information in Figure 7A.

10. In the methods section, it is unclear how R-GFP and F-GFP are purified and prepared, (tag, protease, etc).

We will make this clearer in the methods section.

11. Whenever possible the authors should show some characterization of the purified proteins (e.g. Coomassie gels).

We have these gels and will supply them as a new supplementary figure.

Dr. Sebastian Schuck
Heidelberg University
Biochemistry Center
Im Neuenheimer Feld 328
Baden-Württemberg 69120
Germany

16th Sep 2024

Re: EMBOJ-2024-118461
Reprogramming of the ubiquitin ligase Ubr1 by intrinsically disordered Roq1 through cooperating multifunctional motifs

Dear Dr. Schuck,

Thank you for your careful revision plan and tentative point-by-point responses to the reviews on your recent EMBO journal submission. I have now had a chance to go through them, and I was pleased to see that you seem to be in a good position to address the key issues of the referees. In particular, the proposed incorporation of mutants identified with the help of Roq1 (22-60) into full-length Roq1 for cellular testing should go a long way in alleviating important concerns. I am therefore happy to invite you to prepare and resubmit a new version of the study, revised along the lines proposed in your letter. In this regard, I would agree that general point 3 (page 1) would not be essential within the scope of the revision, nor would be point 6 in case the experiments should not be conclusive.

Please keep in mind that it is our policy to allow only a single round of (major) revision, and do update me should there be any unexpected problems with the revisions, or should you require an extension beyond the default 3-months deadline. As always, competing manuscript published during the course of this revision will not affect our final decision on your study. Finally, please note the detailed information and guidelines on how to prepare a revision below (and in our online Guide to Authors) - closely adhering to them shall greatly facilitate the editorial process at the time of resubmission.

Thank you again for the opportunity to consider this work, and I look forward to receiving your revision in due time.

Yours sincerely,

Hartmut Vodermaier

4) Each main and each Expanded View (EV) figure should be uploaded as individual production-quality files (preferably in .eps,

.tif, .jpg formats). For suggestions on figure preparation/layout, please refer to our Figure Preparation Guidelines: <http://bit.ly/EMBOPressFigurePreparationGuideline>

9) To facilitate reproducibility and cross-laboratory adoption of methodologies, please structure the Materials & Methods section as outlined in our guide to authors, including a completed Reagents and Tools Table that can be downloaded from our author guidelines as well (<https://www.embopress.org/page/journal/14602075/authorguide#structuredmethods>).

10) Digital image enhancement is acceptable practice, as long as it accurately represents the original data and conforms to community standards. If a figure has been subjected to significant electronic manipulation, this must be clearly noted in the figure legend and/or the 'Materials and Methods' section. The editors reserve the right to request original versions of figures and the original images that were used to assemble the figure. Finally, we generally encourage uploading of numerical as well as gel/blot image source data; for details see: embopress.org/page/journal/14602075/authorguide#sourcedata

At EMBO Press, we ask authors to provide source data for the main manuscript figures. Our source data coordinator will contact you to discuss which figure panels we would need source data for and will also provide you with helpful tips on how to upload and organize the files.

In the interest of ensuring the conceptual advance provided by the work, we recommend submitting a revision within 3 months (15th Dec 2024). Please discuss the revision progress ahead of this time with the editor if you require more time to complete the revisions. Use the link below to submit your revision:

Link Not Available

Referee #1:

Peters et al. establish an in vitro assay to investigate re-programming of the ubiquitin ligase Ubr1 by the intrinsically-disordered protein Roq1. Using this assay they nicely show that a truncated version of Roq1 (aa 22-60) binds to Ubr1 and facilitates ubiquitylation of unfolded proteins in a process termed SHRED. Ubiquitylation of proteins harboring a bulky hydrophobic residue at the amino-terminus or containing an internal degradation site (degron) is also facilitated while the modification of proteins exposing a positively-charged amino-terminus is negatively affected. As already known, this largely depends on the binding of Roq1 to Ubr1 via an arginine residue at the amino-terminus. By experiments in yeast cells the authors identify a hydrophobic patch in Roq1 that is specifically required for reprogramming Ubr1 for the ubiquitylation of unfolded proteins but is not involved in modulating the activity of Ubr1 towards substrates containing an internal or an N-degron. This motif seems to constitute a second interaction site to Ubr1. Finally, the authors show that a construct comprising an arginine residue at the amino-terminus connected to a stretch of hydrophobic amino acids by a flexible linker region suffices to induce ubiquitylation of SHRED substrates by Ubr1.

Most of the experiments are well done and include appropriate controls. The authors provide solid evidence that a distinct binding site comprising a hydrophobic region in Roq1 selectively modulates the activity of the Ubr1 ubiquitin ligase towards unfolded proteins. This may be of importance for the design of compounds to regulate such events. Although the mechanism underlying this re-programming process remains enigmatic, the presented study should pave the way for future analyses. Some

points should be experimentally addressed by the authors to increase the quality of the work and to better support their arguments. Most importantly, we lack information on the binding of Roq1 to substrates and on the stability and the Ubr1-binding activity of the truncated Roq1 22-60 version in yeast cells. Other than that, I recommend publication of a revised version of the presented work in the "EMBO Journal".

Major points:

Fig. S1G implies that Roq1 22-60 is slightly more active than Roq1 22-104 in the in vitro ubiquitination assay. This may be due to competition for components of the ubiquitination apparatus by Roq1 22-104 and the substrate. Still, the overexpression of Roq1 22-60 in yeast cells is required to rescue a roq1 knockout to re-program Ubr1 for SHRED (Fig. S1B and Fig. S5), whereas lower expression of Roq1 22-104 suffices for this. This may be due to reduced stability/amounts of Roq1 22-60 in yeast or the binding of Roq1 22-60 to Ubr1 in yeast may be compromised. The cellular levels of both proteins should be compared as well as their ability to bind Ubr1, e.g. by using HA-tagged versions of the proteins. These experiments are crucial for understanding the results of this work since the in vitro ubiquitylation assays were exclusively done with the truncated Roq1 protein and these data were directly correlated with yeast experiments involving the full-length construct. In the end, such analysis may also provide information on the carboxy-terminus of Roq1 in regulating its cellular levels.

Given the activity of Roq1_R22A in the ubiquitination assay in Fig. 2E, I wonder if adding excess amounts of a hydrophobic peptide alone can modulate the activity of Ubr1 towards the ubiquitination of unfolded proteins. Related to this: how does the expression/overexpression of the Roq1_R22A variant affect Rtn1Pho8*-GFP reporter levels in yeast cells when compared to the other Roq1 constructs?

Does Roq1 bind unfolded proteins and, if so, is the hydrophobic patch involved in establishing this interaction? Although Roq1 does not seem to facilitate binding of Pho8*-MBP to Flag-Ubr1 (Fig. S4A) it could still assist in the positioning of the substrate.

The crosslinking experiment shown in Fig. 4E should be complemented by a Roq1 variant containing the Bpa crosslinker at positions that are unlikely to directly interact with Ubr1. This would confirm the specificity of the observed crosslinks as indicators of the binding of Y55 and Y56 in Roq1 to Ubr1.

Minor issues:

The authors use a carboxy-terminal-truncated Roq1 variant (Roq1 22-60) in their in vitro studies to circumvent ubiquitylation of a longer construct (Roq1 22-104) by Ubr1 and "to focus our analysis on the activity of Ubr1 towards designated substrate proteins". In Fig. S1A, they show that Roq1 22-60 is still ubiquitylated. I do not fully agree with the authors that this is to a substantially lesser extent than Roq1 22-104, considering the lower amounts of Roq1 22-60 examined here. The corresponding statement in the Discussion section (line 416) claiming that Roq1 22-60 is "not ubiquitinated" should be changed.

The cellular amounts of Roq1 versions harboring amino acid substitutions in the hydrophobic patch are increased when compared to other Roq1 variants expressed from the same promoter (Fig. 4A, Fig. S3B). The author's claim that this hydrophobic stretch represents a degron to regulate the amounts of the protein could be experimentally addressed, e.g. by cycloheximide decay assays.

The authors should investigate the function of their synthetic constructs comprising an amino-terminal arginine residue and a hydrophobic motif in yeast cells.

Referee #2:

The goal of this manuscript is to investigate the mechanism by which Roq1 alters the substrate specificity of Ubr1. This manuscript is a follow-up study to the authors' previous paper published in *Molecular Cell* (Vol. 70, p.1025, 2018). In the current study, the authors show that in addition to the N-terminal arginine of Roq1 (R22) that has been revealed previously as important in regulating Ubr1 activity, a short hydrophobic motif (YYFV, 55-58) in the Roq1(22-60) peptide is also crucial. The authors have employed a Roq1(V58E) mutant variant to characterize how this short Roq1 hydrophobic motif regulates Ubr1 activity, revealing that not only does the YYFV motif serve as a binding determinant, but it also regulates Ubr1 activity. Since the experimental systems deployed in this study are highly artificial, it is difficult to judge if the findings indeed reflect what occurs in vivo.

Major points:

1. The authors must use "full-length" Roq1 expressed from the endogenous locus (i.e., no overexpression) in vivo to demonstrate the role of Roq1's hydrophobic motifs in Ubr1 regulation.

In this study, the authors have mainly relied on in vitro experiments to decipher Roq1 function in vivo. Nevertheless, as mentioned by the authors in p.14, it is challenging to translate insights from in vitro studies into an understanding of Ubr1 function in vivo. As the authors state, "the outcome of Ubr1 reprogramming in vivo is likely more complex than can be recapitulated easily in reconstitution experiments. We show that Roq1 activates the ubiquitination of both misfolded proteins and

Cup9 in vitro. In contrast, Roq1 appears to suppress Cup9 degradation in vivo." (p.14, ln. 414).

Moreover, rather than using full-length Roq1 to deduce the function of Roq1, the authors have primarily used an "artificially shortened" version of Roq1, i.e., Roq1(22-60), for their in vitro reconstitution assays and in vivo reporter assays. The authors argue that Roq1(22-60) is sufficient to represent the function of wild-type Roq1. Nevertheless, Roq1(22-60) did not support Rho8*-GFP degradation when expressed under the Roq1 promoter (Fig. S5), and only partially supported Rho8*-GFP degradation when expressed under the strong GPD promoter (Fig. S1B), indicating that in fact Roq1(22-60) does not adequately represent wild-type Roq1.

Lastly, the authors applied a non-physiologically relevant protein concentration to examine the activity of Roq1(22-60) in their in vitro reconstitution assays, i.e., a "10-fold molar excess of Roq1(22-60) over Ubr1" (p.5, ln. 110). As revealed by the authors, varying the protein concentrations elicited different results: "Roq1(R22A) did not affect Pho8* ubiquitination when used at a 10-fold molar excess over Ubr1 (Figure 2A). However, further raising its concentration slightly enhanced Pho8* ubiquitination (Figure 2E, for example note the upward shift of ubiquitin-Pho8* in the lane with a 40-fold molar excess of Roq1(R22A) compared with the lane with a 10-fold molar excess). Therefore, Roq1(R22A) still has a limited ability to activate Ubr1" (p.7). Therefore, it is misleading to conclude endogenous Roq1 function based on reconstitution experiments that use a 10-fold molar excess of Roq1(22-60) over Ubr1. The authors need to establish what is the "endogenous" protein concentration of Roq1(22-104) as well as the molar ratio of endogenous Roq1 to Ubr1 in wild-type yeast cells after tunicamycin treatment. Based on the authors' own previous work (Mol. Cell 70, p.1025), Roq1 is intrinsically labile and so the endogenous protein abundance of Roq1 is likely to be low.

2. The authors completely rely on artificial reporter models, such as Rho8*-GFP, to measure Ubr1 activity. They need to test degradation of endogenous misfolded protein substrates to ensure that their conclusions remain the same.

3. The authors have concluded that, apart from binding, the hydrophobic motif of Roq1 exerts additional regulatory activity based solely on their characterization of the Roq1(V58E) mutant variant (Fig. 5 and Fig. 6). They argue that Roq1(V58E) binds Ubr1 almost as efficiently as wild-type Ubr1 (Fig. 4A). Nevertheless, despite that pulldown levels of Flag-Ubr1 look similar in Fig. 4A, the level of Roq1(V58E)-HA pulled down is much greater than that of wild-type Roq1-HA, suggesting that Roq1(V58E) binds Ubr1 less efficiently than wild-type Roq1. Therefore, the results from the Roq1(V58E) experiments are insufficient to infer that endogenous Roq1 exerts additional regulatory activity on Ubr1.

4. It is inappropriate to compare the activity of Roq1 mutant variants without considering their protein abundance.

(1) Control Western blots showing the protein abundances of Roq1 variants for Fig. 3F, 3G, 7B, 7D, S1B, S3C, S3D, and S5 are all lacking.

(2) The protein abundances of the Roq1 E36K, N45K and E67K variants are less than those of wild-type Roq1 (Fig. S3D).

Minor point

1. Labeling of the molecular weight markers is lacking for Figs. 1B-1G, 2A-2C, 2E, 5B-5F, 6A-6C, 7D, S1A, S1C-S1G, S2, and S4A.

Referee #3:

Peters et al. investigate how the Ubr1 ubiquitin ligase is modulated by the Roq1 pseudo-substrate using in vitro assays and validation of key findings in vivo. The authors find that processed Roq1 uses two SLiMs to interact with and modulate Ubr1: the N-terminal arginine, which likely interacts with the type-1 binding pocket of Ubr1, and a newly identified short hydrophobic motif. The authors argue that both SLiMs are required for Roq1 function, with distinct impacts on Ubr1 activity. The region between the SLiMs only functions as a linker, ensuring cooperative binding of the SLiMs needed for Roq1 function. Overall, the experiments are well performed and documented, and the manuscript is clearly written and a nice read.

In my opinion, the claims regarding the function of the linker region and cooperative binding should be strengthened as outlined below.

1. The authors propose a model where a single Roq1 molecule binds to Ubr1 using two SLiMs, and the linker region serves only to connect the two SLiMs for cooperative binding. According to the authors, one argument for this is that replacing the linker region with short artificial linkers preserves Roq1 function in vitro. However, from the data shown in Fig.7d it looks like while Roq1(22 aa), Roq1(GGS) and Roq1(GSP) are equally active, they are less active compared to the 22-60 variant. Can full length artificial linkers in the context of the 22-60 variant and/or full length Roq1 preserve Roq1 function in vitro and/or in vivo?

2. Based on the authors' model, a shortened linker is expected to reduce Roq1 activity. Less trivially, so should a substantially longer one, which the authors could test in vitro and/or in vivo. Moreover, it would be important to examine the impact of linker mutations and replacements in the context of full length Roq1 in vivo.

3. An alternative possibility is that Roq1 oligomerizes (for example via the linker region) and SLiMs from two Roq1 molecules interact with Ubr1. The gradual loss of Roq1 functionality with shorter linkers (Fig.7b) is consistent with this hypothesis. The authors could distinguish between the two models by testing how combining two Roq1 variants (each mutated in one of the SLiMs, with or without the linker region) affects SHRED and Ubr1 in vitro. And/or by combining Roq1(R22A) with arginine dipeptides in vitro. This could also help strengthen the argument for cooperative binding.

4. On the point of arginine dipeptides, the authors could use them to strengthen the point that Roq1 binds to the type-1 pocket of Ubr1.

Minor points

5. Molecular weight markers are missing on most immunoblots.

6. The authors state that "Roq1(R22A) did not affect Pho8* ubiquitination when used at a 10-fold molar excess over Ubr1 (Figure 2A). However, further raising its concentration slightly enhanced Pho8* ubiquitination (Figure 2E, for example note the upward shift of ubiquitin-Pho8* in the lane with a 40-fold molar excess of Roq1(R22A) compared with the lane with a 10-fold molar excess)." In these and other experiments there appears to be inhibition at amounts above 40-fold excess. A comment on this would be appreciated.

7. "Flexible linkers can span substantial distances. Assuming a C α -C α distance in the peptide backbone of 3.8 Å (Chakraborty et al, 2013), a maximally extended 15-residue linker would, theoretically, be more than 5 nm long." Not clear why the authors chose to refer to the artificially shortened linker here, as the natural >30 residue long one would make their point even more striking.

8. "Not shown" data should be shown or not referred to. For example, "Roq1(GGS) and Roq1(GSP) were inactive in vivo, possibly due to low stability (not shown)."

9. From the plasmid table, it is unclear what the sequences of the Roq1 variants in Fig.7A is. What exactly is deleted to shorten the linker region?

10. In the methods section, it is unclear how R-GFP and F-GFP are purified and prepared, (tag, protease, etc).

11. Whenever possible the authors should show some characterization of the purified proteins (e.g. Coomassie gels).

Point-by-point response

Referee 1

Peters et al. establish an in vitro assay to investigate re-programming of the ubiquitin ligase Ubr1 by the intrinsically-disordered protein Roq1. Using this assay they nicely show that a truncated version of Roq1 (aa 22-60) binds to Ubr1 and facilitates ubiquitylation of unfolded proteins in a process termed SHRED. Ubiquitylation of proteins harboring a bulky hydrophobic residue at the amino-terminus or containing an internal degradation site (degron) is also facilitated while the modification of proteins exposing a positively-charged amino-terminus is negatively affected. As already known, this largely depends on the binding of Roq1 to Ubr1 via an arginine residue at the amino-terminus. By experiments in yeast cells the authors identify a hydrophobic patch in Roq1 that is specifically required for reprogramming Ubr1 for the ubiquitylation of unfolded proteins but is not involved in modulating the activity of Ubr1 towards substrates containing an internal or an N-degron. This motif seems to constitute a second interaction site to Ubr1. Finally, the authors show that a construct comprising an arginine residue at the amino-terminus connected to a stretch of hydrophobic amino acids by a flexible linker region suffices to induce ubiquitylation of SHRED substrates by Ubr1.

Most of the experiments are well done and include appropriate controls. The authors provide solid evidence that a distinct binding site comprising a hydrophobic region in Roq1 selectively modulates the activity of the Ubr1 ubiquitin ligase towards unfolded proteins. This may be of importance for the design of compounds to regulate such events. Although the mechanism underlying this re-programming process remains enigmatic, the presented study should pave the way for future analyses. Some points should be experimentally addressed by the authors to increase the quality of the work and to better support their arguments. Most importantly, we lack information on the binding of Roq1 to substrates and on the stability and the Ubr1-binding activity of the truncated Roq1 22-60 version in yeast cells. Other than that, I recommend publication of a revised version of the presented work in the "EMBO Journal".

Major points:

1. Fig. S1G implies that Roq1 22-60 is slightly more active than Roq1 22-104 in the in vitro ubiquitination assay. This may be due to competition for components of the ubiquitination apparatus by Roq1 22-104 and the substrate. Still, the overexpression of Roq1 22-60 in yeast cells is required to rescue a roq1 knockout to re-program Ubr1 for SHRED (Fig. S1B and Fig. S5), whereas lower expression of Roq1 22-104 suffices for this. This may be due to reduced stability/amounts of Roq1 22-60 in yeast or the binding of Roq1 22-60 to Ubr1 in yeast may be compromised. The cellular levels of both proteins should be compared as well as their ability to bind Ubr1, e.g. by using HA-tagged versions of the proteins. These experiments are crucial for understanding the results of this work since the in vitro ubiquitylation assays were exclusively done with the truncated Roq1 protein and these data were directly correlated with yeast experiments involving the full-length construct.

This comment raises the valid question how our in vitro results with truncated Roq1(22-60) variants apply to the situation in vivo, where Roq1(22-104) is the active form. To clarify this issue, we have systematically introduced mutations that proved informative in the context of Roq1(22-60) in vitro into full-length Roq1 in yeast and then tested the activity of the resulting Roq1 variants in vivo. As a result, we now use the artificial Roq1(22-60) almost exclusively as an experimental tool for in vitro experiments and we corroborate any conclusions about the functional architecture of Roq1 using the natural Roq1(22-104) in yeast (new Figure 7). We believe that this strategy obviates any concerns associated with the use of Roq1(22-60).

2. In the end, such analysis may also provide information on the carboxy-terminus of Roq1 in regulating its cellular levels.

For technical reasons, we were unable to determine whether the carboxy-terminus of Roq1 is important for stability. In western blots, we observed that the short Roq1(22-60), which lacks the carboxy-terminus, was retained less well on nitrocellulose and PVDF membranes than the longer Roq1(22-104). This was true regardless of the pore size of the membrane used and also when we cross-linked the proteins with paraformaldehyde after blotting. Therefore, less Roq1(22-60) than Roq1(22-104) was detected even when equimolar amounts of the two proteins were run on SDS-PAGE gels (also see Figure EV1A and page 47, lines 1104-1110). This makes it very difficult to fairly assess the relative abundance of Roq1 variants with and without carboxy-terminal region.

3. Given the activity of Roq1_R22A in the ubiquitination assay in Fig. 2E, I wonder if adding excess amounts of a hydrophobic peptide alone can modulate the activity of Ubr1 towards the ubiquitination of unfolded proteins. Related to this: how does the expression/overexpression of the Roq1_R22A variant affect Rtn1-Pho8*-GFP reporter levels in yeast cells when compared to the other Roq1 constructs?

This would be a desirable experiment but adding large amounts of a hydrophobic peptide to the in vitro ubiquitination reactions would likely be problematic due to peptide insolubility and stickiness. Regarding the second point, we expressed Roq1(R22A) under the *GPD* promoter, which is one of the strongest promoters available in yeast. However, Roq1(R22A) was still unable to induce degradation of the SHRED reporter (new Figure EV3). It is possible that, even with the *GPD* promoter, we could not reach sufficiently high expression levels for Roq1(R22A) to show its residual activity (page 7, lines 143-144 and 167-169).

4. Does Roq1 bind unfolded proteins and, if so, is the hydrophobic patch involved in establishing this interaction? Although Roq1 does not seem to facilitate binding of Pho8*-MBP to Flag-Ubr1 (Fig. S4A) it could still assist in the positioning of the substrate.

We asked whether Roq1 affects binding of Pho8* to Ubr1 but did not test whether Roq1 binds Pho8*. We reasoned that detecting some affinity of the Roq1 hydrophobic motif for Pho8* would be expected because the misfolded Pho8* likely exposes hydrophobic residues. Therefore, it would be unclear what a binary hydrophobic Roq1-Pho8* interaction would mean for the Roq1-Ubr1-Pho8* complex. Only the atomic structure of the ternary

complex will clarify the issue. However, Roq1 could indeed assist in the positioning of the substrate and we have incorporated this scenario in the discussion (page 15, lines 423-426).

5. The crosslinking experiment shown in Fig. 4E should be complemented by a Roq1 variant containing the Bpa crosslinker at positions that are unlikely to directly interact with Ubr1. This would confirm the specificity of the observed crosslinks as indicators of the binding of Y55 and Y56 in Roq1 to Ubr1.

In addition to Y55 and Y56, we incorporated Bpa in another three positions situated in the linker or carboxy-terminal region of Roq1, and we obtained photo-crosslinks with all variants (see figure below). This may be surprising at first, but Roq1 is entirely disordered and hence more flexible than a typical globular protein. As a result, most or all residues of Roq1 may come sufficiently close to Ubr1 during the 15 x 1 second UV irradiation to produce crosslinks. Given this observation, we cannot conclude that R22 and the hydrophobic motif are the only motifs in Roq1 that make physical contact with Ubr1. We therefore removed the crosslinking data from the manuscript. However, the conclusion stands that the hydrophobic motif is required for strong binding of Roq1 to Ubr1 (see Figure 4).

Photo-crosslinking of Roq1-Ubr1. (A) Bpa was incorporated into Roq1 at position F42 or Y55. The resulting Roq1 variants were mixed with Ubr1, irradiated with UV light where indicated, immunoprecipitated and used for western blotting. **(B)** As in panel A but for Y55, L94 and G96. All four Roq1 variants yielded photo-crosslinks.

Minor points:

6. The authors use a carboxy-terminal-truncated Roq1 variant (Roq1 22-60) in their in vitro studies to circumvent ubiquitylation of a longer construct (Roq1 22-104) by Ubr1 and "to focus our analysis on the activity of Ubr1 towards designated substrate proteins". In Fig. S1A, they show that Roq1 22-60 is still ubiquitylated. I do not fully agree with the authors that this is to a substantially lesser extent than Roq1 22-104, considering the lower amounts of Roq1 22-60 examined here. The corresponding statement in the Discussion section (line 416) claiming that Roq1 22-60 is "not ubiquitinated" should be changed.

We agree and have changed our statement regarding Roq1 ubiquitination to "This modification appears dispensable because Roq1(22-60) is less extensively ubiquitinated than Roq1(22-104) but has even higher activity towards Ubr1" (page 16, lines 438-440). There is a technical issue here that makes the relevant Figure EV1A a little confusing. As explained above, the small Roq1(22-60)-HA is retained less well on nitrocellulose or PVDF membranes

than the longer Roq1(22-104)-HA. As a result, less Roq1(22-60)-HA is detected than Roq1(22-104)-HA, even though we used equimolar amounts of the two proteins. Nonetheless, we can still conclude that the amount of ubiquitinated Roq1(22-60) is lower than that of ubiquitinated Roq1(22-104). We now explain this issue in the figure legend (see page 47, lines 1104-1110).

7. The cellular amounts of Roq1 versions harboring amino acid substitutions in the hydrophobic patch are increased when compared to other Roq1 variants expressed from the same promotor (Fig. 4A, Fig. S3B). The author's claim that this hydrophobic stretch represents a degron to regulate the amounts of the protein could be experimentally addressed, e.g. by cycloheximide decay assays.

We thank the referee for prompting us to test this claim experimentally. We have done the suggested cycloheximide chase experiment and find that substitutions in the hydrophobic motif indeed raise Roq1 steady-state levels and slow its degradation. These new data support the idea that the hydrophobic motif acts as a degron (Figure 3F, G).

8. The authors should investigate the function of their synthetic constructs comprising an amino-terminal arginine residue and a hydrophobic motif in yeast cells.

We analyzed Roq1(22-104) variants in which most of the linker between R22 and the hydrophobic motif was replaced, shortened, or both in yeast (revised Figure 8A, B and new Figure EV5E-G). These in vivo experiments show that this linker sequence does not contain functionally crucial residues and can be severely shortened significantly without disturbing Roq1 function in vivo (see page 13, lines 343-359).

Referee 2

The goal of this manuscript is to investigate the mechanism by which Roq1 alters the substrate specificity of Ubr1. This manuscript is a follow-up study to the authors' previous paper published in Molecular Cell (Vol. 70, p.1025, 2018). In the current study, the authors show that in addition to the N-terminal arginine of Roq1 (R22) that has been revealed previously as important in regulating Ubr1 activity, a short hydrophobic motif (YYFV, 55-58) in the Roq1(22-60) peptide is also crucial. The authors have employed a Roq1(V58E) mutant variant to characterize how this short Roq1 hydrophobic motif regulates Ubr1 activity, revealing that not only does the YYFV motif serve as a binding determinant, but it also regulates Ubr1 activity. Since the experimental systems deployed in this study are highly artificial, it is difficult to judge if the findings indeed reflect what occurs in vivo.

Major points:

1. The authors must use "full-length" Roq1 expressed from the endogenous locus (i.e., no overexpression) in vivo to demonstrate the role of Roq1's hydrophobic motifs in Ubr1 regulation.

In this study, the authors have mainly relied on in vitro experiments to decipher Roq1 function in vivo. Nevertheless, as mentioned by the authors in p.14, it is challenging to translate insights from in vitro studies into an understanding of Ubr1 function in vivo. As the authors state, "the outcome of Ubr1 reprogramming in vivo is likely more complex than can be recapitulated easily in reconstitution experiments. We show that Roq1 activates the ubiquitination of both misfolded proteins and Cup9 in vitro. In contrast, Roq1 appears to suppress Cup9 degradation in vivo." (p.14, ln. 414).

Moreover, rather than using full-length Roq1 to deduce the function of Roq1, the authors have primarily used an "artificially shortened" version of Roq1, i.e., Roq1(22-60), for their in vitro reconstitution assays and in vivo reporter assays. The authors argue that Roq1(22-60) is sufficient to represent the function of wild-type Roq1. Nevertheless, Roq1(22-60) did not support Rtn1-Pho8-GFP degradation when expressed under the ROQ1 promoter (Fig. S5), and only partially supported Rtn1-Pho8*-GFP degradation when expressed under the strong GPD promoter (Fig. S1B), indicating that in fact Roq1(22-60) does not adequately represent wild-type Roq1.*

Lastly, the authors applied a non-physiologically relevant protein concentration to examine the activity of Roq1(22-60) in their in vitro reconstitution assays, i.e., a "10-fold molar excess of Roq1(22-60) over Ubr1" (p.5, ln. 110). As revealed by the authors, varying the protein concentrations elicited different results: "Roq1(R22A) did not affect Pho8 ubiquitination when used at a 10-fold molar excess over Ubr1 (Figure 2A). However, further raising its concentration slightly enhanced Pho8* ubiquitination (Figure 2E, for example note the upward shift of ubiquitin-Pho8* in the lane with a 40-fold molar excess of Roq1(R22A) compared with the lane with a 10-fold molar excess). Therefore, Roq1(R22A) still has a limited ability to activate Ubr1" (p.7). Therefore, it is misleading to conclude endogenous Roq1 function based on reconstitution experiments that use a 10-*

fold molar excess of Roq1(22-60) over Ubr1. The authors need to establish what is the "endogenous" protein concentration of Roq1(22-104) as well as the molar ratio of endogenous Roq1 to Ubr1 in wild-type yeast cells after tunicamycin treatment. Based on the authors' own previous work (Mol. Cell 70, p.1025), Roq1 is intrinsically labile and so the endogenous protein abundance of Roq1 is likely to be low.

We agree that it is crucial to show the importance of the hydrophobic motif using full-length Roq1 expressed from the endogenous locus. We therefore used the delitto perfetto approach (Storici and Resnick, 2006) to introduce the R22A, V58E and 4A mutations into the *ROQ1* locus without any other genomic modifications. We show that these mutations blocked Roq1 activity in SHRED, as did the deletion of *ROQ1* (new Figure 7A).

Regarding the abundance of Roq1 and Ubr1, we now provide evidence that Roq1 in yeast is much less abundant than Ubr1 under non-stress conditions but that its levels approach those of Ubr1 upon tunicamycin treatment (new Figure EV1G). Of note, this experiment can only give a rough estimate of the relative abundance of the two proteins. We tagged both Roq1 and Ubr1 with an ALFA tag in their endogenous gene loci and analyzed them by a western blot from the same cell lysates. However, given their different sizes (less than 15 kDa versus over 200 kDa) it is uncertain if they transferred from the SDS-PAGE gel onto the nitrocellulose membrane with the same efficiency. Furthermore, it is possible that the C-terminal ALFA tag interfered with the proteolytic processing of Roq1 by Ynm3 (see page 48, lines 1130-1136). While this experiment indicates that a 1:1 molar ratio of Roq1 and Ubr1 is a reasonable approximation for the situation in stressed yeast, we believe that using an excess of Roq1 for in vitro experiments has advantages. It ensures, to the extent possible, that the in vitro ubiquitination assays measure the activity of the Roq1-Ubr1 complex and not a mixture of the Roq1-Ubr1 complex and unbound Ubr1. This is important to reveal the regulatory capacity of Roq1 variants with reduced Ubr1 affinity, such as Roq1(V58E) (see page 5/6, lines 103-108).

2. The authors completely rely on artificial reporter models, such as Rtn1-Pho8*-GFP, to measure Ubr1 activity. They need to test degradation of endogenous misfolded protein substrates to ensure that their conclusions remain the same.

We do not have such substrates at our disposal. We previously identified endogenous SHRED substrates, i.e. proteins that become Roq1-dependent Ubr1 substrates during stress (Szoradi et al, 2018). The extent of their degradation is minor (likely because only a small fraction of an endogenous substrate misfolds even during stress), their degradation does not exclusively depend on SHRED (likely because of redundancy between protein quality control pathways), or both. Therefore, these substrates are prohibitively inefficient SHRED reporters. In contrast, Rtn1-Pho8*-GFP is extensively degraded in a stress-, Roq1- and Ubr1-dependent manner, making it an excellent tool to investigate the mechanism of the SHRED pathway. Defining the physiological substrate spectrum of SHRED is an issue outside the scope of this study.

3. The authors have concluded that, apart from binding, the hydrophobic motif of Roq1 exerts additional regulatory activity based solely on their characterization of the Roq1(V58E) mutant variant (Fig. 5 and Fig. 6). They argue that Roq1(V58E) binds Ubr1

almost as efficiently as wild-type Ubr1 (Fig. 4A). Nevertheless, despite that pulldown levels of Flag-Ubr1 look similar in Fig. 4A, the level of Roq1(V58E)-HA pulled down is much greater than that of wild-type Roq1-HA, suggesting that Roq1(V58E) binds Ubr1 less efficiently than wild-type Roq1. Therefore, the results from the Roq1(V58E) experiments are insufficient to infer that endogenous Roq1 exerts additional regulatory activity on Ubr1.

This is an important point. In vitro, Roq1(V58E) retains substantial affinity for Ubr1, still inhibits ubiquitination of a type-1 N-degron substrate, still promotes ubiquitination of a type-2 N-degron substrate but cannot properly promote ubiquitination of misfolded proteins (Figure 5). These data indicate that the V58E mutation specifically disrupts the ability of Roq1 to promote the recognition of misfolded proteins by Ubr1. This is in contrast to the R22A and 4A mutations, which strongly reduce binding of Roq1 to Ubr1 and thus prevent Roq1 from impacting any Ubr1 activity. To test if these conclusions hold true in vivo, we used strains in which the R22A, V58E and 4A mutations had been introduced into the endogenous *ROQ1* gene locus (see point 1, above), and we tested the activities of the resulting Roq1 variants towards the SHRED reporter, a type-1 N-degron reporter and a type-2 N-degron reporter. As mentioned above, all of the mutations destroyed the ability of Roq1 to induce degradation of the SHRED reporter (new Figure 7A). Furthermore, both Roq1(R22A) and Roq1(4A) had only a minor impact on the stability of the N-degron reporters. By contrast, Roq1(V58E) more strongly stabilized the type-1 N-degron reporter than Roq1(R22A) and Roq1(4A) (new Figure 7B), and it destabilized the type-2 N-degron reporter at least as strongly as wild-type Roq1 (new Figure 7C). Therefore, the V58E mutation had the same effect as in vitro: it stopped Roq1 from promoting the recognition of misfolded proteins by Ubr1 but still allowed binding of Roq1 to Ubr1 via the remainder of the hydrophobic motif (also see Figure 4A). As a result, Roq1(V58E) still competed with type-1 N-degron substrates for binding to the Ubr1 type-1 site and, via type-1 site binding, still activated recognition of type-2 N-degron substrates (see page p11/12, lines 297-329).

4. It is inappropriate to compare the activity of Roq1 mutant variants without considering their protein abundance.

(1) Control Western blots showing the protein abundances of Roq1 variants for Fig. 3F, 3G, 7B, 7D, S1B, S3C, S3D, and S5 are all lacking.

(2) The protein abundances of the Roq1 E36K, N45K and E67K variants are less than those of wild-type Roq1 (Fig. S3D).

This comment refers to three types of Roq1 variants, namely (1) Roq1(22-60), (2) point mutants in Roq1(22-104), and (3) Roq1 variants with deletions or mutations in the linker sequence between R22 and the hydrophobic motif. Different considerations apply to each:

(1) Unfortunately, Roq1(22-60) levels (relevant for Figure EV1B, formerly S1B) are difficult to assess properly by western blotting. We found that the small Roq1(22-60) is not retained well on nitrocellulose or PVDF membranes, regardless of the pore size of the membrane and also when we chemically crosslinked proteins after blotting. Hence, the amount of Roq1(22-60) compared to Roq1(22-104) will be underestimated. This technical problem is apparent also in Figure EV1A, which shows a weaker signal for Roq1(22-60) than for Roq1(22-104)

even though equimolar amounts of the two proteins were analyzed (page 47, lines 1104-1110). However, overexpressed Roq1(22-60) is nearly as active as overexpressed wild-type Roq1 in vivo (Figure EV1B), justifying its use for in vitro assays.

(2) In the genetic screen and subsequent follow-up experiments, mutant versions of Roq1(22-104)-HA, including the E36K, N45K and E67K variants, were expressed under the strong *GPD* promoter (Figures 3 and Figure EV4E-F, formerly S3C-D). As a result, all Roq1 mutants shown were detectable with an anti-HA antibody by western blotting (see Figure EV4D, formerly S3B). Since wild-type HA-tagged Roq1 expressed under the much weaker endogenous *ROQ1* promoter is undetectable by western blotting (see Figure 4A), all point mutants had higher expression levels than native Roq1. They therefore were sufficiently abundant for activity and we conclude that all mutant variants have genuine SHRED defects (see page 8, line 190-191).

(3) The referee is right that the activities of Roq1 variants with shortened and mutated linkers in Figure 8B (previously Figure 7B) could not be properly assessed without information on their expression levels. To clarify this issue, we now provide western blots for the various constructs (new Figure EV5E-G). We show that shortening the linker progressively destabilized Roq1 and that replacing the remaining linker in functional Roq1(66aa) with generic linkers both reduced expression levels further and impaired function. For these variants, called Roq1(66aa_{G5}) and Roq1(66aa_{GSP}), it was no longer possible to decide from in vivo experiments whether they were inactive or simply not abundant enough to function. To answer this question, we then turned to in vitro experiments, in which we used equimolar amounts of different Roq1 variants (Figure 8D, formerly Figure 7D; see page 13, lines 353-369).

Minor point:

1. Labeling of the molecular weight markers is lacking for Figs. 1B-1G, 2A-2C, 2E, 5B-5F, 6A-6C, 7D, S1A, S1C-S1G, S2, and S4A.

We have added molecular weight markers to all immunoblots.

Referee 3

Peters et al. investigate how the Ubr1 ubiquitin ligase is modulated by the Roq1 pseudo-substrate using in vitro assays and validation of key findings in vivo. The authors find that processed Roq1 uses two SLiMs to interact with and modulate Ubr1: the N-terminal arginine, which likely interacts with the type-1 binding pocket of Ubr1, and a newly identified short hydrophobic motif. The authors argue that both SLiMs are required for Roq1 function, with distinct impacts on Ubr1 activity. The region between the SLiMs only functions as a linker, ensuring cooperative binding of the SLiMs needed for Roq1 function. Overall, the experiments are well performed and documented, and the manuscript is clearly written and a nice read.

In my opinion, the claims regarding the function of the linker region and cooperative binding should be strengthened as outlined below.

Major points:

1. The authors propose a model where a single Roq1 molecule binds to Ubr1 using two SLiMs, and the linker region serves only to connect the two SLiMs for cooperative binding. According to the authors, one argument for this is that replacing the linker region with short artificial linkers preserves Roq1 function in vitro. However, from the data shown in Fig.7d it looks like while Roq1(22 aa), Roq1(GGS) and Roq1(GSP) are equally active, they are less active compared to the 22-60 variant. Can full length artificial linkers in the context of the 22-60 variant and/or full length Roq1 preserve Roq1 function in vitro and/or in vivo?

2. Based on the authors' model, a shortened linker is expected to reduce Roq1 activity. Less trivially, so should a substantially longer one, which the authors could test in vitro and/or in vivo. Moreover, it would be important to examine the impact of linker mutations and replacements in the context of full length Roq1 in vivo.

As suggested by the referee, we tested linker mutations in the context of full-length Roq1 in vivo (revised Figure 8A, B). This was part of our strategy to restrict the use of Roq1(22-60) to in vitro experiments and confirm any conclusions reached with Roq1(22-60) by in vivo experiments with the natural Roq1(22-104) (also see response to referee 1, point 1, above).

These experiments showed that the weakly conserved sequence between S23 and P51 can be replaced with an artificial linker of equal length or shortened from 27 to 10 residues without loss of SHRED activity. Linker shortening decreased Roq1 stability, though (new Figure EV5F), and replacing the 10 remaining residues in shortened Roq1(66aa) with generic linker sequences reduced Roq1 stability even further and impaired its function (see Roq1(66aa_{GGS}) and Roq1(66aa_{GSP}) in revised Figure 8B and new Figure EV5G). However, the complementary in vitro analysis using Roq1(22-60) showed that the shortened artificial linkers supported Roq1 activity (Figure 8C, D). The new data strengthen the model that the sequence between R22 and the hydrophobic motif simply serves as a linker, although its absolute minimal length remains to be defined (see page 13, lines 343-369).

3. An alternative possibility is that Roq1 oligomerizes (for example via the linker region) and SLiMs from two Roq1 molecules interact with Ubr1. The gradual loss of Roq1 functionality with shorter linkers (Fig.7b) is consistent with this hypothesis. The authors could distinguish between the two models by testing how combining two Roq1 variants (each mutated in one of the SLiMs, with or without the linker region) affects SHRED and Ubr1 in vitro. And/or by combining Roq1(R22A) with arginine dipeptides in vitro. This could also help strengthen the argument for cooperative binding.

We have done the suggested experiment and found no restoration of activity when we combined Roq1(R22A) and Roq1(4A) or when we combined Roq1(R22A) and the RA dipeptide (new Figure EV5C, D). It thus appears that R22 and the hydrophobic motif need to be part of the same molecule to cooperate (see page 12/13, lines 333-341).

4. On the point of arginine dipeptides, the authors could use them to strengthen the point that Roq1 binds to the type-1 pocket of Ubr1.

We previously provided in vivo evidence that Roq1 binds to the Ubr1 type-1 site, which was obtained by co-immunoprecipitation experiments with Roq1 WT/R22A and Ubr1 WT/G173R (Figure 5C in Szoradi et al, 2018). We are therefore unsure if additional in vitro evidence is needed and ask the referee for advice. If considered helpful, we could refer the reader to a competition experiment we did as a control for the activity of RA dipeptides (new Figure EV5C). In that experiment, we showed that RA dipeptides interfered with the ubiquitination of R-GFP, a Ubr1 substrate recognized through the type-1 site. Roq1 had the same effect, consistent with the notion that it interacts with the Ubr1 type-1 site. By contrast, Roq1(R22A), which lacks the arginine needed for type-1 site binding, did not affect R-GFP ubiquitination.

Minor points:

5. Molecular weight markers are missing on most immunoblots.

We have added molecular weight markers to all immunoblots.

6. The authors state that "Roq1(R22A) did not affect Pho8* ubiquitination when used at a 10-fold molar excess over Ubr1 (Figure 2A). However, further raising its concentration slightly enhanced Pho8* ubiquitination (Figure 2E, for example note the upward shift of ubiquitin-Pho8* in the lane with a 40-fold molar excess of Roq1(R22A) compared with the lane with a 10-fold molar excess)." In these and other experiments there appears to be inhibition at amounts above 40-fold excess. A comment on this would be appreciated.

It is unclear whether the difference in Pho8* ubiquitination between samples with a 40- or 80-fold excess of Roq1(R22A) reflects inherent assay variability or an inhibitory effect of excessive Roq1(R22A) (see Figure 2E). Similarly, it is unclear whether the apparent decrease in Pho8* ubiquitination with increasing amounts of Roq1 results from random sample-to-sample variation or the fact that extensive ubiquitination can make Pho8* so large that fails

to enter the SDS-PAGE gel (see Figure EV1F). We would like to avoid speculation and therefore simply point out that a 40- or 80-fold molar excess of Roq1(R22A) results in stronger Pho8* ubiquitination than a 10-fold molar excess (see page 7, lines 165-166) and that Roq1 activates Ubr1 already when present at equimolar amounts (see page 5, lines 101-103).

7. "Flexible linkers can span substantial distances. Assuming a C α -C α distance in the peptide backbone of 3.8 Å (Chakraborty et al, 2013), a maximally extended 15-residue linker would, theoretically, be more than 5 nm long." Not clear why the authors chose to refer to the artificially shortened linker here, as the natural >30 residue long one would make their point even more striking.

We thank the referee for this suggestion, which we have incorporated in the discussion (see page 15, lines 407-415).

8. "Not shown" data should be shown or not referred to. For example, "Roq1(GGS) and Roq1(GSP) were inactive in vivo, possibly due to low stability (not shown)."

The revised manuscript no longer contains any 'data not shown'. The question these data addressed is now being dealt with in the revised Figure 8B and the new Figure EV5G.

9. From the plasmid table, it is unclear what the sequences of the Roq1 variants in Fig.7A is. What exactly is deleted to shorten the linker region?

We have included this information in the revised Figure 8A (formerly Figure 7A).

10. In the methods section, it is unclear how R-GFP and F-GFP are purified and prepared, (tag, protease, etc).

We have revised the text to make this clear (see page 30, lines 566-569).

11. Whenever possible the authors should show some characterization of the purified proteins (e.g. Coomassie gels).

We have added these data as a new supplementary figure (see Appendix Figure S1).

Dr. Sebastian Schuck
Heidelberg University
Biochemistry Center
Im Neuenheimer Feld 328
Baden-Württemberg 69120
Germany

20th Jan 2025

Re: EMBOJ-2024-118461R
Reprogramming of the ubiquitin ligase Ubr1 by intrinsically disordered Roq1 through cooperating multifunctional motifs

Dear Dr. Schuck,

Thank you for submitting your revised manuscript for our consideration. All three original referees have now assessed it once more, and except for a minor comment by referee 1 (which I do not think requires further modification) they are fully satisfied with the revisions. Following incorporation of the following remaining editorial issues, we shall therefore be happy to accept the study for The EMBO Journal:

- Most importantly, we still need you to complete and upload the Source Data Checklist that had been sent to you by our Source Data curator, Hannah Sonntag (I am attaching it once more to this message). Please also double-check to make sure that all requested Source Data items have been uploaded.
- Please enter valid email addresses for all coauthors in the submission system, so that they could be informed about the submission and final decision. At resubmission, acknowledgement emails failed to be delivered to Rafael Salazar Claros.
- Please collate all Appendix Figures and Appendix Tables, together with their respective legends next to it, in a dedicated PDF called "Appendix", and headed by a brief Table of Contents listing the article title and the contents with page numbers. Please make sure to correctly name these figures "Appendix Figure S1/2/3..." both within the Appendix and when referencing them in the main text.
- Please include the Conflict of Interest information as a separate section, headed "Disclosure and Competing Interests Statement", in accordance with our updated Guide to Authors (<https://www.embopress.org/competing-interests>)
- As we are switching from a free-text author contribution statement towards a more formal statement based on Contributor Role Taxonomy (CRediT) terms, please remove the present Author Contribution section and instead specify each author's contribution(s) directly in the Author Information page of our submission system during upload of the final manuscript. See <https://casrai.org/credit/> for more information.
- The reference to Figure 8F should likely read Fig 8E - please check and correct.
- Please remove the Reagents & Tools Table from the manuscript text file, it should only be present as a separate upload.
- Finally, during routine pre-acceptance checks, our data editors have raised the following queries regarding figures, data, and legends, which I would ask you to address (ideally using the Track Changes option):
 1. Please note that information related to N is missing in the legends of figures EV4 B.
 2. Please note that the measure of center for the error bars needs to be defined in the legends of figures EV4 B.

I am therefore returning the manuscript to you for a final round of revision, to allow you to make these modifications and upload the revised files. Once we will have received them, we should be ready to swiftly proceed with formal acceptance and production of the manuscript.

With kind regards,

Hartmut

*** PLEASE NOTE: All revised manuscripts are subject to initial checks for completeness and adherence to our formatting guidelines. Revisions may be returned to the authors and delayed in their editorial re-evaluation if they fail to comply to the following requirements (see also our Guide to Authors for further information):

9) To facilitate reproducibility and cross-laboratory adoption of methodologies, please structure the Materials & Methods section as outlined in our guide to authors, including a completed Reagents and Tools Table that can be downloaded from our author guidelines as well (<https://www.embopress.org/page/journal/14602075/authorguide#structuredmethods>).

10) Digital image enhancement is acceptable practice, as long as it accurately represents the original data and conforms to community standards. If a figure has been subjected to significant electronic manipulation, this must be clearly noted in the figure legend and/or the 'Materials and Methods' section. The editors reserve the right to request original versions of figures and the original images that were used to assemble the figure. Finally, we generally encourage uploading of numerical as well as gel/blot image source data; for details see: embopress.org/page/journal/14602075/authorguide#sourcedata

At EMBO Press, we ask authors to provide source data for the main manuscript figures. Our source data coordinator will contact you to discuss which figure panels we would need source data for and will also provide you with helpful tips on how to upload and organize the files.

In the interest of ensuring the conceptual advance provided by the work, we recommend submitting a revision within 3 months (20th Apr 2025). Please discuss the revision progress ahead of this time with the editor if you require more time to complete the revisions. Use the link below to submit your revision:

Link Not Available

Referee #1:

In my first review, I was already quite positive about publishing the manuscript entitled "Reprogramming of the ubiquitin ligase Ubr1 ..." by Peters et al. in "EMBO Journal". Now, in a revised version, the authors provide additional experimental material that adequately addresses most of my initial criticisms. For various technical reasons, the authors still use different Roq1 constructs in their in vitro analyses and in their studies in yeast, which limits the interpretation of some of their results. This point should be emphasized more strongly in the discussion. Otherwise, I support publishing the revised manuscript in "EMBO Journal".

Referee #2:

The authors have addressed my comments and made significant improvements to their manuscript.

Referee #3:

The authors have adequately addressed all my comments and suggestions.

Dr. Sebastian Schuck
Heidelberg University
Biochemistry Center
Im Neuenheimer Feld 328
Baden-Württemberg 69120
Germany

24th Jan 2025

Re: EMBOJ-2024-118461R1
Reprogramming of the ubiquitin ligase Ubr1 by intrinsically disordered Roq1 through cooperating multifunctional motifs

Dear Dr. Schuck,

Thank you for submitting your final revised manuscript for our consideration. I am pleased to inform you that we have now accepted it for publication in The EMBO Journal.

Yours sincerely,

Hartmut Vodermaier
